# Genome-Scale Metabolic Models and Machine Learning Reveal Genetic Determinants of Antibiotic Resistance in *Escherichia coli* and Unravel the Underlying Metabolic Adaptation Mechanisms

Nicole Pearcy,[a] Yue Hu,[a] Michelle Baker,[a] Alexandre Maciel-Guerra,[a,b] Ning Xue,[a] ⓘ Wei Wang,[c] Jasmeet Kaler,[a] Zixin Peng,[c] ⓘ Fengqin Li,[c] ⓘ Tania Dottorini[a]

aSchool of Veterinary Medicine and Science, University of Nottingham, Sutton Bonington, Leicestershire, United Kingdom
bSchool of Computer Science, University of Nottingham, Jubilee Campus, Nottingham, Nottinghamshire, United Kingdom
cNHC Key Laboratory of Food Safety Risk Assessment, China National Center for Food Safety Risk Assessment, Beijing, China

Nicole Pearcy and Yue Hu contributed equally to this work. The order was determined by increasing seniority on the project.

**ABSTRACT** Antimicrobial resistance (AMR) is becoming one of the largest threats to public health worldwide, with the opportunistic pathogen *Escherichia coli* playing a major role in the AMR global health crisis. Unravelling the complex interplay between drug resistance and metabolic rewiring is key to understand the ability of bacteria to adapt to new treatments and to the development of new effective solutions to combat resistant infections. We developed a computational pipeline that combines machine learning with genome-scale metabolic models (GSMs) to elucidate the systemic relationships between genetic determinants of resistance and metabolism beyond annotated drug resistance genes. Our approach was used to identify genetic determinants of 12 AMR profiles for the opportunistic pathogenic bacterium *E. coli*. Then, to interpret the large number of identified genetic determinants, we applied a constraint-based approach using the GSM to predict the effects of genetic changes on growth, metabolite yields, and reaction fluxes. Our computational platform leads to multiple results. First, our approach corroborates 225 known AMR-conferring genes, 35 of which are known for the specific antibiotic. Second, integration with the GSM predicted 20 top-ranked genetic determinants (including *accA*, *metK*, *fabD*, *fabG*, *murG*, *lptG*, *mraY*, *folP*, and *glmM*) essential for growth, while a further 17 top-ranked genetic determinants linked AMR to auxotrophic behavior. Third, clusters of AMR-conferring genes affecting similar metabolic processes are revealed, which strongly suggested that metabolic adaptations in cell wall, energy, iron and nucleotide metabolism are associated with AMR. The computational solution can be used to study other human and animal pathogens.

**IMPORTANCE** *Escherichia coli* is a major public health concern given its increasing level of antibiotic resistance worldwide and extraordinary capacity to acquire and spread resistance via horizontal gene transfer with surrounding species and via mutations in its existing genome. *E. coli* also exhibits a large amount of metabolic pathway redundancy, which promotes resistance via metabolic adaptability. In this study, we developed a computational approach that integrates machine learning with metabolic modeling to understand the correlation between AMR and metabolic adaptation mechanisms in this model bacterium. Using our approach, we identified AMR genetic determinants associated with cell wall modifications for increased permeability, virulence factor manipulation of host immunity, reduction of oxidative stress toxicity, and changes to energy metabolism. Unravelling the complex interplay between antibiotic resistance and metabolic rewiring may open new opportunities to understand the ability of *E. coli*, and potentially of other human and animal pathogens, to adapt to new treatments.

Address correspondence to Tania Dottorini, tania.dottorini@nottingham.ac.uk.

Genome-scale metabolic models and machine learning reveal metabolic adaptation mechanisms to #ABR. #AntimicrobialResistance #MachineLearning @DottoriniResLab @NicolePearcy10 @InnovateUK

KEYWORDS antimicrobial resistance, *Escherichia coli*, genome-scale metabolic model, machine learning

Antimicrobial resistance is a major threat to global health. Worryingly, a growing number of pathogens exhibit an extraordinary capacity for acquiring new antibiotic resistance traits in the bacterial population worldwide (1). New multidrug resistance mechanisms have emerged and spread globally, resulting in current treatments becoming less effective against common bacteria that cause severe and often deadly infections. Consequently, the development of new drugs and novel treatment strategies is urgently needed (2, 3).

The opportunistic pathogen *Escherichia coli* plays a major role in the antimicrobial resistance (AMR) global health crisis. First, the ability of *E. coli* to acquire resistance via single nucleotide polymorphisms (SNPs) in its existing genome (4–7) and via acquisition of resistance genes through horizontal gene transfer (HGT) from surrounding species (8–10) has resulted in increased levels of resistance to many antibiotic classes, including penicillins, carbapenems, cephalosporins, fluoroquinolones, aminoglycosides, and tetracyclines (11–15). Second, the ease of its transmission from humans and environmental sources has resulted in alarming numbers of multidrug-resistant *E. coli* strains being reported worldwide (16, 17). Third, the ease by which the bacteria can transfer genetic material via HGT, combined with the bacterium's ability to colonize different environments, including the gut where it has particularly close interaction with many other species, allows *E. coli* to act as a reservoir of AMR genes for other opportunistic pathogens, while also acquiring further resistance (18–21). For these reasons, the World Health Organization (WHO) has recently classified *E. coli* as a critical priority pathogen for which the development of a new treatment is high priority (22).

Recent advances in data generation and data mining, combined with machine learning (ML), have led to invaluable results in the identification of specific genomic markers which could be used to effectively predict resistant strains and to detect AMR genes (23–30). Most of these methods work to identify known AMR mutations giving rise to phenotypic resistance. This has great potential for fast diagnostic evaluation of bacteria compared to laboratory methods. Furthermore, ML-based approaches offer further powerful opportunities compared to conventional methods, as they allow for the genome-wide identification of truly novel features (i.e., k-mers and SNPs) ranked on their strength of correlation with the resistance phenotype. Recently, several studies have used these approaches (29, 30), which not only allow the identification of genes with known functional relationship with the resistance phenotype but also allow the identification of genes which have no prior association with a specific resistance phenotype. This creates a path for generating nonintuitive testable hypotheses about the association of antibiotic resistance to a wider repertoire of genes, including deletions and functional mutations altering metabolism, and therefore provides a significant advantage in comparison with the conventional use of annotated gene databases.

Recent findings have shown the interconnectivity of antibiotic resistance with metabolism and emphasize the importance of considering this relationship in the design of new antibiotic regimens (31–33). Through its ease of HGT, *E. coli* has been able to adopt a highly flexible carbon and energy metabolism for adaptation against stresses in niche environments (34, 35). For this reason, the bacterium is an ideal organism for investigating the interplay between AMR and metabolic adaptation mechanisms. Connecting antimicrobial genes and specific mutations and alleles to metabolic phenotypes, however, still remains a significant challenge (36, 37). Black-box ML predictions lack biological interpretation of the genetic determinants (30), and therefore, previous methods have often not accounted for the characterization of new advantageous genetic variants occurring in targets beyond annotated drug resistance genes (29, 38), therefore neglecting important metabolic adaptations that allow resistance and tolerance to antibiotic stress (39–41).

A genome-scale metabolic model (GSM) offers a way of mechanistically evaluating the genetic determinants identified using ML. A GSM is a computational model of metabolism, which includes all known biochemical reactions and their corresponding gene-protein-reaction (GPR) rules. The GPR rules provide important information linking genes to the reactions

that are catalyzed by the enzymes they encode and provide a means of simulating the metabolic system-level behavior of the bacteria to perturbations in the gene. While GSMs have proven invaluable tools for predicting genotype-phenotype relationships (42), they lack the power of machine learning algorithms (30). Recent studies have therefore been developing new approaches that integrate the power of ML with GSMs to allow for a mechanistic interpretation of the genetic associations discovered by machine learning, which offers a significant advantage over ML approaches alone (30, 43).

In this study, we developed a computational solution integrating the discriminant power of ML with GSM models to reveal the systemic relationships connecting the genetic determinants of AMR to important metabolic evolutionary adaptations in *E. coli*. Using our approach, first we were able to accurately predict AMR resistant and susceptible phenotypes against 11 out of 12 different antibiotics, as well as identifying 225 (35 of which matched the specific antibiotic class reported in AMR-related databases) known AMR-conferring genes in 3,616 *E. coli* strains. Second, by elucidating the effects of genetic discriminants on bacterial growth, metabolite yields, and biochemical fluxes using the GSM, we were able to relate genetic determinants to a number of metabolic adaptation mechanisms, including reduced growth, alternative carbon source utilization, changes to energy metabolism, iron metabolism, nucleotide metabolism, and modifications to cell wall metabolism.

## RESULTS

**Framework of the computational pipeline that combines machine learning with genome-scale metabolic models.** To identify the genomic features correlated with the selected AMR phenotypes and to interpret the systemic relationships between genetic determinants of resistance and metabolism, we developed a computational pipeline that combines ML with genome-scale metabolic models (see Fig. S1 in the supplemental material). A set of unique *E. coli* genomes for which AMR testing and metadata were available from public databases was selected. To efficiently analyze the AMR phenotypic variability that is likely to arise from a combination of SNPs and changes in gene content, we used an integrated k-mer and SNP-based ML approach. A gradient boosting classifier (GBC) (44, 45) was chosen as it is a powerful approach to quickly and efficiently scan entire genomes against selected phenotypes, allowing for the identification of arbitrary numbers of genomic features ranked on strength of correlation with the antimicrobial-resistant and -susceptible phenotype. The ML approach offers the opportunity to identify genes and/or mutations which, individually or in combinations, feature a strong correlation with resistance to antibiotics. A set of thresholds were applied to select only the top-ranked AMR genetic determinants strongly contributing to the performance of the ML classifier. The interconnectivity of antibiotic resistance, antimicrobial genes, and specific mutations and alleles to metabolic phenotypes, as well as the identification of new advantageous genetic variants occurring in targets beyond annotated drug resistance genes was determined using the GSM (Fig. S1). Flux balance analysis (FBA), a constraint-based approach, was used to predict the effects of the genetic determinants on the metabolic network. Importantly, we considered the protein-coding regions only in the ML classifiers, and therefore, the genetic variants are potentially increasing or decreasing enzymatic activity, or in some cases completely block the function of the gene. Here, we evaluated the effect of each genetic determinant by blocking the flux through its corresponding enzyme and assessed the propagation of this "loss of function" through the entire metabolic network. Specifically, we used the GSM to predict the effect of each genetic determinant on bacterial growth, production of individual metabolites, and the feasible flux range through individual reactions. Changes to metabolic phenotype capabilities in each gene knockout model (i.e., reduction in growth rate, reduced metabolite production or reduction in flux span through a reaction) were assessed using the wild-type model of *E. coli* K-12 MG1655.

**Genomic and metadata characteristics of the *E. coli* cohort.** Our first goal was to characterize the genetic content and diversity of *E. coli* strains. We selected a set of

3,616 unique *E. coli* genomes for which AMR testing and metadata were available from the Pathosystems Resource Integration Centre (PATRIC) (46).

Importantly, the genome sequences of these strains were all listed in PATRIC as "good" quality assemblies, had less than 250 contigs, and were labeled as either "WGS" (for whole genome sequenced) or "Complete" as the genome status in PATRIC. The genomes have experimentally measured AMR phenotypes, which are annotated as either "susceptible" or "resistant."

These isolates included a wide variety of geographic locations (see Fig. S2a) and AMR phenotypes for a diverse set of antibiotic classes, including penams (ampicillin), carbapenems (meropenem), monocyclic beta-lactam (aztreonam), cephalosporins (cefoxitin, cefepime, and cefuroxime), fluoroquinolones (ciprofloxacin and levofloxacin), aminoglycosides (gentamicin and tobramycin), diaminopyrimidines (trimethoprim), and tetracyclines (tetracycline). The number of resistant strains for each of the 12 individual antibiotics ranged between 427 (levofloxacin) and 2,600 (ciprofloxacin) of the 3,616 strains (see Fig. S2b).

Next, the pan-genome was extracted for the selected strains using the default parameters in Roary version 3.13.0 (47), which classified the catalogue of annotated genes as either core (i.e., occurring in >99% of strains) or accessory (i.e., occurring in <99% of strains).

**k-mer-based genomic feature selection through a gradient boosting classifier model identifies AMR-conferring genes.** The next goal of this analysis was to identify features in the genome sequence of each isolate which strongly correlated with resistance or susceptibility to each of the 12 antibiotics described above. To this aim, we implemented a gradient boosting classifier model for each antibiotic studied. DNA segments (k-mers) of 13 bp long were used as features in the classifiers, with the AMR phenotype used as the model labels (resistant or susceptible). For each classifier, 10,000 features were selected based on the chi-square test. We used the performance metrics accuracy, area under the receiver operator characteristic curve (AUC), precision, and recall to evaluate each model. A synthetic minority oversampling technique (SMOTE) was used to reduce the impact of unbalanced classes in the antimicrobial label groups and achieve robust classification results. The performance metrics were calculated as the mean of 50 simulations (Fig. 1). The performance metrics for the 12 antibiotics ranged from 90% to 99% for accuracy, 75% to 98% for precision, 62% to 95% for recall, and 88% to 98% for AUC. All antibiotics except meropenem and cefuroxime achieved an AUC of >95%. Features were selected from the remaining 10 AMR classifiers based on this AUC threshold.

The maximum importance in the 50 runs was captured for each k-mer. To identify important genes, the k-mers with a maximum importance score greater than 0 for each antibiotic model (as assigned by the GBC), were cross-referenced to the pan-genome of the 3,616 genomes. The identified k-mers, their corresponding genes, and maximum importance scores obtained by the GBC are shown in Table S1 and Fig. S3 in the supplemental material. When mapped to the CARD (48) and MutationDB databases (49), 84 unique AMR genes were identified in the top 10% of features (ranked according to the maximum weight found in the 50 runs), 25 of which had evidence of the AMR gene for the specific antibiotic class (Table 1).

**SNP-based machine learning approach uncovers additional and different AMR genetic determinants.** Together with the k-mer-based approach, we also analyzed the contribution of SNPs to the acquisition of drug resistance phenotypes by using them as features in a ML approach to find correlations with resistance or susceptibility to specific antibiotics. The variant sites (SNPs) in the protein-coding genes of the core genome of the pan-genome were identified using the SNPsites tool (www.github.com/sanger-pathogens/snp-sites) and used as the features in the GBC model for fitting AMR labels. A synthetic minority oversampling technique was applied to oversample data of the minority class, compensating for unbalanced classes. The performance metrics were calculated as the mean of 50 simulations (Fig. 2). Performance metrics were calculated for each model as the mean of 50 simulations. The performance metrics for the 12 antibiotics ranged from 75% to 98% for accuracy, 75% to 99% for precision, 71% to

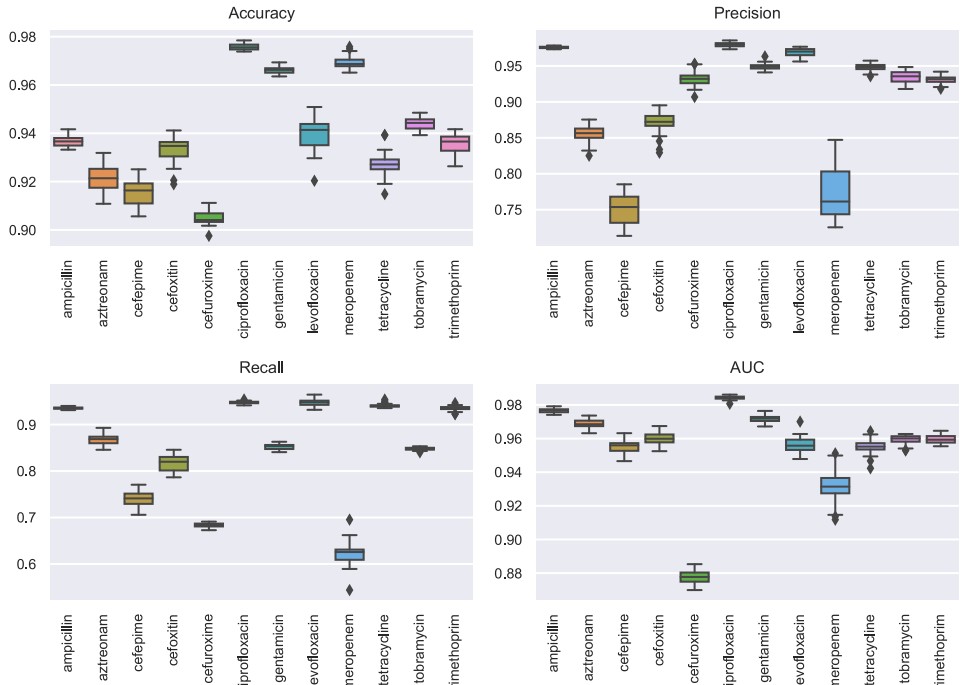

**FIG 1** k-mer-based supervised machine learning prediction of antibiotic resistance signature profiles to 12 antibiotics in the *E. coli* cohort. Boxplots showing the prediction performance results of the gradient boosting classifier for the 50 iterations. The performance indicators (y axis) are accuracy, precision, recall, and AUC. Predictive models were generated to classify the resistance versus susceptibility profiles of 12 different antibiotics (x axis).

98% for recall and 75% to 98% for AUC. The best predicted antibiotic was ciprofloxacin with a mean accuracy of 98% (±1%), precision of 99% (±1%), recall of 97% (±1%), and AUC of 98% (±1%). This suggests that SNPs in ciprofloxacin may have significant implications for the evolution of resistance, which is consistent with the study of K. Bhatnagar and A. Wong (50). The levofloxacin and meropenem models also achieved high performances, with an AUC of >95%.

The maximum importance in the 50 runs was captured for each SNP. To understand the relationship between AMR phenotype and genotype, we cross-referenced the SNPs that acted as predictors for AMR phenotype for each antibiotic to the pan-genome for each model data set and identified the corresponding genes. The identified SNPs, their corresponding genes, and the maximum importance scores obtained by the GBC are shown in Table S2. Importantly, the SNP-based approach could identify additional AMR genes that were not identified by the k-mer-based approach. By comparisons with the CARD and MutationDB databases, we identified 146 unique AMR genes associated with at least one antibiotic (Table 2) that were in the top 10% of features (ranked according to the maximum feature importance in the 50 runs). Out of these 146 genes, 8 had evidence in the database of the AMR gene for the specific antibiotic class (Table 2). Note, however, that the MutationDB database does not include entries for AMR genes for the levofloxacin and meropenem antibiotics.

**The AMR-related signatures occur in targets beyond annotated drug resistance genes and are associated with a wide range of metabolic systems.** To understand the systemic relationships connecting the identified AMR genetic signatures on a mechanistic level and to elucidate their mechanistic effects beyond genes encoding proteins targeted by drugs (i.e., positive selection in basal biosynthetic, regulation, and repair pathways), we integrated the genetic determinants with the GSM iML1515 (51) of *E. coli* K-12 MG1655. We limited our GSM analysis to the top 10% ranked genetic determinants identified, for each antibiotic classifier with an AUC of >95%, by the k-mer and SNP ML-based methods.

**TABLE 1** Known AMR genes identified by the k-mer-based AMR classifiers[a]

| Antibiotic | Drug class | Known AMR gene(s) to the antibiotic[b] | Known AMR genes associated with other antibiotics[b] |
|---|---|---|---|
| Ampicillin | Beta-lactam | TEM-1**, CTX-M-15, yicJ* | sul1**, folP**, APH(3'')-Ib, katE*, yadV*, arnC, fsr, nmpC, pepT, yeeJ, yhdJ |
| Aztreonam | Beta-lactam | CTX-M-55* | AAC(6')-Ib-cr, acrD, catIII, nmpC, pitA, yicI, cpdB, yoaE, rapA, dinG, yeeJ, oppA, arnC |
| Cefepime | Beta-lactam | CTX-M-1**, CTX-M-15, CTX-M-55 | dfrA25*, AAC(6')-Ib10*, AAC(3)-IId, catB3, AAC(6')-Ib-cr, folA*, yadV*, citF, yeeJ, ftsI |
| Cefoxitin | Beta-lactam | CMY-2*, ybiW*, betT, chiP, cra, envZ, htrE, lyxK, mdlA, yeeJ, yghA | dfrA25, AAC(3)-IId, catIII, blc, yaiY, folA, putA, lpoA |
| Ciprofloxacin | Fluoroquinolone | gyrA** | OXA-1*, CTX-M-15*, arnC, nmpC, htrE, cpdB, arcA, flu |
| Gentamicin | Aminoglycoside | AAC(3)-IId**, AAC(6')-Ib7**, aadA13*, AAC(3)-IIe*, AAC(6')-Ib9*, aadA7, ANT(2'')-Ia | floR, CTX-M-15, dfrA17, mphA, intS*, fliC*, arnC, yicJ |
| Levofloxacin | Fluoroquinolone | gyrA** | lacI*, yqiK, flu, arcA, fimC, phoE, ybiH, dadA |
| Tetracycline | Tetracycline | tet(A)**, tet(B)**, mdfA | APH(6)-Id, sul2, yeeJ, folP, csiD |
| Tobramycin | Aminoglycoside | AAC(3)-IId**, AAC(6')-Ib-cr**, AAC(3)-IIe, AAC(6')-Ib7 | catB3*, CTX-M-55, dfrA17, OXA-1, fliC*, pinR, ydfU, dnaQ |
| Trimethoprim | Diaminopyrimidine | ANT(2'')-Ia**, sul2*, aadA16*, aadA25*, APH(3'')-Ib*, TEM-1, tet(A), APH(6)-Id, mphA, TEM-150, sul1, folP*, dosP, valS, nmpC, htrE, groL, putP | |

[a]Genes in the top 10% features, ranked according to their maximum feature important assigned by the GBC classifier, are presented.
[b]Symbols: **, gene was associated with feature in the top 10% features; *, gene was associated with feature in the top 50% features.

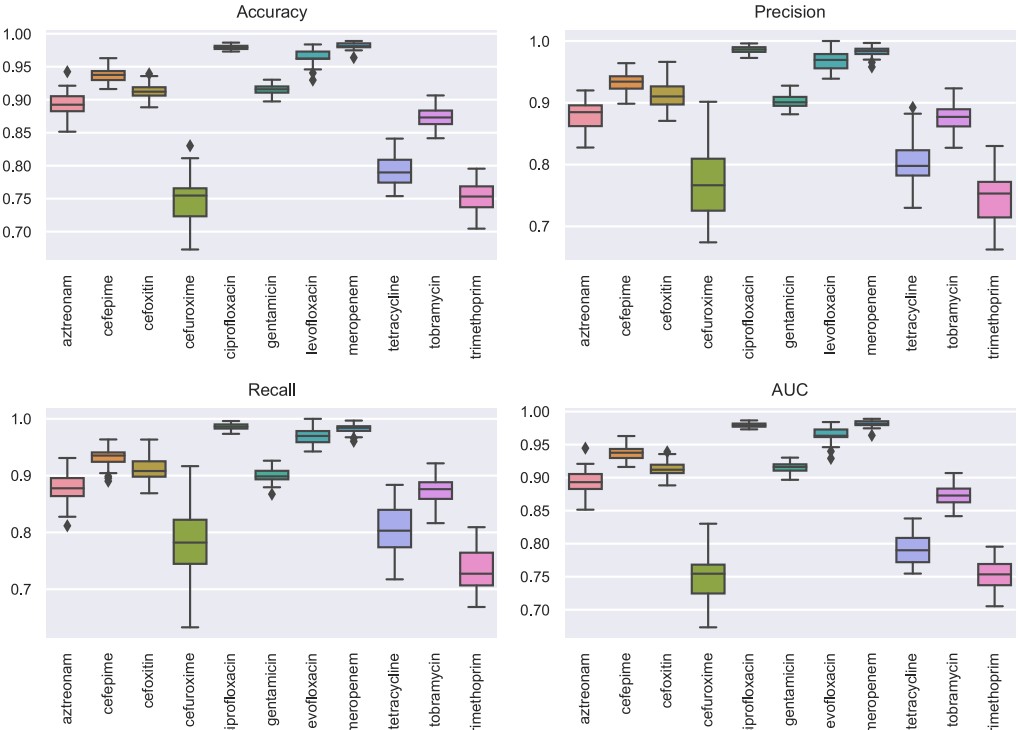

**FIG 2** SNP-based supervised machine learning prediction of antibiotic resistance signature profiles to 12 antibiotics in the *E. coli* cohort. Boxplots showing the prediction performance results of the gradient boosting classifier of the 50 iterations. The performance indicators (*y* axis) are accuracy, precision, recall, and AUC. Predictive models were generated to classify the resistance versus susceptibility profiles of 12 different antibiotics (*x* axis).

The ratio of metabolic genes to total genes corresponding to the top-ranked 10% of important features was considerably higher for the SNP-based models than the k-mer-based models (Fig. 3a and Fig. S3). The percentage of metabolic genes accounted for in iML1515 from the top features, for example, ranged from 43% (ciprofloxacin) to 48% (levofloxacin) in SNP-based AMR models. The percentage of metabolic genes identified by the k-mer-based approach and present in iML1515 were considerably lower, ranging from 5% (tobramycin) to 19% (levofloxacin). A large number of genes identified by the k-mer-based approach, however, were from the accessory genome, which currently lack many functional annotations, as shown in Fig. S3 (see the decrease in cyan bars to yellow bars). Additionally, since the GSM is based on the K-12 strain, accessory genes missing from this reference genome will not be included in the

**TABLE 2** Known AMR genes identified by the SNP-based AMR classifiers[a]

| Antibiotic | Known AMR genes to the antibiotic[b] | Known AMR genes associated with other antibiotics[b] |
|---|---|---|
| Ciprofloxacin | *gyrA**, parC**, parE*, typA*, hofN, valS, pnp, gyrB* | *speB**, yegU*, ugpB*, ampH*, fhuB*, poxB*, gss*, hybB*, phoE, speC, bglX, ftnA, pphA, yjfF, yjaB, yjjV, hofQ, yidC, prmB, hisF, plaP, truC, gcvP, mltC, rstB, mtlD, folA, metH, rnd, waaA, upp, putP, yohK, aidB, yegQ, uvrB, trmH, ulaG, yqjG, cpxA, proC, uvrA, recJ, hflX, tamB, cysK, metC, nrdB, mutM, mpl, osmF, mrcA, dcd, ravA, pepD, yejA, ribC, cstA, yeiQ, nusA, hemA, yaiZ, hybF, mglA, ysaA, potA, hemY, yjjP, recG, yebY, aroC* |
| Levofloxacin | | *parC**, gyrA**, hemF*, recG*, mysB*, metC*, tktA*, aceF*, yicR*, blgX*, fabD*, mutS*, chaA*, msyB*, rbsA*, gcvP, glnE, pcnB, mdtB, hisF, purT, menD, nikC, ftnA, frwB, yjiN, nadR, cyoB, fumC, mdtD, citG, glgX, valS, ldcC, yebQ, adiA* |
| Meropenem | | *parC**, gyrA**, creC**, yrfF**, valS**, bglX**, fucI*, hisF*, parE*, plaP*, nikA*, pykF*, aidB*, yjjG*, gcvP*, yjfF*, dsbD*, lepA*, thrA*, hybB, yccS, mdtB, murC, yegR, ravA, yjjV, yjjK, mscM, menD, mutS, metF, mglA, yjcD, nuoL, nadR, rplL, dusB, yegU, sufB, nudl, ulaG, ccmD, rnr, tamB, pdxA, dld, asd, ychO, soxR, yebK, nrdB, argD, baeS, glgX, osmF, trml, yegS, dnaX, yejH, waaC, fhuE, aroP, folA, ycbZ, rbbA, polA, recJ, speC* |

[a]Genes in the top 10% features, ranked according to their maximum contribution to the classifier, are presented.
[b]Symbols: **, gene was associated with feature in the top 10% features; *, gene was associated with feature in the top 50% features.

analysis. Nevertheless, a total of 289 genes present in iML1515 were identified by combining the genes that were associated with the top-ranked 10% of genes in the two machine learning models, which motivates the integration with GSM analysis.

The contribution of genes from each AMR classifier ranged between 1 (tobramycin) and 123 (ciprofloxacin), with a small number of genes overlapping between antibiotic AMR models (Fig. 3b and Table S3). These 289 important metabolic genes were associated with a wide range of metabolic systems (Fig. 3c), including transport metabolism, cofactor and prosthetic group metabolism, cell wall metabolism, alternative carbon metabolism, nucleotide metabolism, and amino acid metabolism were particularly prevalent across the diverse antibiotic classes (Fig. 3c and d). We performed gene pathway enrichment tests using the 40 metabolic subsystems included in iML1515 and also using the 352 gene-pathway annotation list downloaded from EcoCyc (52) (see Materials and Methods). The significant pathways with a false discovery rate (FDR) of less than 1% are provided in Table S3.

We found genes enriched in amino acid metabolism (histidine and arginine), the pyrimidine salvage pathway, putrescine biosynthesis pathway, and transport metabolism. Importantly, histidine metabolism has been found to play an important role in stress resistance in *E. coli* (53, 54), while putrescine, which is a polyamine, has been found to relieve the effects of oxidative stress in *E. coli* (55). Additionally, changes to genes involved in the pyrimidine salvage pathway have been found linked to the production of important biofilm components in *E. coli* (56), and therefore induce persistence (57). Furthermore, transport reactions are known to play a role in multidrug resistance by restricting the uptake of the antibiotic to reduce the toxicity (58, 59).

Next, using the GSM, we investigated the system-level effect of each important gene on metabolism, beyond the pathways they are encoded for. To this aim, we blocked the flux through reactions associated with an important gene (gene knockout) and evaluated the metabolic processes that were affected. In doing so, we can infer potential metabolic adaptation mechanisms that can be linked to a change in gene function (i.e., downregulation, overexpression, or deletion).

**GSM knockout analysis reveals genes related to growth limitation, auxotrophic behavior, and alternative carbon source utilization.** Next, to investigate further the metabolic processes involved in adaptation to antibiotic stresses, we considered the effects of the 289 genes on bacterial growth. The ability of bacteria to adjust their metabolism to slow down growth has for example been found to be advantageous for reducing the damage that occurs as a result of being the primary target of antibiotics (60–62). Identifying those that are essential for growth, while also being highly important in the ML models, may therefore provide a novel opportunity for selecting targets with dual mechanism.

To this aim, the GSM was used to simulate the behavior of *E. coli* with mutations in the 289 genes. Single gene deletions under rich environmental conditions were carried out in iML1515 to mimic the effect of a "loss of function" mutation on the entire system (see Materials and Methods). Importantly, we found a total of 20 gene knockouts that were lethal to the bacteria. These genes show a high level of agreement with *in vivo* gene essentiality results (63), as shown in Table 3. The lethal genes with the highest contribution (i.e., associated with the top 50 features) to the ML models, and therefore of greatest interest, included the following: *accA* and *metK* for ciprofloxacin, *fabD* and *fabG* for levofloxacin, *murG*, *lptG*, and *mraY* for meropenem, *folP* for ampicillin and trimethoprim, and *glmM* for gentamicin. These genes play essential roles in fatty acid elongation (*fabD*, *fabG*, and *accA*), peptidoglycan metabolism (*murG*, *mraY*, and *glmM*), lipolysaccharide biosynthesis (*lptG*), S-adenosyl-L-methionine metabolism (*metK*), and folate metabolism (*folP*) (Fig. 4). Importantly, *folP*, *lptG*, *fabG*, and *murG* are already known AMR-conferring genes, as shown by Tables 1 and 2.

Next, we considered genes that were growth limiting when the bacteria were grown on minimal medium with glucose as the carbon source. We found an additional 26 genes that were essential under these conditions (Table 3), which again showed high agreement with the *in vivo* results (64). Under poor nutrient conditions of the

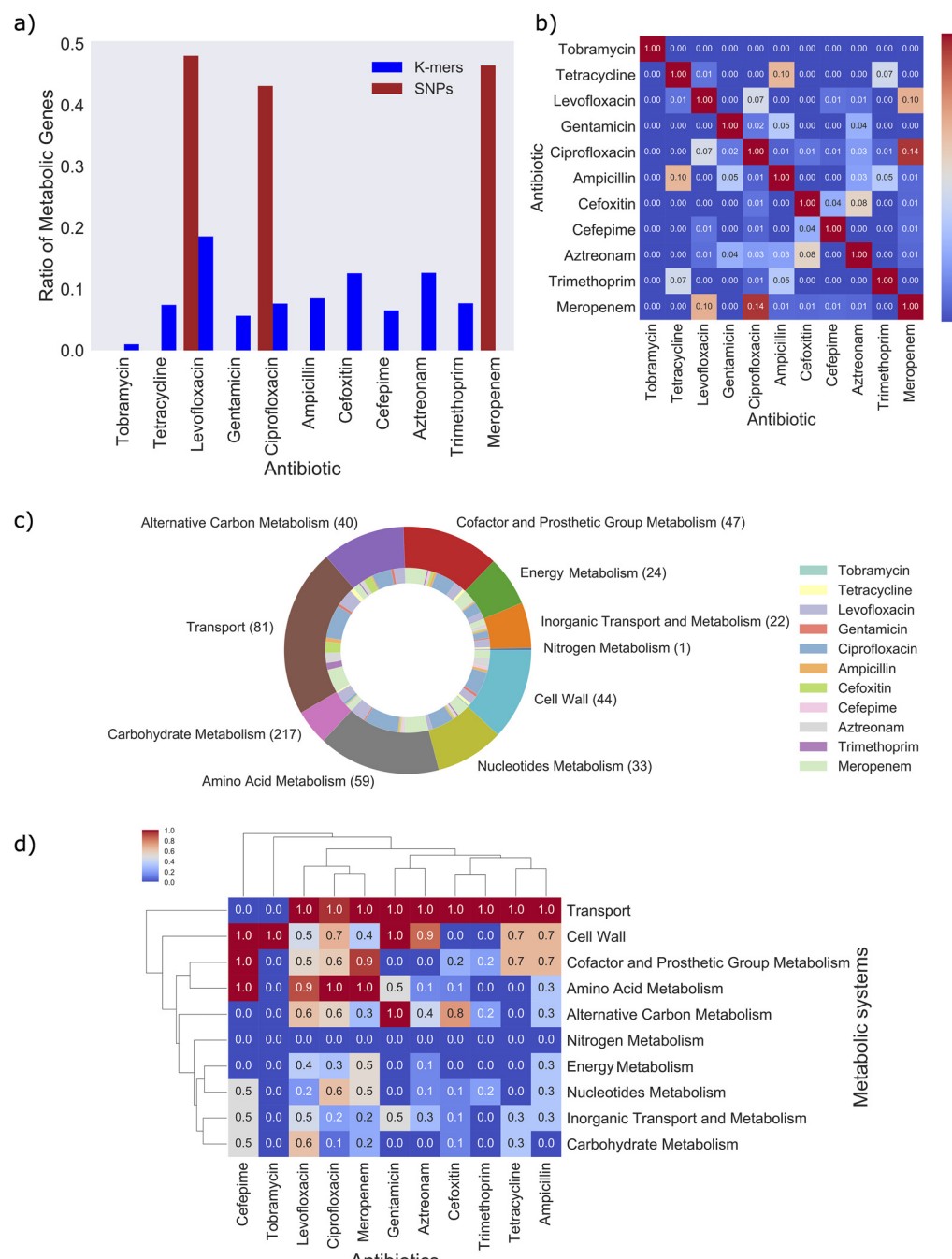

**FIG 3** Number of metabolic genes occurring in the 11 AMR classifiers. (a) Bar chart showing proportions of metabolic genes compared to the entire set of genes found in each AMR model. The blue bars represent gene proportions from the k-mer AMR models, whereas the red bars represent gene proportions from the SNP AMR models (AUC > 95%). (b) Heatmap showing the Jaccard index comparing the gene sets between two antibiotic classes. (c) Pie chart showing the proportions of genes associated with 10 metabolic systems (outer ring presented using the "tab10" color theme in Matplotlib). The inner ring shows the proportion of genes from each antibiotic class associated with each metabolic system and is presented using the "Set3" color theme in Matplotlib. Note that genes contributing to multiple antibiotic classifications will contribute multiple times in the pie chart, and therefore, the total area of the pie chart does not amount to 289. (d) Heatmap showing the normalized number of genes associated with each metabolic system. Note that the number of genes was normalized via column standardization. Hierarchical clustering was applied to both rows (metabolic systems) and columns (antibiotic classes) using the single linkage method and Euclidean distance as the metric. Each panel shows the results for the top 10% of genes identified in each AMR classifier. Panels b, c, and d show the results for the 289 genes found by combining the genes that correspond to the features in the top 10% of the k-mer and SNP classifications.

**TABLE 3** *In silico*-predicted gene lethality from the top-ranked discriminant genes in k-mer-based and SNP-based classifiers listed for each antibiotic

| Antibiotic | Essential genes (rich media)[a] | Essential genes (glucose minimal medium only)[a] |
|---|---|---|
| Ampicillin | folP* | |
| Aztreonam | | asd, purL |
| Cefepime | | pyrF |
| Cefoxitin | | |
| Ciprofloxacin | murJ*, lptG, hemG, ribC, accA*, **cysG, aroC**, waaA, hemA, metK*, lptF | purA, pheA, hisD, hisG, purL*, hisF, dapE, panD, purM, hisI*, ilvD, iscS, **thiD**, hisA, hisB*, hisH*, proC, purD |
| Levofloxacin | fabG*, fabD* | hisF*, purL*, **thiD*** |
| Gentamicin | glmM*, **cysG** | cysH |
| Meropenem | lptG*, mraY*, murG*, ispA | cysJ, hisD*, pdxA, hisC*, asd, hisF*, metF, murC, iscS, hisA, hisB, hisH*, hisG |
| Tetracyline | folP, murB | |
| Tobramycin | | iscS |
| Trimethoprim | folP*, ftsI | |

[a]Symbol: *, genes associated with top 50 ranked features of the antibiotic AMR model. Boldface genes have not been found essential in experimental studies.

host, changes in the function of these genes may contribute to slowing the growth rate, as before. However, if the environment is rich in nutrients, then a loss of function of these genes may have led to advantageous auxotrophic behavior. To test this hypothesis, we reran the knockout simulations for growth on glucose, while also allowing for individual metabolites to be utilized. Importantly, we found that 17 of these genes could be linked to auxotrophic behavior to amino acids, including cysteine (meropenem and gentamicin), histidine (levofloxacin, ciprofloxacin, and meropenem), phenylalanine (ciprofloxacin), and proline (ciprofloxacin) (Table 4). Auxotrophy for the vitamins thiamine (levofloxacin, tobramycin, and meropenem) and pantothenate (ciprofloxacin) was also found. Auxotrophy to peptidoglycan precursors was also found for the antibiotics ciprofloxacin and meropenem, while purine and pyrimidine precursors were found for ciprofloxacin and cefepime. Importantly, auxotrophy for histidine and thiamine has previously been found to elevate fitness (65).

Additionally, gene modifications that affect the utilization of alternative carbon sources was also investigated. Alternative carbon source utilization has been found advantageous for pathogenic survival of bacteria, including *E. coli*, *Salmonella*, *Vibrio cholerae*, and *Campylobacter jejuni* (66–68). To this aim, we used the GSM to test the effect of the 289 genes on the 297 different carbon sources in the iML1515 model. Single gene knockouts were repeated for each individual carbon source, under minimal medium conditions. We found 39 genes whose deletion blocked growth on a variety of alternative carbon sources (Table 5). The carbon sources that were blocked by the genes with the highest importance (i.e., associated with the top 50 features) in the ML models included the following: fucose (cefoxitin and meropenem), galactonate (cefoxitin), tartrate (levofloxacin), agmatine (ciprofloxacin), galacturonate (ciprofloxacin and levofloxacin), methyl-beta-D-glucuronate (cefoxitin), and a variety of nucleosides (ciprofloxacin).

**Flux balance analysis elucidates the effects of AMR-conferring genes on metabolite yields and reaction fluxes and suggests important metabolic adaptations in cell wall, energy metabolism, purine and pyrimidine metabolism and iron metabolism that increase antibiotic resistance.** Next, the GSM was used to investigate whether the genetic determinants could be linked to additional metabolic adaptation mechanisms, beyond those affecting the growth rate and alternative carbon utilization. For this analysis, we examined the effect of each gene on metabolite reproducibility and reaction fluxes. More specifically, we simulated single gene knockouts as before, however this time, we captured the effect on metabolite yields and flux spans (i.e., the variation of possible flux values for a given reaction) for all metabolites and reactions in the iML1515 model. The output of this analysis is twofold: (i) to identify clusters of genes that have similar metabolic phenotypes and (ii) to elucidate the metabolic adaptations that are most important in providing bacteria with possible resistance to antibiotic stress. Genes that confer a similar phenotype could give rise to higher variation of strains, while providing similar advantages for resistance (69).

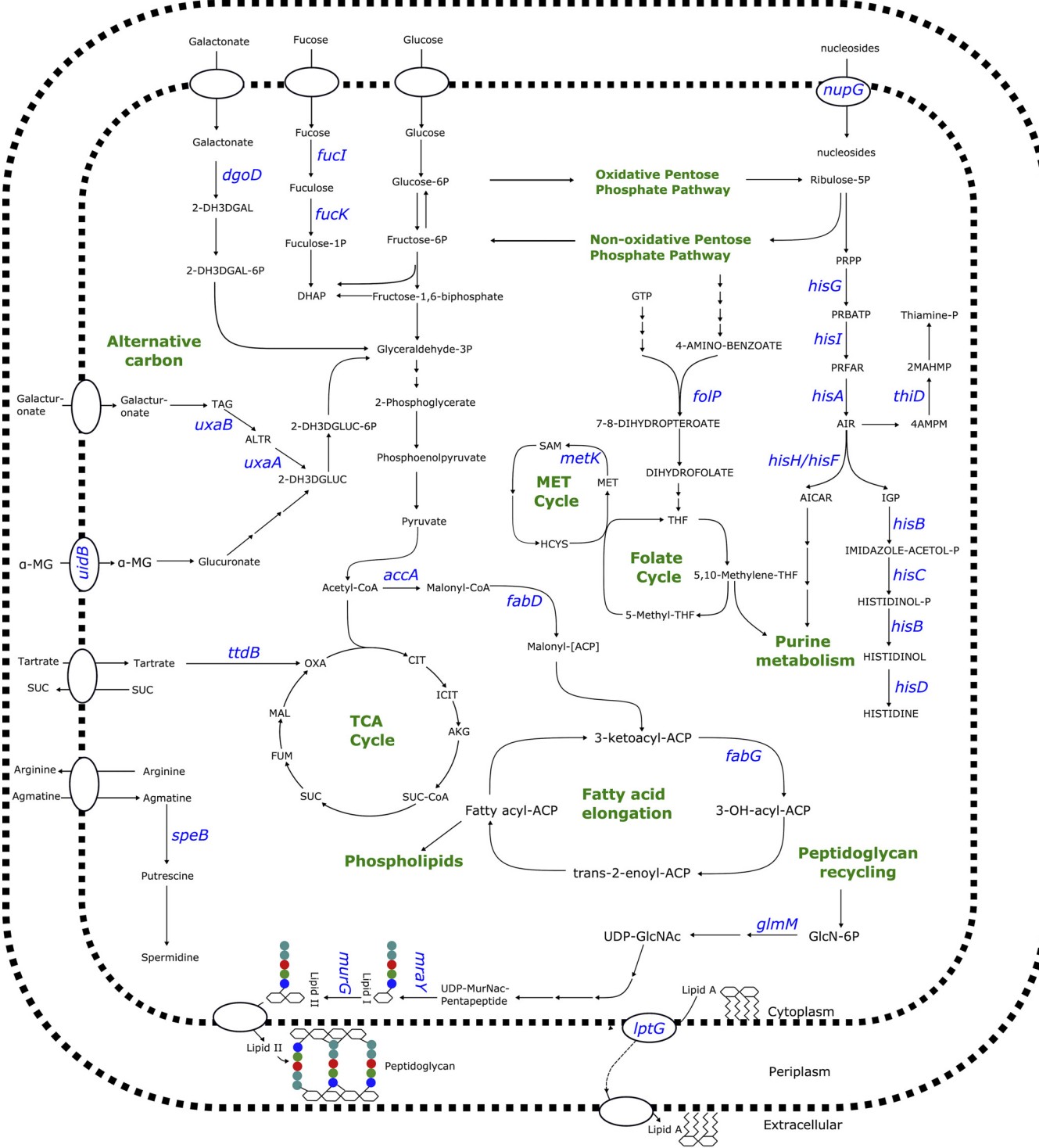

**FIG 4** An overview of the metabolic pathways involving potential gene targets for *E. coli*. The genes *accA*, *lptG*, *fabD*, *fabG*, *murG*, *mraY*, *folP*, *glmM*, and *metK* were all found to be essential in the GSM of *E. coli*, whereas knockout of the genes *hisA* and *thiD* all resulted in auxotrophic behavior. The genes *fucK*, *fucI*, *nupG*, *speB*, *uxaA*, *uxaB*, *dgoD*, *uidB*, and *ttdB* were all found to be essential to the growth on alternative carbon sources. Note that all genes presented corresponded to the top 50 features of the AMR models. Note that the antibiotic that each of these genes were found important to by the AMR models are provided. Abbreviations: 2-dehydro-3-deoxy-ᴅ-galactonate (2-DH3DGAL), 2-dehydro-3-deoxy-ᴅ-galactonate 6-phosphate (2-DH3DGAL-6P), fuculose 1-phosphate (fuculose-1P), dihydroxyacetone phosphate (DHAP), glyceraldehyde 3-phosphate (glyceraldehyde-3P), tagaturonate (TAG), altronate (ALTR), 2-dehydro-3-deoxy-ᴅ-galactonate 6-phosphate (2-DH3DGLUC-6P), 2-dehydro-3-deoxygluconate (2-DH3DGLUC), 1-*O*-methyl-beta-ᴅ-glucuronic acid (MG), oxalacetate (OXA), citrate (CIT), isocitrate (ICIT), alpha-ketoglutarate (AKG), succinyl-CoA (SUC-CoA), succinate (SUC), fumarate (FUM), malate (MAL), tetrahydrofolate (THF), glucose 6-phosphate (glucose-6P), fructose 6-phosphate (fructose-6P), guanosine-triphosphate (GTP), ribulose 5-phosphate (ribulose-5P), 5-phospho-alpha-ᴅ-ribose 1-diphosphate (PRPP), phosphoribosyl-ATP (PRBATP), phosphoribulosyl-formimino-5-

**TABLE 4** *In silico*-predicted gene knockouts that lead to auxotrophy from the top-ranked discriminant genes in k-mer-based and SNP-based classifiers listed for each antibiotic

| Antibiotic | Gene(s) leading to specific auxotrophy[a] |
|---|---|
| Ampicillin | |
| Aztreonam | |
| Cefepime | Pyrimidine compounds (*pyrF*) |
| Cefoxitin | |
| Ciprofloxacin | Phenylalanine (*pheA*), histidine (*hisA*, *hisB**, *hisD*, *hisF*, *hisG*, *hisI**, *hisH**), pantothenate (*panD*), thiamine (*iscS*, *thiD*), proline (*proC*), nucleosides (*purA*), peptidoglycan precursors (*dapE*) |
| Levofloxacin | Histidine (*hisF*), thiamine (*thiD**) |
| Gentamicin | Cysteine-derived compounds (*cysH*) |
| Meropenem | Histidine (*hisA*, *hisB*, *hisD**, *hisC**, *hisF**, *hisH**), S-methyl-L-methionine (*metF*), thiamine (*iscS*), pyridoxine (*pdxA*), cysteine (*cysJ*), peptidoglycan precursors (*murC*) |
| Tetracycline | |
| Tobramycin | Thiamine (*iscS*) |
| Trimethoprim | |

[a]Symbol: *, genes associated with the top 50 ranked features of the antibiotic AMR model.

Determining the most important metabolic adjustments that provide resistance to antibiotic stress may inform the development of novel treatments. The genetic determinants that have the largest system-level impact, i.e., an increase or decrease in their functionality (modeled here via gene knockouts) disrupts the largest number of metabolite yields and/or reaction fluxes, could provide promising new targets.

**Maximum theoretical yields of metabolites affected by AMR-conferring genes.** The lethality of each genetic determinant on all metabolites in the iML1515 model was determined using FBA. A gene knockout was considered lethal to the production of a specific metabolite if it results in blocking the biosynthesis of the metabolite (see also Materials and Methods). The results are represented as a bipartite graph of 98 genes and 508 metabolites. A gene is connected to a metabolite via an edge if its knockout results in preventing the metabolite's production. Using the Clauset-Newman-Moore greedy modularity maximization algorithm, we clustered the genes and metabolites into groups of similar phenotypes (Table S3). The largest six clusters are shown in Fig. 5. The metabolites within each cluster are involved in a variety of metabolic processes, including cell wall metabolism, nucleotide metabolism, transport metabolism, alternative carbon metabolism, amino acid metabolism, and cofactor and prosthetic group metabolism (Fig. 5b). To test which of these metabolic systems was significantly affected, we performed a pathway enrichment hypergeometric test on the metabolites in each cluster (see Materials and Methods). The most significant pathways associated with each cluster (FDR < 0.01) are shown in Fig. S4a and b.

A number of clusters could be linked to cell wall metabolites (Fig. S4 and S5). First, all 12 genes in cluster 5 affect the production of metabolites involved in lipopolysaccharide (LPS) metabolism. LPS are important compounds on the outer membrane and therefore have been found to play an important role in virulence (70, 71). Additionally, the genes *murC*, *ftsI*, *dapE*, *glmM*, *murB*, *murG*, *mraY*, and *mpl* in cluster 2 had a significant effect on the production of the metabolites involved in peptidoglycan (PG) metabolism. Peptidoglycan is a mesh-like structure that provides the strength and shape of the outer cell membrane, as well as providing protection against osmotic pressure. Modifications to PG can prevent the release of cell wall components, which initiate the host immune response (72), while also protecting the cell against antibiotic uptake (73). Similarly, changes to metabolites involved in fatty acid oxidation and phospholipids, specifically

**FIG 4** Legend (Continued)
aminoimidazole-4-carboxamide ribonucleotide phosphate (PRFAR), 5′- 5-aminoimidazole ribonucleotide (AIR), 4-amino-2-methyl-5-phosphomethylpyrimidine (4AMPM), 2-methyl-4-amino-5-hydroxymethylpyrimidine diphosphate (2MAHMP), thiamine phosphate (thiamine-P), phosphoribosylaminoimidazolecarboxamide formyltransferase(AICAR), D-erythro-imidazole-glycerol-phosphate (IGP), imidazole acetol-phosphate (IMIDAZOLE-ACETOL-P), histidinol-phosphatase (HISTIDINOL-P), glucosamine-6-phosphate (GlcN-6P), UDP N-acetylglucosamine (UDP-GlcNAc), UDP-N-acetylmuramyl-pentapeptide (UDP-MurNac-Pentapeptide), S-adenosyl-L-methionine (SAM), methionine (MET), homocysteine (HCYS).

**TABLE 5** *In silico*-predicted essential genes on specific carbon sources from the top-ranked discriminant genes in k-mer-based and SNP-based classifiers listed for each antibiotic

| Antibiotic | Lethal genes for growth on specific carbon sources important in AMR model*a* |
|---|---|
| Ampicillin | *gatC, mhpB* |
| Aztreonam | *adiC, yihP, cpdB, garD, mngB, paaK* |
| Cefepime | |
| Cefoxitin | *garD, kgtP, fucK*, ulaC, putA, fecA, mngB, uidB*, dgoD** |
| Ciprofloxacin | *malF, ulaG, nupG*, nanE, deoA, pepD, deoC, tonB, nanA, mtlD, xylA, uxaA*, putP, speB*, mngB, cpdB, lamB* |
| Levofloxacin | *adiC, ttdT, uxuB, uxuA*, ttdB** |
| Gentamicin | *hcaB* |
| Meropenem | *manZ, adiC, ulaG, exuT, fucI** |
| Tetracyline | |
| Tobramycin | |
| Trimethoprim | *putP, emrE* |

*aSymbol: \*, genes associated with top 50 ranked features of the antibiotic AMR model.

CDP-diaglycerol, whose production is affected by the six genes *purM*, *purD*, *ilvD*, *panD*, *purA*, and *purL* in cluster 1, may also provide protection against the immune response. The immune system, for instance, has been found to take advantage of the antimicrobial properties of long-chain fatty acids, which disrupt cell wall permeability when in excess in the extracellular environment (74). Pathogens have been found capable of modifying the biophysical properties of the cell membrane via changes to fatty acid structure, to increase the resistance to these antimicrobial peptides produced by the immune system (74).

In addition to cell wall metabolism, the genes in cluster 1 and cluster 4 are associated with a large number of pathways involved in purine and pyrimidine metabolism. Purine and pyrimidine metabolism is involved in the generation of DNA and RNA production; therefore, changes to the genes in this cluster may be important in repairing DNA from reactive oxygen species (ROS) (75). Importantly, metabolomics analysis showed purine metabolism pathways were highly enriched in multidrug-resistant *E. coli* strains (76). The genes involved in purine metabolism in cluster 1, *purL*, *purD*, *purM*, and *purA* in particular also have a downstream effect on many other metabolic pathways, including nitrogen metabolism, ppGpp metabolism, and allantoin biosynthesis, all of which can be linked to the regulation of the stringent response (77–79). Importantly, changes in ppGpp concentration play an important role in controlling cellular growth, and depletion of this metabolite has been found to trigger a dormant cell metabolic state, promoting antibiotic-tolerant persistence cells (80, 81). *E. coli* cells starved of nitrogen have been found to have increased ppGpp, which again has been found to induce tolerance to ciprofloxacin (78). Allantoin degradation has been found as an important adaptive response to recovery after nitrogen starvation (77). Furthermore, these genes, as well as the *folP* gene, also affect metabolites involved in folate metabolism, tetrahydrofolate (THF) biosynthesis in particular. Importantly, point mutations in *folP* have been identified to prevent sulfonamides from inhibiting THF production (82). We identified *folP* in the trimethoprim, tetracycline, and ampicillin ML models. Folate metabolism, including THF, however, are again important for nucleotide biosynthesis and have in fact been found important for persistence in *E. coli* cells exposed to ampicillin (83). The production of coenzyme A (CoA) is also affected by these genes, as well as the *ilvD* and *panD* genes. CoA is an important cofactor in many metabolic processes, including fatty acid biosynthesis, which are used in LPS, and the tricarboxylic acid (TCA) cycle. The concentration of acetyl-CoA, an important derivative of CoA, has also been found to play a key role in assessing the cell metabolic state, which, in turn, determines the fate of either cell growth, survival, or death (84).

The production of metabolites relating to iron metabolism were affected by genes in both clusters 2 and 6. The four genes in cluster 6 affect metabolites involved in heme biosynthesis. The capability (or improved capability) for heme synthesis may provide pathogens with a competitive advantage for colonization, since heme is the

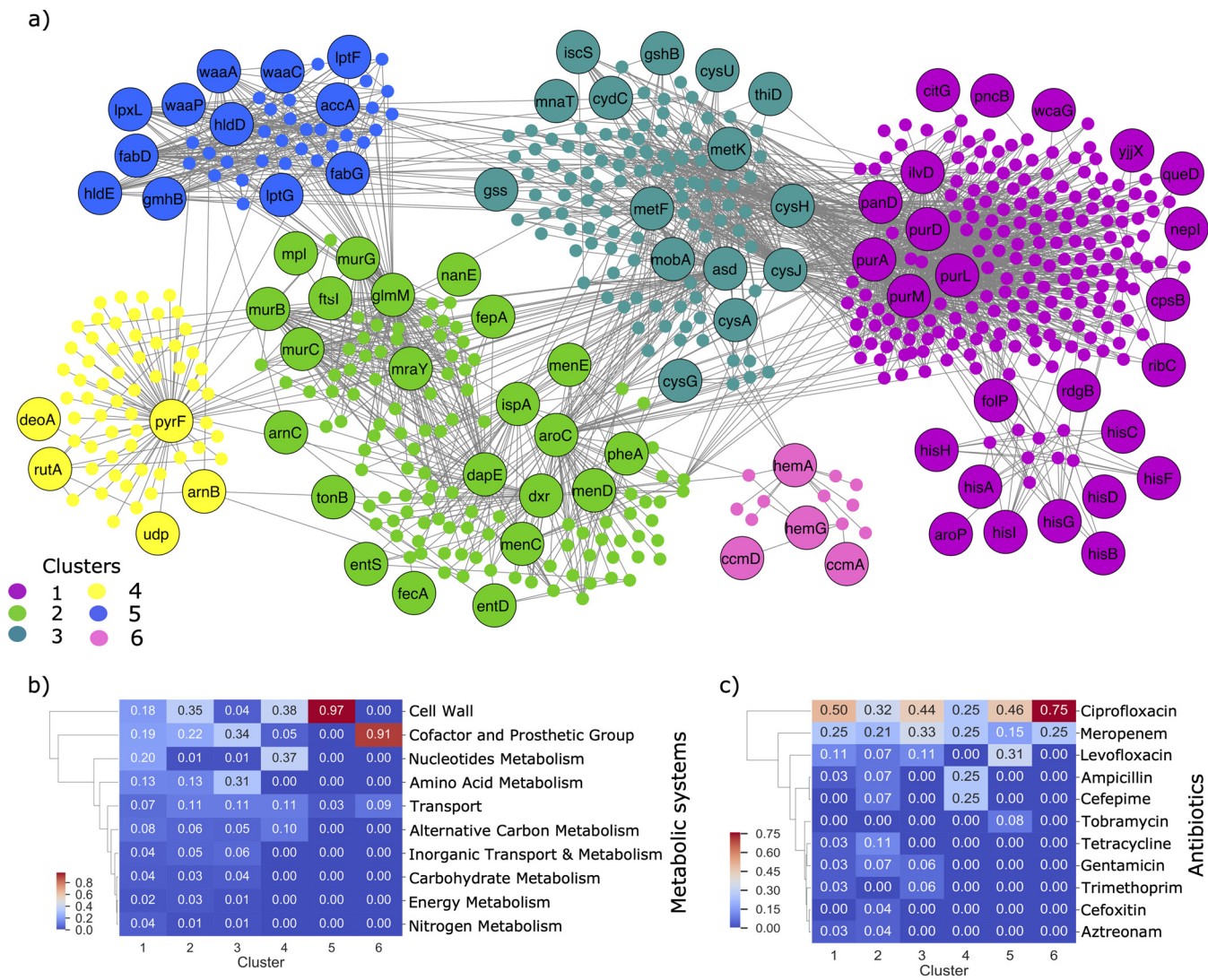

**FIG 5** Effects of genetic determinants on metabolite yields. (a) Bipartite network with genes and metabolites as nodes. Labeled nodes represent genes, whereas unlabeled nodes represent metabolites. A gene and metabolite are connected by an edge if the deletion of the gene blocks the metabolite production. Genes and metabolites are highlighted according to the cluster they were assigned to via the Networkx modularity algorithm. The number of clusters in the figure was reduced by considering only those of size greater than 10. (b) Heatmap showing the metabolic systems associated with each of the six clusters. A gene was associated with a metabolic system, if at least one metabolite correlated with the system could no longer be produced after the gene was deleted. (c) Heatmap showing the antibiotics associated with each cluster. Note that genes occurring in multiple antibiotics were accounted for twice. Hierarchical clustering was applied to the rows of each heatmap (metabolic systems or antibiotic class) using the single linkage method and Euclidean distance as the metric. The gene counts have been normalized by the total number of genes in each cluster in each heatmap.

largest source of iron for the cell. Excess heme, however, increases the level of ROS and therefore is extremely toxic to the cell, so the regulation of heme concentration is essential (85). The genes *menD*, *entS*, *aroC*, *dxr*, *pheA*, *menC*, *menE*, *ispA*, and *entD* in cluster 2 all affect enterobactin biosynthesis, either directly or via the chorismate biosynthesis pathway. Enterobactin is an iron-scavenging siderophore and has been found important for pathogen virulence (86–88). An important response of the immune system is to use nutrient immunity by limiting iron availability, which has an important function in energy metabolism and DNA replication (89, 90). Changes to genes affecting iron metabolism may therefore enhance the resistance by improving their ability to scavenge iron from the environment. Additionally, however, genes *menD*, *dxr*, *aroC*, *menE*, *ispA*, and *menC* also affect metabolites involved in the electron transport chain (ETC). Importantly, previous work has found reduced respiration via the ETC resulted in mutant strains highly resistant against ampicillin and gentamicin (91). The ETC reduces the proton

mSystems®

motive force that is necessary for gentamicin uptake. Reduced flux through ETC, however, also reduces the growth rate, which as previously discussed enables multidrug level persistence (60–62).

The genes in cluster 3 are also affecting metabolites involved in electron carrier metabolism. The eight genes *cysJ*, *metK*, *asd*, *gss*, *cysH*, *cydC*, *metF*, and *gshB* for instance are all affecting metabolites involved in glutathionylspermidine (GSP) biosynthesis. Importantly, GSP can be recycled back to glutathione and spermidine. Glutathionine is an important antioxidant metabolite required for detoxifying ROS (92, 93), while spermidine is a polyamine also found to provide protection against ROS exposure (94). A subset of these genes, *cysJ*, *metK*, *asd*, *cysH*, and *metF*, are also affecting biotin production. Importantly, biotin has been identified as important for the virulence of enteropathogenic *E. coli* (EPEC) strains, due to its involvement in the regulation of the locus of enterocyte enfacement (LEE). The LEE system is essential to these pathogenic bacteria in order to attach and infect host epithelium cells (95). Increased biotin concentrations have been shown to limit enterohemorrhagic *E. coli* (EHEC) infections in mice (96).

In general, the metabolic processes described here are affected by genes identified in the ML models for diverse antibiotic classes (Fig. 5c and Fig. S4c). This is not too surprising, however, since these processes are suggested to increase antibiotic resistance via protection from the immune response, oxidative stress, and/or the stringent response, which are multidrug adaptation mechanisms for enhancing fitness, persistence, and/or virulence (97–99).

**Flux variability analysis identifies the biochemical reactions whose flux span was affected by AMR-conferring genes.** Next, we investigated the system-level effect of the AMR-conferring genes on metabolic fluxes. Specifically, flux variability analysis (FVA) was used to identify the biochemical reactions whose flux span was affected by mutations in the genetic determinants. The results are represented as a bipartite graph of 145 genes and 861 affected reactions (Table S3). A gene is connected to a reaction via an edge if its knockout results in reduced flux span through the reaction. As before, the Clauset-Newman-Moore greedy modularity maximization algorithm was used to cluster the genes and reactions into groups of similar phenotypes (Table S3). The largest nine clusters are shown in Fig. 6a, i.e., those with greater than 10 nodes (genes and metabolites). A variety of metabolic processes were enriched in the clusters (Fig. 6b), similar to the gene-metabolite clusters. To test which of these metabolic systems was significantly being affected, we performed a pathway enrichment hypergeometric test on the reactions in each cluster. The most significant pathways associated with each cluster (FDR < 0.01) are shown in Fig. S6a and b.

The gene-reaction network was clustered into similar groups of genes to the gene-metabolite network. Again, the clusters were enriched with metabolic processes, including cell wall metabolism (LPS, PG, fatty acids, and phospholipids), nucleotide metabolism (purine, pyrimidine, and folate metabolism), amino acid metabolism (histidine and methionine), and iron metabolism (heme). The main differences between networks involve the set of genes in cluster 1. Unlike before, this analysis reveals the genes *gmhB*, *waaC*, *waaP*, *accA*, *lptG*, *lptF*, *waaA*, *hldD*, *fabG*, *lpxL*, *hldE*, *fabD*, and *glmM* are affecting the biosynthesis of nucleotide sugars. These sugars are incorporated into the O-antigen region of LPS, which is located in immunodominant part of LPS (100). Furthermore, the genes *accA*, *nuoL*, *nuoN*, *fabG*, *tesA*, and *fabD* are affecting fatty acid biosynthesis, as well as biotin biosynthesis. As discussed previously, both fatty acids and biotin metabolites can affect the host immune response and bacterial virulence.

Furthermore, the FVA analysis also revealed that the genes in cluster 7 all affect iron transport, which, as previously discussed, may be important for scavenging iron from the host. Additionally, disruptions to the genes in cluster 2, specifically *asd*, *gcvP*, *gcvT*, *serA*, *metF*, *cysH*, *cysJ*, and *serB*, were found to affect amino acid metabolism (cysteine, serine, glycine, aspartate and/or methionine), all of which are involved in folate transformation of *E. coli*. As previously discussed, folate metabolism can affect persistence to antibiotic exposure. Alternatively, however, sulfur amino acid residues in proteins,

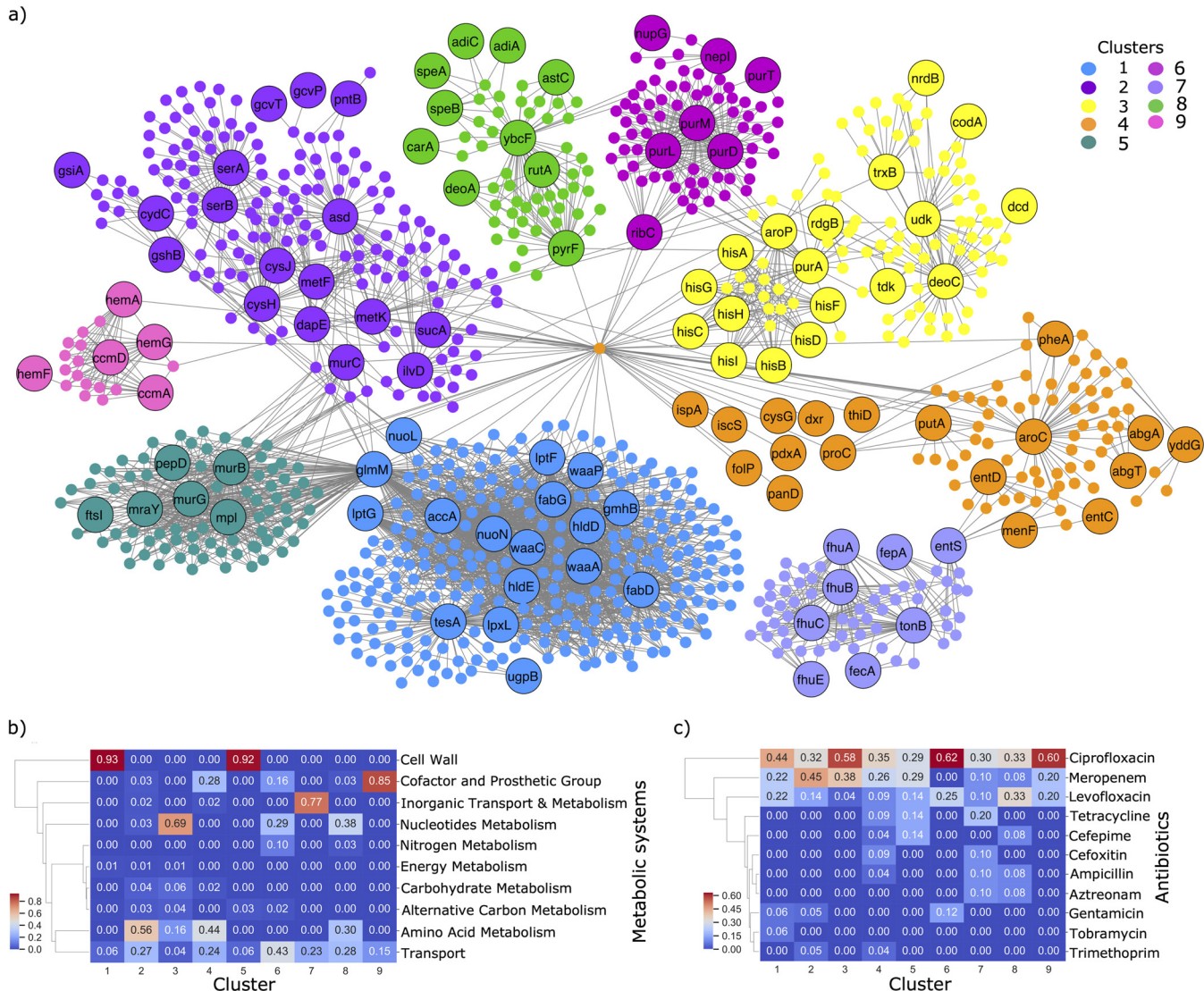

FIG 6 Effect of genetic determinants on reaction fluxes. (a) Bipartite network with genes and reactions as nodes. Labeled nodes represent the genes, whereas unlabeled nodes represent reactions. A gene and reaction are connected by an edge if the deletion of the gene reduces the reaction flux by at least 10%. Genes and reactions are highlighted according to the cluster they were assigned to via the Networkx modularity algorithm. Note that to reduce the initial size of the network, we only included clusters of size greater than 10. (b) Heatmap showing the metabolic systems associated with each of the nine clusters. A gene was associated with a metabolic system, if the flux span of at least one reaction correlated with the system was reduced after the gene was deleted. (c) Heatmap showing the antibiotics associated with each cluster. Genes occurring in multiple antibiotics were accounted for twice. Hierarchical clustering was applied to the rows of each heatmap (metabolic systems or antibiotic class) using the single linkage method and Euclidean distance as the metric. The gene counts have also been normalized by the total number of genes in each cluster in each heatmap.

including methionine and cysteine, are found to be extremely reactive with ROS; therefore, changes to the genes specifically affecting these amino acids may play a role in ROS detoxification (101).

Again, these metabolic pathways could be associated with a diverse set of antibiotic classes (Fig. 5c and Fig. S5c), suggesting the changes in these genes are linked to secondary multidrug adaptation mechanisms.

## DISCUSSION

Machine learning provides powerful and robust means for predicting AMR phenotypes and their genetic determinants. ML methods have proven successful in identifying known AMR mechanisms (23–29). Interpreting ML models, however, remains a challenge due to their complexity and large number of contributing features. Current approaches mostly consider only the genes with known AMR associations in AMR

databases and neglect genetic determinants relating to important metabolic phenotypes, which are known to play a role in antibiotic resistance. Kavvas et al. (30) have recently developed the first computational pipeline that combines machine learning with genome-scale metabolic models to enable biochemical interpretation of genetic determinants. In their pipeline, the effect of alleles on the flux solution space was used to successfully classify AMR phenotypes of *Mycobacterium tuberculosis* strains. In our work, we take an alternative two-step approach. First, a combination of a k-mer- and SNP-based machine learning approach is used to identify genetic determinants. Second, a genome-scale metabolic model is used to assess the effect of genetic determinants on metabolite producibility and biochemical fluxes to elucidate possible metabolic adaption mechanisms. Our approach produced AMR models of *E. coli* that achieved performance accuracies competitive with the current approaches. Moreover, we were able to reveal novel biomarkers based on the systemic effect the genetic determinants have on growth, metabolite yields, and metabolic fluxes.

The competitiveness of our ML approaches is that two methods were applied in parallel: a k-mer-based approach and a SNP-based approach, and only the genes identified in either the top 10% of the k-mer or SNP classifiers with an AUC of >95% were used for the GSM. Notably, the k-mer-based approach outperformed the SNP-based approach for 8 of the 11 antibiotics (AUC > 0.95), specifically, 8% higher for aztreonam, 17% higher for ampicillin, 2% higher for cefepime, 5% higher for cefoxitin, 6% higher for gentamicin, 28% higher for trimethoprim, 10% higher for tobramycin, and 21% higher for tetracycline. A possible reason for this is due to the inclusion or exclusion of accessory genes in the two approaches. That is, the k-mer-based approach allows for discriminating between resistance and susceptibility according to both the core and accessory genome, whereas the SNP-based approach is restricted to the core genome only. The antibiotics that performed well only via the k-mer-based approach may therefore be highly dependent on acquired resistance genes, such as the highly discriminant beta-lactamases. The SNP-based approach, however, successfully predicted AMR resistance for the two fluoroquinolone antibiotics ciprofloxacin and levofloxacin, and for the beta-lactam antibiotic meropenem. Importantly, the SNP-based approach performed extremely well (AUC > 0.98) for ciprofloxacin, suggesting that the antibiotic induces mutations, which is consistent with the literature (50). Importantly, the k-mer- and SNP-based approaches identified different known AMR genes, validating the advantage of combining the important features from both approaches. The combined approaches identified 225 known AMR genes corresponding to the top 10% of ranked features recognized as discriminant by the AMR classifiers. Out of these 225 genes, 35 matched the specific antibiotic class that has been reported in the databases.

Importantly, a number of the genes identified by both the k-mer- and SNP-based models were associated with metabolic reactions. Using the GSM iML1515, we found a total of 289 metabolic genes from the top 10% of features from both the k-mer- and SNP-based models. The number of metabolic genes from the SNP-based models was considerably higher than the number of metabolic genes from the k-mer-based models. This is not too surprising, however, since the k-mer-based approach included the important accessory genes responsible for drug target modifications, drug efflux, and enzymatic inhibition. Metabolic-gene-specific mutations provide a secondary adaptation mechanism to reduce the antibiotic efficacy. Importantly, previous studies have also found that metabolic-gene-specific mutations are present in the core genes of *E. coli* (102).

The 289 total metabolic genes were significantly enriched in various metabolic pathways, including transport metabolism, nucleotide metabolism, and amino acid metabolism. To understand the mechanistic effects of these 289 genes, we used flux balance analysis to predict the system-level metabolic changes that result from genetic variants of the genes (i.e., mutations or absence). More specifically, we predicted metabolic phenotypes of genetic variants via gene knockouts and identified the metabolic processes that were being affected. Importantly, using our new ML-FBA integrated approach, we could reveal interesting links between genes and potential metabolic

adaptation mechanisms that, importantly, were not identified using standard gene pathway enrichment analysis.

Using the GSM, we found 20 genes essential for growth under rich environmental conditions. The essential genes with the highest importance in the ML models may be promising targets for generating dual-mechanism antibiotics. That is, the antibiotic targets pathways that would lead to inhibition of an essential metabolic process, while simultaneously reducing the ability of the pathogen to adapt. The most promising new candidates as targets included the following: accA and metK for ciprofloxacin; fabD and fabG for levofloxacin; murG, lptG, and mraY for meropenem; folP for ampicillin; and glmM for gentamicin. Modifications to these genes may result in slower growth, which has previously been found advantageous to pathogenic bacteria, including E. coli and Salmonella, for reducing the damage that occurs as a result of being the primary target of antibiotics (60–62, 103). Alternatively, however, the genes accA, fabG, fabD, lptG, murG, and mraY affect biosynthesis of cell wall components and therefore may have had an effect on membrane properties for antibiotic uptake or manipulation of the host's immune response (73, 74). The gene folP, which is involved in folate metabolism, has previously been identified to prevent sulfonamide drugs from inhibiting folate metabolism (82). Importantly, however, we identified folP in the trimethoprim, tetracycline, and ampicillin ML models. Folate metabolism, including tetrahydrofolate (THF), however, are again important for nucleotide biosynthesis and have in fact been found important for persistence in E. coli cells exposed to ampicillin (83). Importantly, a number of additional genes affecting folate metabolism were also identified in the metabolite reproducibility analysis and flux variability analysis.

Interestingly, we also found a number of gene knockouts which resulted in auxotrophic behavior to a number of amino acids, including histidine, cysteine, phenylalanine, and proline, as well as auxotrophy to the vitamins thiamine and pantothenate. The production of these metabolites are particularly energy intensive, and therefore, their acquisition from the host may provide pathogens with a competitive fitness advantage against commensal bacteria (104). Alternatively, auxotrophy may have developed due to the critical role the metabolite plays in host-pathogen interactions. Using these genes as new drug targets has the disadvantage that the pathogen may be able to utilize exogeneous nutrients from the host environment.

Additionally, we identified 39 genes whose knockout affected the growth of E. coli on alternative carbon sources. The genetic determinants with the highest importance in the ML models affected growth on various carbohydrates. Interestingly, a previous study found that various carbohydrates, including fucose, promote natural transformation of E. coli, therefore potentially contributing to the acquisition of antibiotic resistance and virulence (105). Fucose is particularly interesting as it has also been found to positively regulate microbiome bacterial colonization and host immune activation (106).

Furthermore, clustering of genes according to metabolic phenotypes also revealed a strong link to cell wall metabolism adaptations. Genes were found to affect phospholipids, lipolysaccharides, fatty acid, and peptidoglycan metabolism, all of which can be associated with increased antibiotic tolerance via increased permeability of the membrane, as well as playing a role in virulence by manipulating the host immune response (73). Pathogens have been found to modify the cell wall components, for instance, that are usually recognized by the host's innate immune response (74, 107). Changes to a number of genes that were affecting cofactor biosynthesis may also be involved in immune response manipulation, including the biosynthesis of biotin and iron. Increased biotin concentrations, for example, have been found to reduce the ability of EHEC to attach and infect host epithelium cells (95, 96). Furthermore, genes affecting both enterobactin metabolism and heme metabolism were also found, both of which may improve resistance to nutrient immunity by increasing the pathogen's ability to scavenge iron from the environment (108, 109). Iron is important for many enzymes in bacteria, particularly those involved in oxidative phosphorylation and DNA synthesis; therefore, it is essential for the bacterium's survival.

Purine and pyrimidine metabolism was also enriched in the gene clusters. Modifications to these genes may limit the inhibitory effect of antibiotics that target DNA replication, such as ciprofloxacin and levofloxacin. Importantly, however, the genes encoding purine and pyrimidine biosynthesis enzymes have a large system-level effect involving many different metabolic processes. The genes *purL*, *purD*, *purA*, and *purM* in particular affect the production of DNA building blocks, which may be important for DNA repair against antibiotic-induced ROS (75). Furthermore, these genes also affect ppGpp metabolism, which is important for regulating cellular growth and inducing antibiotic-induced persistence (80, 81). Additionally, these genes also affect the production of important cofactors of energy metabolism, such as ATP, NAD, and NADPH, which are important for the electron transport chain (ETC). Other ETC metabolites, including ubiquinone, menaquinone, and flavin, were also being affected by the important genetic determinants. Changes in the flux through ETC may contribute to antibiotic resistance in a number of ways. Reduced ETC reduces the proton motive force (PMF) required for aminoglycoside uptake (110), while also reducing the growth rate for increased persistence (60–62). Furthermore, the ETC reactions are also responsible for ROS production. A related study that applied gene knockout simulations on an extended GSM of *E. coli*, which included specific ROS-producing reactions, identified genes associated with the ETC as ROS-inducing targets for improved antibiotic killing (111). Further evidence to suggest adaptation to ROS was found by a number of additional genes, whose knockout was found to affect glutathionine, spermidine, methionine, or cysteine biosynthesis. Importantly, these metabolites have all previously been found to provide protection of *E. coli* cells by acting as antioxidants (92, 93, 101).

Importantly, the genetic determinants associated with the metabolic adaptation mechanisms described here were identified in the ML models for diverse antibiotic classes. Changes in these genes are therefore suggested to be contributing to secondary resistance mechanisms via a generic response against toxicity and stress, but it is nonetheless essential for their survival (97–99).

In summary, we have demonstrated that our new approach is capable of identifying several metabolic adaptation mechanisms, including reduced reactive oxidative stress toxicity, reduced proton motive force, increased colonization via utilization of alternative nutrients, increased persistence via reduced growth and host immunity defense mechanisms. These metabolic adjustments occur downstream of the initial drug target inhibition and are suggested here to play a role in antibiotic resistance. Targeting the most important genetic determinants with the highest effect on these secondary adaptation mechanisms while simultaneously targeting essential metabolic processes, however, may provide novel new treatments that increase antibiotic efficacy (112).

Our new approach can be applied to study genetic determinants of any pathogen of interest, providing a large cohort of AMR phenotypes are available and a genome-scale metabolic model exists for a reference genome. The second step of our approach depends only on the GSM, and therefore, precomputing the metabolic changes (e.g., effects on metabolite yields or metabolic fluxes) for the entire set of genes in the model is possible, which could be readily available for future AMR studies to draw insights on potential new AMR genes. Future efforts may precompute all of these deeper metabolic effects for each gene in a given GSM(s). Such future endeavors will offer the possibility to future AMR genome-wide association studies (GWAS) to readily draw insights from potential AMR gene metabolic effects as predicted by these methods without needing to set-up and solve all the GSM problems independently.

While this information is useful, this new approach has many other areas of future development that could lead to deeper understanding of the metabolic changes that facilitate antibiotic resistance. The k-mer-based AMR models, for example, included the primary mechanisms of resistance, and while these strong genetic determinants provide highly accurate models and a means of validation, other resistance mechanisms may be diluted or "washed out" (25). Additionally, many of the genes corresponding to the important k-mers in the AMR models had unknown functional annotation,

therefore limiting the power of integration with the GSM. Furthermore, our approach was limited to protein-coding genes only, and therefore lacks the ability to identify important non-protein-coding regions, which have previously been found to confer resistance, such as *eis* and *rrs* (113). Likewise, we are not considering synonymous changes in the protein that also have been related to resistance (114). However, as also pointed out by Kavvas et al. (29), these types of computational platforms are open to account for non-protein-coding genes and synonymous SNPs in future work. Additionally, using more advanced GSM frameworks, such as regulatory FBA (115) and GEM-PRO (116), for example, would allow us to investigate the effects of genetic determinants on metabolic phenotypes via changes to gene regulation and protein structure. The characterization of the AMR-associated SNPs, in respect to a reference genome such as *E. coli* K-12 MG1655, would allow us to link the specific amino acid substitutions or deletions to antibiotic resistance. One-dimensional (1D)−three-dimensional (3D) structure-function prediction analysis may then enable us to determine whether the SNPs result in a loss or gain of function, which is directly integrated as constraints into models such as GEM-PRO. The effects of the SNPs on the genes (i.e., loss of function or gain of function) is not determined in our approach and if considered would allow further insights into the biological interpretation. Finally, our approach was applied to a GSM model of the *E. coli* K-12 MG1655 strain, and therefore was limited to the genes in this genome. Developing a GSM of the pan-genome of the 3,616 strains used may reveal additional metabolic genes that are important for resistance. Furthermore, an extended version of iML1515 has been developed that includes ROS-specific reactions (111). Applying the approach developed here to this model would therefore be useful future work for exploring the most important genetic determinants for improving antibiotic efficacy via ROS-associated cell death (111).

Taken together, our new pipeline was able to determine known AMR genes and suggest new ones that may weaken the pathogen's resistance to antibiotics. Continued improvement to the approach by increased availability of AMR phenotype data, the further enhancement of ML tools, further development of GSM representations of pathogenic bacteria, and improved functional annotation of genes will provide a means to confidently predict the metabolic responses that facilitate AMR resistance.

## MATERIALS AND METHODS

**Data collection and antimicrobial susceptibility phenotypes.** Resistance phenotypes and isolation country data for *E. coli* genomes were downloaded from the PATRIC database (https://www.patricbrc.org/). We selected genomes that were annotated as either "susceptible" or "resistant" to a single antibiotic. These AMR phenotypes were derived from laboratory analyses only and included a mixture of both Clinical and Laboratory Standard Institute (CLSI) and European Committee on Antimicrobial Susceptibility Testing (EUCAST) AMR standards. The list of the laboratory method standards used to determine the AMR phenotypes is detailed for each isolate in the supplemental Excel file "Ecoli_genomes_metadata.xlsx," which is provided on https://github.com/tan0101/GSM_mSystems_2021. All the genome sequences of the isolates that were used in this study were listed in PATRIC as "good" quality assemblies. Isolates labeled "good" quality in PATRIC meet the criteria set by Parrello et al. (117) that contamination is less than 10%, fine consistency is greater or equal to 87%, and the completeness of the sequence is greater or equal to 80%. We have also included only isolates that were labeled "WGS" (for whole genome sequenced) or "Complete" in the genome status in PATRIC, which removes any cases that are "plasmid-only." Finally, we also included an additional filtering that removed any isolates with a contig number greater than 250, as previously done by Hyun et al. (25). The 12 antibiotics chosen to be studied had at least 200 genomes annotated as "susceptible" or "resistant." These antibiotics were ampicillin, aztreonam, cefepime, cefoxitin, cefuroxime, ciprofloxacin, gentamicin, levofloxacin, meropenem, tetracycline, tobramycin, and trimethoprim. These antibiotics encompass a range of classes, including beta-lactams, aminoglycosides, and carbapenems, as well as multiple generations of antibiotics.

**Genome assembly and annotation, *in silico* subtyping identification, pangenome construction, and core genome alignment.** The genomes of all selected isolates were annotated with Prokka v1.13 (118) using default parameters. All annotated files by the antibiotic model were taken as input for pangenome analysis with core gene alignments through Roary v3.13.0 (47). SNP sites 2.5.12 was then used for extracting the core gene variant sites from the core gene multiple alignment obtained for each different antibiotic. The variants were used as features for the machine learning classifiers (119). Each core gene nucleotide sequence was further aligned, and single nucleotide variants were identified. The position of a SNP in a gene was selected as a feature in the machine learning if the nucleotide varied in

more than 5% of strains (i.e., was constant in less than 95% of strains). Such variation could be determined only if including genes present in 100% of isolates (core genome) within the study population and by aligning the core genome (the genes present in all isolates). This is why we considered only the core genome of the pan-genome for this analysis.

**k-mer counting and dimensionality reduction.** Lists of k-mers of length 13 which occurred in at least one of the genome files were generated for each antibiotic using all the genomes for each antibiotic with GenomeTester4 (https://github.com/bioinfo-ut/GenomeTester4). These k-mers were then counted in each individual genome. All the counts were compiled into an *n* samples $\times$ *n*-k-mers + 1 matrix with the additional column for the resistance phenotype (0 for susceptible, 1 for resistant).

Because the number of k-mers was on average over 2 million for each antibiotic, we performed an initial downsampling of the k-mers to be used as features using pairwise testing between any given k-mer and the resistance phenotype. We used the chi-square test to find the 10,000 features most associated with the resistance phenotype.

**SNP counting.** The variant sites (SNPs) in the core genome alignment were extracted using a SNP sites tool (www.github.com/sanger-pathogens/snp-sites) based on homologous gene groups produced by Roary (https://github.com/sanger-pathogens/Roary). Each strain was processed according to its AMR phenotype for each antibiotic as follows. (i) First, the genome sequences were used to obtain the core gene sets, which are present in ≥99% of each set, ranging from 1,627 to 2,903 depending on the strain sets in each antibiotic. (ii) Each core gene nucleotide sequence was further aligned, and single nucleotide variants were identified. The position of a SNP in a gene was selected as a feature in the machine learning if the nucleotide varied in more than 5% of strains (i.e., was constant in less than 95% of strains). (iii) The data set of SNPs was assigned (1 for "A," 2 for "G," 3 for "T," and 4 for "C") for each allele of the strain as the matrix for machine learning.

**Machine learning.** The k-mers and SNPs (features) were analyzed using the gradient boosting classifier (GBC) model in scikit-learn (120) (v0.19.1) in Python (v3.6) using the default parameters. For both analyses, initially, the features were standardized by removing the mean and scaling to unit variance. The synthetic minority oversampling technique (SMOTE) (121) was applied to oversample data of minority class, compensating for unbalanced classes. For the k-mer analysis, the data were split randomly using a fivefold stratified cross-validation, while for the SNP analysis, the data were split in 70% for training and 30% for testing. In both analyses, 50 iterations were carried out, and the following four performance metrics were recorded for each classifier, P and N indicating positive and negative cases, respectively, and T indicating true (correct) and F indicating false (wrong) predictions:

- Recall (true positive rate [TPR]) = TP/P
- Precision (positive predictive value [PPV]) = TP/(TP + FP)
- Accuracy (ACC) = (TP + TN)/(P + N)
- Area under the receiver operator characteristic curve (AUC)

The mean of these 50 iterations was then used as the result statistic for the performance. Boxplots from the Seaborn (122) package were used to show the final prediction metrics. While the model was being simulated, we captured the maximum importance of each k-mer or SNP, as well as the number of times each k-mer or SNP was assigned an importance greater than zero. The features were ranked using the maximum importance, that is, the maximum weight that the feature contributes to the GBC in the 50 runs. Features that had a maximum importance of zero were removed from the results.

**k-mer searching and SNP filtering.** The GBC assigns an importance score between 0 and 1. This does not indicate which phenotype the k-mer is more associated with. To work out which phenotype the important k-mers were associated with, we compared the number of times a k-mer occurred in the susceptible condition to the number of times it occurred in the resistant condition. Next, we performed a chi-square test on these counts to determine whether there was a significant ($P > 0.05$) difference between the number of times a k-mer occurred in the susceptible or resistant condition. The k-mers found by the GBC were searched against the pan-genome of all our genomes using BLAST. The parameters for the search were as follows: E-value, 1,000; word size, 13 (same size as the k-mers); gap opening penalty, 5; and gap extension penalty, 2. The search hits were annotated by searching a gene transfer format (GTF) file corresponding to the pan-genome and by retrieving the information about each hit.

**Genome-scale metabolic model and flux balance analysis.** The cobra toolbox in python was used for all simulations. The model iML1515 (51) of *E. coli* K-12 MG1655 strain was downloaded from the BiGG database (123) using the cameo python toolbox (124). Flux balance analysis (FBA) and its variants were used to predict optimal flux distributions. FBA, based on linear programming, identifies the flux distribution that either minimizes or maximizes some objective function given a set of constraints (125). All simulations assumed M9 minimal medium (unless stated otherwise), such that the sulfate, phosphate, and ammonium were allowed to freely enter the system. Oxygen uptake was constrained to have a maximum uptake of 18.5 mmol/g (dry cell weight [DCW])/h to mimic aerobic conditions (126). A knockout model for each gene of interest was constructed by blocking all corresponding reactions to zero, given that the reaction is not catalyzed by an isozyme.

**Gene essentiality under various nutritional environments.** FBA was used with maximization of growth rate as the objective function to predict the lethality of each gene of interest (i.e., those associated with the important features from the SNPs and k-mers). We considered the essentiality of a gene under both rich medium conditions and M9 minimal medium conditions. To mimic rich medium conditions, the model was constrained to allow all carbon sources into the system, with a fixed uptake rate of 1 mmol/gDCW/h. If a feasible solution exists, while maximizing the biomass equation as the objective function, then

the knockout of the gene was not essential. To mimic M9 minimal medium conditions, the model was constrained so one individual carbon source had a maximum uptake of 10 mmol/gDCW/h. This simulation (minimal medium condition) was repeated for each carbon source in the model. The genes whose corresponding knockout model achieved a growth rate of 0.0001 h$^{-1}$ or less were considered essential.

**Calculating metabolite yields and construction of the gene-metabolite bipartite network.** A drain reaction was added to iML1515 for each metabolite in the model (i.e., a reaction that consumes the metabolite of interest). The maximum theoretical yield of each metabolite was calculated by setting its corresponding drain reaction as the objective function, with glucose as the only carbon source in aerobic minimal M9 medium conditions. The simulations were carried out for the wild-type model and each gene knockout model. The maximum theoretical yield of metabolite $i$ for the wild type (wt) and the $j$th gene knockout model is denoted as $y_{i,wt}$ and $y_{i,j}$, respectively.

The networkx package (127) in python was then used to construct a bipartite graph $G = (U, V, E)$, such that the nodes, $N$, represent genes ($U$) and metabolites ($V$), and the edges, $E$, connect a gene in $U$ to a metabolite in $V$. The $i$th metabolite is connected by an edge to the $j$th gene, if $y_{i,j} = 0$, given that $y_{i,wt} > 0$. Networkx's greedy modularity algorithm was applied to the network to assign genes and metabolites to clusters that were densely connected. The algorithm minimizes the number of interconnections between clusters, while maximizing the number of intraconnections. Cytoscape v3.7.1 was used for visualization of the clusters in the bipartite network (128).

**Flux balance impact degree and construction of the gene-reaction bipartite network.** We adapted the method by Zhou et al. (129) to model the effects of genetic mutations on metabolic fluxes. Here, flux variability analysis (FVA) was applied to the wild-type model and each knockout model using the cobra toolbox in python (130). FVA calculates the minimum and maximum flux through each reaction in the model, given a set of constraints, resulting in the range of possible fluxes for each reaction (flux span). FVA was simulated using glucose as the only carbon source in aerobic minimal M9 medium conditions. Note that reaction loops in the solution were not allowed.

Similar to before with metabolite yields, a bipartite graph $G = (U, W, E)$ was constructed using networkx, such that the nodes represent genes ($U$) and reactions ($W$), and the edges ($E$) connect a gene in $U$ to a reaction in $W$. The $i$th gene is connected by an edge to the $j$th reaction, if the knockout of the $i$th gene reduces the flux span by at least 10% compared to the wild type. As before, networkx's greedy modularity algorithm was applied to assign genes and metabolites to a cluster in order to identify groups of genes that have a similar impact on the metabolic fluxes. As with the metabolite yields, Cytoscape v3.7.1 was used to then visualize the clusters in the bipartite network.

**Gene pathway enrichment analysis.** We identified metabolic pathways that were enriched in each cluster of the bipartite networks using hypergeometric enrichment tests using the scipy function hypergeom (131). We considered a pathway as significantly enriched in a cluster if the false discovery rate (FDR) was less than 1% and used the Benjamini-Hochbery method for correction against multiple testing. We considered two sets of pathway lists for the enrichment. The first used the 40 subsystems as defined in the iML1515 GSM. A second list of pathways was downloaded from the BioCyc database using the SMART tables for *E. coli* (52), which provided a more extensive list of specific metabolic pathways.

**Data availability.** The accession numbers of the 3,616 *E. coli* genomes and the metadata available from PATRIC and used in this study, as well as the code used for the machine learning analysis and Jupyter notebooks for the genome-scale model analysis (including bipartite network generation and plotting of results and the Cytoscape networks), are available on https://github.com/tan0101/GSM_mSystems_2021.

## SUPPLEMENTAL MATERIAL

Supplemental material is available online only.
**FIG S1**, PDF file, 0.04 MB.
**FIG S2**, PDF file, 0.1 MB.
**FIG S3**, PDF file, 2 MB.
**FIG S4**, TIF file, 2.7 MB.
**FIG S5**, PDF file, 0.1 MB.
**FIG S6**, TIF file, 2.7 MB.
**FIG S7**, PDF file, 0.1 MB.
**TABLE S1**, XLS file, 0.4 MB.
**TABLE S2**, XLS file, 0.1 MB.
**TABLE S3**, XLS file, 2.4 MB.

## ACKNOWLEDGMENTS

This work was supported by the InnovateUK grant (104986), FARMWATCH (fight AbR with machine learning and a wide array of sensing technologies), and by the Ministry of Science and Technology of People's Republic of China under Grant Key Project of International Scientific and Technological Innovation Cooperation Between Governments (2018YFE0101500). The work was also supported by the University of Nottingham Internal

GCRF Scheme grant CARE Bangladesh (cholera antibiotic resistance in Bangladesh) big data mining and machine learning to improve diagnostics and treatment selection.

We gratefully acknowledge support received from the University of Nottingham Research Beacon of Excellence: Future Food and Green Chemicals.

Conceptualization, Nicole Pearcy and Tania Dottorini; Methodology, Nicole Pearcy, Yue Hu, Ning Xue, Alexandre Maciel-Guerra, Wei Wang, Jasmeet Kaler, Zixin Peng, Fengqin Li, and Tania Dottorini; Supervision, Zixin Peng, Fengqin Li, and Tania Dottorini; Writing, Editing & Reviewing the draft, Nicole Pearcy, Michelle Baker, and Tania Dottorini; Formal analysis and Visualization, Nicole Pearcy, Michelle Baker, Yue Hu, Alexandre Maciel-Guerra, and Ning Xue analyzed the data; Funding Acquisition, Fengqin Li, Zixin Peng, and Tania Dottorini.

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
