## [Reviewer comments · mSystems]

Genome-scale metabolic models and machine learning reveal genetic determinants of antibiotic resistance in *Escherichia coli* and unravel the underlying metabolic adaptation mechanisms.

Nicole Pearcy, Yue Hu, Michelle Baker, Alexandre Maciel Guerra, Ning Xue, Wei Wang, Jasmeet Kaler, Zixin Peng, Fengqin Li, and Tania Dottorini

Corresponding Author(s): Tania Dottorini, University of Nottingham

Review Timeline:

Submission Date:	September 16, 2020
Editorial Decision:	November 24, 2020
Revision Received:	April 12, 2021
Editorial Decision:	May 30, 2021
Revision Received:	June 12, 2021
Accepted:	June 24, 2021

Editor: Xiaoxia Lin

Reviewer(s): The reviewers have opted to remain anonymous.

Transaction Report:

DOI: <https://doi.org/10.1128/mSystems.00913-20>

November 24, 2020

Dr. Tania Dottorini
University of Nottingham
Loughborough
United Kingdom

Re: mSystems00913-20 (Genome-scale metabolic models and machine learning reveal genetic determinants of antibiotic resistance in *Escherichia coli* and unravel the underlying metabolic adaptation mechanisms.)

Dear Dr. Tania Dottorini:

Below you will find the comments of the reviewers.

To submit your modified manuscript, log onto the eJP submission site at <https://msystems.msubmit.net/cgi-bin/main.plex>. If you cannot remember your password, click the "Can't remember your password?" link and follow the instructions on the screen. Go to Author Tasks and click the appropriate manuscript title to begin the resubmission process. The information that you entered when you first submitted the paper will be displayed. Please update the information as necessary. Provide (1) point-by-point responses to the issues raised by the reviewers as file type "Response to Reviewers," not in your cover letter, and (2) a PDF file that indicates the changes from the original submission (by highlighting or underlining the changes) as file type "Marked Up Manuscript - For Review Only."

Due to the SARS-CoV-2 pandemic, our typical 60 day deadline for revisions will not be applied. I hope that you will be able to submit a revised manuscript soon, but want to reassure you that the journal will be flexible in terms of timing, particularly if experimental revisions are needed. When you are ready to resubmit, please know that our staff and Editors are working remotely and handling submissions without delay. If you do not wish to modify the manuscript and prefer to submit it to another journal, please notify me of your decision immediately so that the manuscript may be formally withdrawn from consideration by mSystems.

Sincerely,

Xiaoxia "Nina" Lin

Editor, mSystems

Journals Department
Reviewer comments:

Reviewer #1 (Comments for the Author):

This work presents a series of novel, flux balance analysis methods to better understand the metabolic effects of *E. coli* genes associated with AMR as determined by machine learning (or potentially any GWAS method). These analyses are technically sound and distinct from other approaches towards integrating FBA with ML, and appear to only require a metabolic model for the organism of interest to implement. However, the manuscript is in need of clarification regarding specifics of the data and methods employed, as well as potential improvements in how some of the key results are presented.

- (Line 165) Was there any filtering for assembly quality or completeness when selecting which genomes to use? Please specify.
- (Line 167) State Which AMR standard(s) the data comprised of (i.e. CLSI, EUCAST).
- (Line 182-193) What statistical test was used to determine correlation between AMR phenotype and sequence type/clonal complex? Are they significant under multiple hypothesis correction, assuming all antibiotic x organism x clonal complex cases were tested? Please specify.
- (Line 208) Can the authors comment on why the piperacillin (and possibly also cefuroxime) cases were more challenging for the GBC model?
- (Line 212) Were there any modifications to the GBC to achieve more robust and/or sparse models for feature selection, such as subsampling or limiting max features?
- (Line 215) How were features ranked to extract the top 10/50 features for each antibiotic? The methods mention that the number of non-zero importance cases, mean importance, and maximum importance were recorded for all features.
- (Line 230) Can the authors comment on why the kmer-based approach appears to consistently outperform the SNP-based approach for phenotype prediction? Please specify.
- (Line 240/Table S3) Future analysis of these AMR-associated SNPs may benefit from being characterized with respect to a reference genome. For example, what are the parE SNPs/resulting amino acid substitutions that are linked to fluoroquinolone AMR, relative to K12 MG1655?
- (Line 267) The analysis of metabolic systems among AMR genes may be strengthened by applying association tests for whether the AMR genes (across all or individual drugs) are enriched for a particular metabolic system compared to all other genes in iML1515.
- (Line 267/Figure 3d) For the hierarchical clustering, what data is being clustered and which distance metric and linkage were used? Please specify.
- (Line 434) "first computational pipeline that combined machine learning and genome-scale metabolic model analysis." Update this section with a comparison to another recent approach to

integrating FBA with ML for AMR: <https://doi.org/10.1038/s41467-020-16310-9>

- (Line 439) "first time by integrating the GSM with the ML" see above
- (Line 461) The discussion of GSM-related results does a good job of highlighting metabolic systems that are associated with multi-drug AMR through broad mechanisms. Drug-specific associations discovered here should also be emphasized.
- (Line 559) Assessing the metabolic effects of a given gene using the methods here appear to depend only on a metabolic model and not on the sequence or AMR data. A near-term extension of this work that may be worth discussing is the prospect of precomputing all of these deeper metabolic effects for each gene in a given GSM(s), so that future AMR GWAS studies can readily draw insights on potential AMR gene metabolic effects as predicted by these methods without needing to setup and solve all the GSM problems independently.
- (Line 615-616) What are "variants having $\leq 95\%$ constant nucleotides"?
- (Line 699-700) 100 carbon atom restriction could be made more clear with a formal description of the corresponding linear constraint(s) added to the model. Can the authors explain the rationale behind allowing the model to use any carbon sources in the FVA analysis?

- Minor edits
 - (Line 52) Current wording suggests that 52 known AMR genes were recovered, but Table 1+2 reports that only 22+3 known AMR genes were matched to the correct drug. Update to report separate counts for correctly associated genes and other AMR genes detected.
 - (Line 83) "bacterias" -> "bacteria's"
 - (Line 128) See above regarding known vs. correctly matched AMR genes.
 - (Line 200) "which strongly correlated to" -> "which are strongly correlated with"
 - (Line 244) See above regarding known vs. correctly matched AMR genes
 - (Line 273) Reference Supplement Table 1 for carbon sources tested
 - (Line 457) See above regarding known vs. correctly matched AMR genes
 - (Line 528) "effects" -> "affects"
 - (Figure 1) Make plot arrangement and dimensions consistent with Figure 2
 - (Figure 3b) May be useful to also present ratios between shared vs. combined metabolic genes (i.e. Jaccard index for each pair)
 - (Figure 3d) Update caption, currently reads "number of genes" but heatmap shows non-integer values
 - (Figure 5b-c) An alternative presentation of pie chart data that may better highlight which mechanisms/antibiotics are associated with which clusters is a heatmap or clustermap similar to Figure 3d, showing what fraction of each cluster is of a given mechanism/antibiotic.
 - (Figure 6b-c) See comment on Figure 5b-c

Reviewer #2 (Comments for the Author):

This article combines available knowledge, statistical inference and metabolic modeling to provide an increased understanding of specific metabolic processes that may contribute to confer resistance to specific antibiotics in *E. coli*. The work presented is overall creative and thought-provoking, and I feel it contributes an original and thorough analysis that could inform and inspire other researchers. However, I also think that there are multiple aspects of the writing that need substantial clarification and rephrasing:

1. A major point that I would like to bring up is that the rationale and hypothesis underlying the GSM analysis that is central to the paper is not clearly justified in the introduction and beginning of the result section, and barely justified later in the results. As a reader trying to follow the rationale of the

approach, I found it hard to figure out how and why the ML and GSM analysis can inform each other. For example, at line 117, the authors mention that GSM "offers a way of mechanistically evaluating the genetic determinants identified using ML". It is not clear what it means to evaluate a genetic determinant. If I understand correctly, what you are evaluating is really the role of that gene (or even more precisely the presence/absence, or different variants of that gene) on the resistance phenotypes. Furthermore, it is not clear nor obvious why the deletion of a gene should inform the resistance phenotype. A mutation/k-mer pattern could be in principle associated with the increase of expression of a gene. So it is not clear (as somehow implied in the text) if and why the deletion of a gene whose variation is correlated with resistance would help inform the underlying mechanisms of resistance. It is entirely possible (and, apparently consistent with the findings) that gene deletions end up being informative, but - again - to me this was not self-evident at the start. I would expect the authors to revise their presentation of the underlying motivation and hypothesis with more details, and a more precise description of why they would expect GSM to be informative. A similar unsatisfactory description of this link is also appearing in the first section of the results, where, towards the end (lines 156-158) the notion that GSM lethality of AMR-related genes would yield interesting results is mentioned very briefly and without a real rationale. The first hint to a rationale (one of many possible) appears only at line 273-275.

2. I am curious whether the authors considered using experimental gene deletion data to cross-validate some of the predictions used in the analysis. I understand that such data may not be available for all strains and conditions, but it may be nice to at least see a mention of how the availability of such data could impact your analysis or help refine it.

3. The phylogenetic analysis (starting at line 160) is potentially interesting but fairly disconnected from the rest of the manuscript. At first I was confused in trying to figure out whether and how that analysis informed the subsequent ML inference. I would suggest that the authors consider embedding that section differently in the flow of the manuscript, or at least make it very clear how it connects (or doesn't connect) to other portions.

4. Lines 256-259: It would be important to make portion clearer. If I understand correctly, SNPs are in (annotated) genes only, whereas k-mers could be anywhere in the genome. However it is not clear whether SNPs are only in the coding regions of the genes (it doesn't have to be the case). Also, I expect many genes to be non-metabolic and therefore not in the GSM, because only a portion (~1/4th?) of genes in *E. coli* are metabolic enzymes. So it is not clear what are the genes that the authors call "accessory genes": all the non-metabolic ones? In short, this whole part seems either misinformed about the connection between SNPs, k-mers and GSM, or just poorly explained.

5. Lines 289-290: The opening sentence of this paragraph doesn't make sense to me, making the whole paragraph a bit shaky.

6. Lines 307-308: There is no effect of a gene on a metabolite. Again, the setup of this portion could be much clearer with a sharper and more rigorous opening sentence. I gather from reading on (especially the methods) that the author meant effect of a gene on the producibility of a metabolite. Also: what metabolites? All metabolites or biomass components only? What is the rationale for either choice? These points can be clarified with minor sentence tweaks, but they can greatly enhance clarity.

7. Prior work (Brynildsen et al., *Nature Biotechnology* volume 31, pages 160-165) had used extended FBA models that included the production of ROS to study in detail the metabolic processes associated with cell death upon antibiotic killing. I would expect the authors to comment on the

relevance of this prior work to their study. Is there an overlap of emerging pathways, despite the distinct approaches? Would it be beneficial to extend the new methodology to an FBA model that explicitly includes ROS production?

Minor points:

Line 83: bacterias -> bacteria's

Line 145: Provide references for gradient boosting classifier

Dear Prof. Xiaoxia (Nina) Lin,

Manuscript # mSystems00913-20

“Genome-scale metabolic models and machine learning reveal genetic determinants of antibiotic resistance in *Escherichia coli* and unravel the underlying metabolic adaptation mechanisms” by Nicole Percy, Yue Hu, Michelle Baker, Alexandre Maciel Guerra, Ning Xue, Wei Wang, Jasmeet Kaler, Zixin Peng, Fengqin Li and Tania Dottorini.

We wish to thank the reviewers for the useful insight and comments, which no doubt contributed to improving this paper. We hope we have addressed all the issues identified. Before answering the specific questions, we would like to make the reviewers aware of some minor errors found in the AMR phenotype labels, which were identified when addressing the comments posed by the reviewers (see below), but required a complete re-run of the pipeline to ensure robustness of results.

Firstly, when searching for the additional information to include in the metadata file as requested by the first two comments from Reviewer 1, we identified some isolates that had been labelled with incorrect AMR phenotypes. Fortunately, this error occurred in only 22 of the isolates, and therefore very unlikely to have a major impact on the results, but nevertheless we considered it important to re-run the entire pipeline to avoid any bias with the results. Additionally, when investigating the reason for the poor performance of the piperacillin AMR model, as requested by Reviewer 1, we also identified that some antibiotic classes included isolates that had a mixture of AMR phenotypes for the single drug, as well as the drug combined with a compound (piperacillin + tazobactam, for example), which may have introduced some bias.

Furthermore, when making these corrections and re-collecting the data we found that the number of *E. coli* isolates included in the PATRIC database had been updated in September 2020 with 100s more isolates for some antibiotic classes. Due to the substantial change in data availability of the updated database, and since we had to rerun our analysis anyway due to the error described above, we decided to incorporate in latest updated list of *E. coli* isolates present in PATRIC into our new analysis. Although this caused a delay in submitting our response, the use of a larger dataset dramatically improved the performance of the machine learning models, whilst also demonstrating that our approach is robust as we found similar findings in both versions, as explained in the next answers. The main changes to results are summarised in the following:

- A total of 3616 isolates up from 1520 in the original manuscript.
- Number of isolates in each antibiotic class increased to the following:

Antibiotics	Original manuscript	Updated manuscript
ampicilin	436	2490
aztreonam	397	763
cefepime	267	1028
cefoxitin	240	592

cefuroxime	494	1903
ciprofloxacin	1043	2600
gentamicin	434	2579
levofloxacin	152	427
meropenem	105	1118
piperacillin	414	
tetracycline	177	494
tobramycin	236	1185
trimethoprim	309	978

- Average AUC for k-mer models increased from 76%-98% to 88%-98% (see also results lines 248-252).
- Average AUC for SNP models increased from 47%-99% to 75%-98% (see also results lines 346-372).
- Total known AMR genes identified increased from 139 (in total from all features) to 225 (which were just in the top ranked 10% of features) (see also results lines 330-333 and lines 381-386)
- 289 of the genetic determinants identified in iML1515 GSM, compared to 361

Note that due to the significant improvements to the performance of the supervised learners, we were able to use a more stringent threshold criteria of AUC > 95% for each antibiotic classifier (before we selected classifiers with AUC > 80%). Likewise, we limited our GSM analysis to the genetic determinants corresponding to the top 10% ranked features (before we used a 20% threshold) identified, for each antibiotic classifier with AUC > 95%, by both the k-mers and SNPs ML based methods. With reduced selection criteria, we aimed at embracing more certainty in the shared results by increasing robustness and reproducibility of claimed associations.

Importantly, many of the metabolic pathways, identified through the GSM analysis, which were described in the original manuscript, including lipopolysaccharides metabolism, phospholipid metabolism, purine and pyrimidine metabolism pathways, folate metabolism pathways (tetrahydrofolate), heme metabolism and energy metabolism pathways were coming out as significant, showing that our approach consistently identified similar metabolic processes in the previous and updated datasets. Due to re-running the pipeline, which meant the genes and clusters required re-analysing, and since we also improved the way in which we analysed the clusters (i.e. significance test based on the pathways in BioCyc and the model) as suggested by Reviewer 1, the results for the metabolite reproducibility and flux variability analysis, were partially rewritten. We provide the new results for the gene clustering analysis (metabolite reproducibility analysis and FVA analysis) under Reviewer 1's comment where the pathway enrichment analysis tests were suggested, or please see the revised main manuscript lines 606-889.

In the following, a point-by-point response to all the questions is provided. The original questions from the reviewers are in blue, whilst our responses are in black.

Reviewer #1 (Comments for the Author):

This work presents a series of novel, flux balance analysis methods to better understand the metabolic effects of *E. coli* genes associated with AMR as determined by machine learning (or potentially any GWAS method). These analyses are technically sound and distinct from other approaches towards integrating FBA with ML, and appear to only require a metabolic model for the organism of interest to implement. However, the manuscript is in need of clarification regarding specifics of the data and methods employed, as well as potential improvements in how some of the key results are presented.

- (Line 165) Was there any filtering for assembly quality or completeness when selecting which genomes to use? Please specify.

The *E. coli* isolates were filtered for assembly quality and status, as defined in PATRIC, when selecting which genomes to use. A 100% of the genomes we used in the new dataset have a 'good' quality labelling in PATRIC, which meet the following criteria, as described in Parrello et al., 2019 (<https://doi.org/10.1186/s12859-019-3068-y>): contamination -less than or equal to 10%, fine consistency – greater than or equal to 87% and completeness – greater than or equal to 80%. We also included an additionally filtering that removed out any 'plasmid-only' isolates and isolates with a contig number greater than 250, as also was done by Hyun et al., 2020 (1). Using this filtering criteria resulted in a total of 3616 isolates fulfilling all the above quality and completeness criteria. We have now updated the supplementary excel file provided on the dropbox folder 'PATRIC_metadata.xlsx' to include this new list of isolates and their resistance phenotypes, genome status and quality. The manuscript has been edited to describe the quality and completeness criteria used for collecting the new dataset (see Results lines 209-211, Methods lines 1310-1320 and below

Results - Line 209-211: Importantly, the genome sequences of these strains were all listed in PATRIC as 'good' quality assemblies, had less than 250 contigs and were labelled as either 'WGS' or 'Complete' as the genome status in PATRIC.

Methods – Lines: 1310-1320: All the genome sequences of the isolates that were used in this study were listed in PATRIC as 'good' quality assemblies. Isolates labelled 'good' quality in PATRIC meet the criteria set by Parrello et al., 2019 (117) that contamination is less than 10%, fine consistency is greater or equal to 87% and the completeness of the sequence is greater or equal to 80% We have also only included isolates that were labelled 'WGS' or 'Complete' in the genome status in PATRIC, which removes any cases that are 'plasmid-only'. Finally, we also included an additionally filtering that removed out any 'plasmid-only' isolates and isolates with a contig number greater than 250, as previously done by Hyun et al., 2020 (25).

25. Hyun JC, Kavvas ES, Monk JM, Palsson BO. 2020. Machine learning with random subspace ensembles identifies antimicrobial resistance determinants from pan-genomes of three pathogens. *PLoS Comput Biol* 16:e1007608.

117. Parrello B, Butler R, Chlenski P, Olson R, Overbeek J, Pusch GD, Vonstein V, Overbeek R. 2019. A machine learning-based service for estimating quality of genomes using PATRIC. BMC Bioinformatics 20:486.

- (Line 167) State Which AMR standard(s) the data comprised of (i.e. CLSI, EUCAST).

Thanks for the suggestion we have now updated the manuscript to include this information. AMR phenotypes were recorded based on PATRIC records. PATRIC collects AMR phenotype data generated using antimicrobial susceptibility testing methods (AST) from published studies and collaborators. From the records in PATRIC, the AMR standards included a mixture of CLSI and EUCAST standards. However, 1503 strains from the study PMID:28720578, 27 strains from study PMID:27208899, and 3 strains without publication records available were not supplied with the AMR standards information from both PATRIC database or the literature. For these isolates for which we could not find the AMR standard information we have written to the authors asking for this information to be provided if possible. We have since received information that the 27 strains from study PMID:27208899 were using EUCAST testing standard, but we are still waiting to hear back for the remaining. We have now added an additional column to the 'PATRIC_metadata.xlsx' file, available on the dropbox folder, which lists the laboratory method standard used to determine the AMR phenotypes, where available. We have updated the manuscript to include this information in the results lines 211-212 and methods lines 1305-1310, see also below

Results Lines 211-212: The genomes have experimentally measured AMR phenotype, which are annotated as either "susceptible" or "resistant".

Methods Lines 1305-1310: These AMR phenotypes were derived from laboratory analyses only and included a mixture of both Clinical and Laboratory Standard Institute (CLSI) and European Committee on Antimicrobial Susceptibility Testing (EUCAST) AMR standards. The list of the laboratory method standard used to determine the AMR phenotypes is detailed for each isolate in the supplementary excel file 'PATRIC_metadata.xlsx', which is provided on the dropbox folder.

- (Line 182-193) What statistical test was used to determine correlation between AMR phenotype and sequence type/clonal complex? Are they significant under multiple hypothesis correction, assuming all antibiotic x organism x clonal complex cases were tested? Please specify.

Chi-squared testing was used to test for significant correlations between groups. Bonferroni correction was applied to test for significance under multiple hypothesis correction. To make this clearer in the text the adjusted p values have now been supplied and the methods clarified.

However, Reviewer 2 correctly pointed out that this section of the manuscript had no link to other parts of the paper, so we therefore have removed this section.

- (Line 208) Can the authors comment on why the piperacillin (and possibly also cefuroxime) cases were more challenging for the GBC model?

As correctly pointed out by the reviewer this information was indeed relevant but missing in the manuscript. To address this question, we found that many of the isolates that were used for the piperacillin machine learning model, were actually resistant to piperacillin with tazobactam. After further inspection, we found that only 4 isolates were included that had an AMR phenotype for just piperacillin on its own, whilst 163 isolates had the AMR phenotype for the combined piperacillin and tazobactam. Tazobactam acts to inhibit the beta-lactamases, which are found in the piperacillin resistance isolates. One possible hypothesis is that the resistance mechanisms for bacteria exposed to a combination of antibiotics and drugs may not be occurring at the DNA level, but are more complex mechanisms that are changing the RNA expression. The paper <https://doi.org/10.1038/s41467-020-18668-2> found that isolates resistant to piperacillin with tazobactam occur due to higher expression of translocatable units, which results in a higher beta-lactamase copy-number. This suggests that additional features are required, and not just the DNA sequence, to identify such differences between the resistant and susceptible isolates. When rerunning the pipeline, due to the error identified thanks to Reviewer 1's first comment, we decided to use isolates that had single antibiotics for simplicity and consistency. Since the majority of isolates with available AMR phenotype are for piperacillin-tazobactam, we removed piperacillin when rerunning through the pipeline with the updated set of PATRIC isolates. This reduced the number of antibiotic classes in the new analysis to 12. We have slightly edited the methods lines 1303-1304, to make it clear that just single antibiotics were used, see also below

Methods lines 1303-1304: We selected genomes which were annotated as either "susceptible" or "resistant" to a single antibiotic.

- (Line 212) Were there any modifications to the GBC to achieve more robust and/or sparse models for feature selection, such as subsampling or limiting max features?

Again, thanks for the useful suggestion, as this part was indeed missing. We used the default parameters in the 'sklearn.ensemble.GradientBoostingClassifier' function. The subsampling parameter the default value is 1, which means that all the samples were used to fit the individual base learners. The default value for the maximum number of features is 'None', which means the maximum number of features is equal to the number of features in the input data. We did however, use a synthetic minority oversampling technique (SMOTE) (<https://dl.acm.org/doi/10.5555/1622407.1622416>) in the GBC to oversample data of minority class, compensating for unbalanced classes and hence to achieve more robust results for prediction for both k-mer and SNP analysis. SMOTE was implemented on the training part of the classifier to ensure the classes were balanced during the training phase to make sure the results were robust and not bias to any class. To obtain robust feature importance values, 50 simulations were conducted on random training (70%) and test (30%) split for each run for each antibiotic for the SNPs analysis. The mean of these 50 iterations was then used as the result statistic for the performance. Feature importance was calculated by the maximum importance score of each feature and ranked according to their maximum feature importance score over the 50 runs of the GBC. That is the maximum weighting that

the feature contributed in any of the 50 runs of the GBC. For the k-mers analysis, to obtain robust feature importance, the data were split randomly using a 5-fold stratified cross-validation. We have now updated the Methods to provide more details about the methodology. Thanks for this comment, this has given us the opportunity to improve our results lines 243-248 and lines 342-345 and the methods section lines 1388-1418 to make it more clear what was used, see also below

Results – Lines 243-248: For each classifier, 10000 features were selected based on the chi-square test. We used the performance metrics: accuracy, area under the curve (AUC), precision and recall to evaluate each model. A synthetic minority oversampling technique (SMOTE) was used to reduce the impact of unbalanced classes in the antimicrobial label groups and achieve robust classification results. The performance metrics were calculated as the mean of 50 simulations (Figure 1).

Results- Lines 342-345: Synthetic Minority Oversampling Technique (SMOTE) was applied to oversample data of minority class, compensating for unbalanced classes. The performance metrics were calculated as the mean of 50 simulations (Figure 2).

Methods – Lines 1388-1418: The k-mers and SNPs (features) were analysed using the gradient boosting classifier (GBC) model in scikit-learn (121) (v0.19.1) in Python (v3.6) using the default parameters. For both analyses, initially, the features were standardized by removing the mean and scaling to unit variance. Synthetic Minority Oversampling Technique (SMOTE) (122) was applied to oversample data of minority class, compensating for unbalanced classes. For the k-mer analysis, the data were split randomly using a 5-fold stratified cross-validation; while for the SNP analysis the data was split in 70% for training and 30% for testing. In both analyses, 50 iterations were carried out and the following four performance metrics were recorded for each classifier, P and N indicating positive and negative cases respectively, T indicating true (correct) and F false (wrong) predictions:

- Recall (TPR - true positive rate) = TP/P
- Precision (PPV - positive predictive value) = $TP/(TP + FP)$
- Accuracy (ACC) = $(TP + TN)/(P + N)$
- Area under the received-operator characteristic curve (AUC)

The mean of these 50 iterations was then used as the result statistic for the performance. Box plots from the Seaborn (123) package were used to show the final prediction metrics. While the model was being simulated, we captured the maximum importance of each k-mer or SNP, as well as the number of times each k-mer or SNP was assigned an importance greater than zero. The features were ranked using the maximum importance, that is, the maximum

weight that the feature contributes to the GBC in the 50 runs. Features that had a maximum importance of zero were removed from the results.

121. Pedregosa F, Varoquaux G, Gramfort A, Michel V, Thirion B, Grisel O, Blondel M, Prettenhofer P, Weiss R, Dubourg V, Vanderplas J, Passos A, Cournapeau D, Brucher M, Perrot M, Duchesnay É. 2011. Scikit-learn: Machine Learning in Python. *J Mach Learn Res* 12:2825–2830.
122. Chawla NV, Bowyer KW, Hall LO, Kegelmeyer WP. 2002. SMOTE: synthetic minority over-sampling technique. *J Artif Int Res* 16:321–357.
123. Waskom ML. 2021. Seaborn: statistical data visualization. *Journal of Open Source Software* 6:3021.

• (Line 215) How were features ranked to extract the top 10/50 features for each antibiotic? The methods mention that the number of non-zero importance cases, mean importance, and maximum importance were recorded for all features.

As correctly pointed out by the reviewer, in the original manuscript we have indeed given a misleading definition of how the features were ranked. The features were ranked according to their maximum feature importance. That is the maximum weighting that the feature contributed in any of the 50 runs of the GBC. We have updated the results lines 325-330 and methods lines 1407-1418 to make it clearer which measure was used, see also below.

Results- Line 325-330: The maximum importance in the 50 runs was captured for each k-mer. To identify important genes, the k-mers with a maximum importance greater than 0 for each antibiotic model, were cross referenced to the pan-genome for each antibiotic class. The identified k-mers, their corresponding genes and maximum importance obtained by the GBC are shown in Supplementary Table 1 and Supplementary Figure 3.

Methods – Lines 1407-1418: The features were ranked using the maximum importance, that is, the maximum weight that the feature contributes to the GBC in the 50 runs. Features that had a maximum importance of zero were removed from the results.

• (Line 230) Can the authors comment on why the kmer-based approach appears to consistently outperform the SNP-based approach for phenotype prediction? Please specify.

This is an interesting question. Thanks for the useful suggestion as we have now updated the manuscript to reflect this comment (See Discussion lines 962-1000 and below).

The SNP-based approach considered mutational changes in genes from the core genome of the isolates only and lacks the ability to consider the accessory genes (i.e. variable genes that do not occur in all strains). The k-mer based approach on the other hand considered the genes from both core and accessory genome. For this reason, resistance genes such as beta-lactamases, acquired by the bacteria via horizontal gene transfer, will only be identified as

strong discriminants of resistance/susceptibility via the K-mer based approach. Isolates resistant to ciprofloxacin and levofloxacin have previously been found to acquire chromosomal mutational in the genome and therefore likely the reason why the SNP-based approach performed very well for these two antibiotic classes. We suggest however, that using only the k-mer based approach, where often the acquired resistant genes are highly discriminant, is likely to dilute the importance of other additional resistance mechanisms, and so emphasise the advantage of combining both approaches. Importantly, we found the machine learning models for both levofloxacin and ciprofloxacin as two of the highest performers in our new and old results. The following was added to the discussion (lines 962-1000) of the manuscript to highlight this interesting point:

Discussion - Lines 962-1000: Notably, the k-mer based approach outperformed the SNP-based approach for 8 of the 11 antibiotics (AUC > 0.95), specifically: 8% higher in aztreonam, 17% higher in ampicillin, 2% higher in cefepime, 5% higher in ceftazidime, 6% higher in gentamicin, 28% higher in trimethoprim, 10% higher in tobramycin and 21% higher in tetracycline. A possible reason for this is due to the inclusion or exclusion of accessory genes in the two approaches. That is, the k-mer based approach allows for discriminating between resistance and susceptible according to both the core and accessory genome, whereas the SNP-based approach is restricted to the core genome only. The antibiotics that performed well only via the k-mer based approach may therefore be highly dependent on acquired resistance genes, such as the highly discriminant beta-lactamases. The SNP-based approach, however, successfully predicted AMR resistance for the two fluoroquinolone antibiotics ciprofloxacin and levofloxacin, and for the beta-lactam antibiotic, meropenem. Importantly, the SNP-based approach performed extremely well (AUC > 0.98) for ciprofloxacin, suggesting that the antibiotic induces mutations, which is consistent with the literature (50). Importantly, the k-mer- and SNP-based approaches identified different known AMR genes, validating the advantage of combining the important features from both approaches.

50. Bhatnagar K, Wong A. 2019. The mutational landscape of quinolone resistance in *Escherichia coli*. PLoS One 14:e0224650.

- (Line 240/Table S3) Future analysis of these AMR-associated SNPs may benefit from being characterized with respect to a reference genome. For example, what are the *parE* SNPs/resulting amino acid substitutions that are linked to fluoroquinolone AMR, relative to K12 MG1655?

We would like to thank to reviewer for this suggestion which supports the idea behind our ongoing research. We are currently working on developing a tool that can rank the AMR-associated SNPs based on their functional and 3D structural effects on the target protein that are linked to specific AMR phenotypes and relative to a reference genome such as K12 MG1655. As pointed out by the reviewer this would benefit the entire pipeline. We hope to be able to use this information to predict the functional changes to the genes, which could then lead to a deeper integration with the genome scale model (i.e. identifying whether a SNP has potentially caused an increase or decrease in reaction flux). More advanced GSM frameworks (GEM-PRO), are already available for integrating protein-structure information

into the models and therefore offer a way of developing our pipeline for future work. We now have added this suggestion to the discussion lines 1259-1276 (see also below), and we are now looking at developing a tool that will allow us to integrate this into our pipeline.

Discussion - Lines 1269-1276: The characterisation of the AMR-associated SNPs, in respect to a reference genome such as K12 MG1655, and would allow us to link the specific amino acid substitutions or deletions to antibiotic resistance. 1D-3D Structure-function prediction analysis may then enable us to determine whether the SNPs result in a loss or gain of function, which is directly integrated as constraints into models such as GEM-PRO. The effects of the SNPs on the genes (i.e. loss of function or gain of function) is not determined in our approach and if considered would allow further insights into the biological interpretation.

- (Line 267) The analysis of metabolic systems among AMR genes may be strengthened by applying association tests for whether the AMR genes (across all or individual drugs) are enriched for a particular metabolic system compared to all other genes in iML1515.

We would like to thank the reviewer for this useful point for improving the analysis. We have now added gene pathway enrichment analysis using a hypergeometric test to the analysis, as was carried out in <https://doi.org/10.1038/s41467-020-16310-9>. We carried out the test using the 40 defined metabolic subsystems in the iML1515 model as well as the gene- pathway association list included in the EcoCyc specialised SMART tables. The manuscript has been updated (see Results lines 448-462, discussion lines 1015-1026 and methods line 1520-1528, also see below). This useful point, as well as the paper <https://doi.org/10.1038/s41467-020-16310-9> that was also pointed out by Reviewer 1, also gave us the idea to test for the significance of the metabolic pathways that are enriched in each of the clusters in Figure 5 and Figure 6. In the previous analysis we considered the number of metabolites/reactions involved in each cluster and commented on metabolic pathways based on the frequency. However, we feel by considering a hypergeometric pathway enrichment test on the number of metabolites/reactions within each pathway strengthens our findings. Similar to the gene enrichment analysis, we used the 40 pathways already defined in the iML1515 model, as well as the metabolite-pathway and reaction- pathway association lists available in the SMART tables of the EcoCyc database. Notably, the reactions in BioCyc database are poorly annotated in the reactions of the iML1515, and therefore additional reaction-pathway associations were extracted using the gene-pathway annotation list, again downloaded from the EcoCyc database. To complement the text, for describing the most significantly enriched pathways, we have now included two new supplementary figures (Supplementary Figure 4 and 5). These figures show a network diagram created in Cytoscape of the genes in each cluster linked by edges to pathways that were significantly enriched in each pathway. Note that the colour of each cluster is co-ordinated with the clusters presented in the gene-metabolite and gene-reaction bipartite networks in Figure 5 and 6 of the main text. Importantly, using the significance tests found many of the pathway that were already being discussed in the previous results. A few additional pathways were identified, which were not discussed in the previous manuscript, including biotin and ppGpp. We have updated the results lines 607-749 and lines 753-889 and discussion lines 1104-1197 for the metabolite reproducibility analysis and the FVA

analysis, respectively. We also provide two new figure captions for the two supplementary figures 4 and 6. Note that these results (see below also) have undergone a considerable rewrite for these sections, due to firstly the clusters involving different genes due to updating the dataset and secondly due to the including this additional significance tests, however, the main findings and relation to AMR resistance, in general, remains the same as the previous manuscript.

Results – Lines 448-462: We performed gene pathway enrichment tests using the 40 metabolic subsystems included in iML1515 and also using the 352 gene-pathway annotation list downloaded from EcoCyc (52). The significant pathways with a false discovery rate (FDR) of less than 1% are provided in Supplementary Table 3.

We found genes enriched in amino acid metabolism (histidine and arginine), the pyrimidine salvage pathway, putrescine biosynthesis pathway and transport metabolism. Importantly, histidine metabolism has been found to play an important role in stress resistance in *E. coli* (53, 54), whilst the polyamine putrescine has been found to relieve the effects of oxidative stress in *E. coli* (55). Additionally, changes to genes involved in the pyrimidine salvage pathway have been found linked to the production of important biofilm components in *E. coli* (56), and therefore induce persistence (57). Furthermore, transport reactions are known to play a role in multidrug resistance by restricting the uptake of the antibiotic to reduce the toxicity (58, 59).

Results – Lines 607-749: The lethality of each genetic determinant on all metabolites in the iML1515 model was determined using flux balance analysis (FBA), as described in Materials and Methods. The results are represented as a bipartite graph of 98 genes and 508 metabolites. A gene is connected to a metabolite via an edge if its knockout results in preventing the metabolite's production. Using the Clauset-Newman-Moore greedy modularity maximisation algorithm, we clustered the genes and metabolites into groups of similar phenotypes (Supplementary Table 3). The largest 6 clusters are shown in Figure 5. The metabolites within each cluster are involved with a variety of metabolic processes, including cell wall metabolism, nucleotides metabolism, transport metabolism, alternative carbon metabolism, amino acid metabolism and cofactor and prosthetic group metabolism (Figure 5b). To test which of these metabolic systems was significantly being affected, we performed a pathway enrichment hypergeometric test on the metabolites in each cluster (see Materials and Methods). The most significant pathways associated with each cluster (FDR < 0.01) are shown in Supplementary Figure 4a-b.

A number of clusters could be linked to cell wall metabolites (Supplementary Figure 4, Supplementary Figure 5). Firstly, all 12 genes in cluster 5 affect the production of metabolites involved in lipopolysaccharide (LPS) metabolism. LPS are important compounds on the outer membrane and therefore have been found to play an important role in virulence (70, 71).

Additionally, the genes *murC*, *ftsI*, *dapE*, *glmM*, *murB*, *murG*, *mraY* and *mpl* in cluster 2 had a significant effect on the production of the metabolites involved in peptidoglycan (PG) metabolism. Peptidoglycan is a mesh-like structure that provides the strength and shape of the outer cell membrane, as well as providing protection against osmotic pressure. Modifications to PG can prevent the release of cell wall components, which initiate the host immune response (72), whilst also protecting the cell against antibiotic uptake (73). Similarly, changes to fatty acid oxidation and phospholipid (specifically CDP-diacylglycerol), whose production is affected by the 6 genes *purM*, *purD*, *ilvD*, *panD*, *purA* and *purL* in cluster 1, may also provide protection against the immune response. The immune system, for instance, has been found to take advantage of the antimicrobial properties of long chain fatty acids, which disrupt cell wall permeability when in excess in the extracellular environment (74). Pathogens have been found capable of modifying the biophysical properties of the cell membrane via changes to fatty acid structure, to increase the resistance to these antimicrobial peptides produced by the immune system (74).

In addition to cell wall metabolism, the genes in cluster 1 and cluster 4 are associated to a large number of pathways involved in purine and pyrimidine metabolism. Purine and pyrimidine are involved in the generation of DNA and RNA production, therefore changes to the genes in this cluster may be important in repairing DNA from ROS (75). Importantly, metabolomics analysis showed purine metabolism pathways were highly enriched in multidrug resistant *E. coli* strains (76). The genes involved in purine metabolism in cluster 1, *purL*, *purD*, *purM* and *purA* in particular, also have a downstream effect on many other metabolic pathways, including nitrogen metabolism, ppGpp metabolism and allantoin biosynthesis, all of which can be linked to the regulation of the stringent response (77-79). Importantly, changes in ppGpp concentration plays an important role in controlling cellular growth, and depletion of this metabolite has been found to trigger dormant cell metabolic state, promoting antibiotic-tolerant persistence cells (80, 81). *E. coli* cells starved of nitrogen have been found to increase ppGpp, which again has been found to induce tolerance to ciprofloxacin (78). Allantoin degradation has then been found as an important adaptive response to recovery after nitrogen starvation (77). Furthermore, these genes, as well as the gene *folP*, also affect metabolites involved in folate metabolism, tetrahydrofolate (THF) biosynthesis in particular. Importantly, point mutations in *folP* have been identified to prevent sulfonamides from inhibiting THF production (82). We identified *folP* in the trimethoprim, tetracycline and ampicillin ML models. Folate metabolism, including THF, however, are again important for nucleotides biosynthesis and have in fact been found important for persistence in *E. coli* cells exposed to ampicillin (83). The production of coenzyme A (CoA) is also affected by these genes, as well as the genes *ilvD* and *panD*. CoA is an important cofactor in many metabolic processes including fatty acid biosynthesis, which are used in lipopolysaccharides, and the TCA cycle. The concentration of acetyl-CoA, an important derivative of CoA, has also been found to play a key role in assessing the cell

metabolic state, which, in turn, determines the fate of either cell growth, survival or death (84).

The production of metabolites relating to iron metabolism were affected by genes in both clusters 2 and 6. The 4 genes in cluster 6 affect metabolites involved in heme biosynthesis. The capability (or improved capability) for heme synthesis may provide pathogens with a competitive advantage for colonisation, since heme is the largest source of iron for the cell. Excess heme however, increases the level of ROS and therefore is extremely toxic to the cell, so the regulation of heme concentration is essential (85). The genes *menD*, *entS*, *aroC*, *dxr*, *pheA*, *menC*, *menE*, *ispA* and *entD* in cluster 2 for instance, all affect enterobactin biosynthesis, either directly or via the chorismate biosynthesis pathway. Enterobactin is an iron scavenging siderophore and has been found important for pathogen virulence (86-88). An important response of the immune system is to use nutrient immunity by limiting iron availability, which has an important function in energy metabolism and DNA replication (89, 90). Changes to genes affecting iron metabolism may therefore enhance the resistance by improving their ability to scavenge iron from the environment. The genes *menD*, *dxr*, *aroC*, *menE*, *ispA* and *menC* also affect metabolites involved in the electron transport chain (ETC). Importantly, previous work has found reduced respiration via the ETC resulted in mutant strains highly resistant against ampicillin and gentamicin (91). The ETC reduces the proton motive force that is necessary for gentamicin uptake. Reduced flux through ETC however, also reduces the growth rate, which as previously discussed enables multidrug level persistence (60-62).

The genes in cluster 3 are also affecting metabolites involved in electron carriers metabolism. The 8 genes *cysJ*, *metK*, *asd*, *gss*, *cysH*, *cydC*, *metF* and *gshB* for instance are all affecting metabolites involved in glutathionylspermidine (GSP) biosynthesis. Importantly, GSP can be recycled back to glutathione and spermidine. Glutathione is an important antioxidant metabolite required for detoxifying ROS (92, 93), whilst spermidine is a polyamine also found to provide protection against ROS exposure (87). A subset of these genes, *cysJ*, *metK*, *asd*, *cysH* and *metF*, are also affecting biotin production. Importantly, biotin has been identified as important for the virulence of enteropathogenic (EPEC) *E. coli* strains, due to its involvement in the regulation of the locus of enterocyte enfacement (LEE). The LEE system is essential to these pathogenic bacteria in order to attach and infect host epithelium cells (95). Increased biotin concentrations have been shown to limit EHEC infections in mice (96).

In general, the metabolic processes described here are affected by genes identified in the ML models for diverse antibiotic classes (Figure 5c, Supplementary Figure 4c). This is not too surprising however, since these processes are suggested to increase antibiotic resistance via protection from the immune response, oxidative stress and/or the stringent response, which are multidrug adaption mechanisms for enhancing fitness, persistence and/or virulence (97-99).

Results – Lines 753 – 889: Next, we investigated the system level effect of the AMR conferring genes on metabolic fluxes. Specifically, flux variability analysis (FVA) was used to identify the biochemical reactions whose flux span was affected by mutations in the genetic determinants. The results are represented as a bipartite graph of 145 genes and 861 affected reactions. (Supplementary Table 3). A gene is connected to a reaction via an edge if its knockout results in reduced flux span through the reaction. As before, the Clauset-Newman-Moore greedy modularity maximisation algorithm was used to cluster the genes and reactions into groups of similar phenotypes (Supplementary Table 3). The largest 9 clusters are shown in Figure 6a-b, i.e. those with greater than 10 nodes (genes and metabolites). A variety of metabolic processes were enriched in the clusters (Figure 6b), similar to the gene-metabolite clusters. To test which of these metabolic systems was significantly being affected, we performed a pathway enrichment hypergeometric test on the reactions in each cluster. The most significant pathways associated with each cluster (FDR < 0.01) are shown in Supplementary Figure 6a-b.

The gene-reaction network was clustered into similar groups of genes to the gene-metabolite network. Again, the clusters were enriched with metabolic processes including cell wall metabolism (LPS, PG, fatty acids and phospholipids), nucleotides metabolism (purine, pyrimidine and folate metabolism) and iron metabolism (heme). The main differences between networks involve the set of genes in cluster 1. Unlike before, this analysis reveals the genes *gmhB*, *waaC*, *waaP*, *accA*, *lptG*, *lptF*, *waaF*, *hldD*, *fabG*, *lpxL*, *hlpE*, *fabD* and *glmM* are affecting the biosynthesis of nucleotide sugars. These sugars are incorporated into the O antigens region of LPS, which is located in immunodominant part of LPS (100). Furthermore, the genes *accA*, *nuoL*, *nuoN*, *fabG*, *tesA* and *fabD* are affecting fatty acid biosynthesis, as well as biotin biosynthesis. As discussed previously, both fatty acids and biotin metabolites can affect the host immune response and bacterial virulence.

Furthermore, the FVA analysis also revealed that the genes in cluster 7 all affect iron transport, which, as previously discussed, may be important for scavenging iron from the host. Additionally, disruptions to the genes in cluster 2, specifically *asd*, *gcvP*, *gcvT*, *serA*, *metF*, *cysH*, *cysJ*, and *serB*, were found to affect amino acid metabolism (cysteine, serine, glycine, aspartate and/or methionine), all of which are involved in folate transformation of *E. coli*. As previously discussed, folate metabolism can affect persistence to antibiotic exposure. Alternatively, however, sulfur amino acid residues in proteins, including methionine and cysteine, are found to be extremely reactive with reactive oxidative species, therefore changes to the genes specifically affecting these amino acids may play a role in ROS detoxification (101).

Again, these metabolic pathways could be associated to a diverse set of antibiotic classes (Figure 5c, Supplementary Figure 5c), suggesting the changes in these genes are linked to secondary multidrug adaptation mechanisms.

Discussion – Lines 1015-1026: The 289 total metabolic genes were significantly enriched in various metabolic pathways, including transport metabolism, nucleotides metabolism and amino acid metabolism. To understand the mechanistic effect of these 289 genes, we used flux balance analysis (FBA) to predict the system-level metabolic changes that result from genetic variants of the genes (i.e. mutations or absence). More specifically, we predicted metabolic phenotypes of genetic variants via gene knockouts and identified the metabolic processes that were being affected. Importantly, using our new ML-FBA integrated approach, we could reveal interesting links between genes and potential metabolic adaptation mechanisms that, importantly, were not identified using standard gene pathway enrichment analysis.

Discussion – Lines 1104-1197: Furthermore, clustering of genes according to metabolic phenotypes also revealed a strong link to cell wall metabolism adaptations. Genes were found to affect phospholipids, lipopolysaccharides, fatty acids and peptidoglycan metabolism, all of which can be associated to increased antibiotic tolerance via increased permeability of the membrane, as well as playing a role in virulence by manipulating the host immune response (73). Pathogens have been found to modify the cell wall components, for instance, that are usually recognised by the hosts innate immune response (74, 107). Changes to a number of genes that were affecting cofactor biosynthesis may also be involved in immune response manipulation, including the biosynthesis of biotin and iron. Increased biotin concentrations, for example, has been found to reduce the ability of EHEC *E. coli* to attach and infect host epithelium cells (95, 96). Furthermore, genes affecting both enterobactin metabolism and heme metabolism were also found, both of which may improve resistance to nutrient immunity by increasing the pathogens ability to scavenge iron from the environment (108, 109). Iron is important for many enzymes in bacteria, particularly those involved in oxidative phosphorylation and DNA synthesis, therefore is essential for the bacteria's survival.

Purine and pyrimidine metabolism was also enriched in the gene clusters. Modifications to these genes may limit the inhibitory effect of antibiotics that target DNA replication, such as ciprofloxacin and levofloxacin. Importantly however, the genes encoded by purine and pyrimidine biosynthesis enzymes have a large system-level effect involving many different metabolic processes. The genes, *purL*, *purD*, *purA* and *purM* in particular, affect the production of DNA building blocks, which may be important for DNA repair against antibiotic-induced ROS (75). Furthermore, these genes also affect ppGpp metabolism, which is important for regulating cellular growth and inducing antibiotic-induced persistence (80, 81). Additionally, these genes also affect the production of important cofactors of energy metabolism, such as ATP, NAD and NADPH, which are important for the electron transport

chain (ETC). Other ETC metabolites, including ubiquinone, menaquinone and flavin, were also being affected by the important genetic determinants. Changes in the flux through ETC may contribute to antibiotic resistance in a number of ways. Reduced ETC reduces the PMF required for aminoglycoside uptake (110), whilst also reducing the growth rate for increased persistence (56-58). Furthermore, the ETC reactions are also responsible for ROS-production. A related study that applied gene KO simulations on an extended GSM of *E. coli*, which included specific ROS-producing reactions, identified genes associated to the ETC as ROS-inducing targets for improved antibiotic killing (111). Further evidence to suggest adaption to ROS was found by a number of additional genes, whose knockout was found to affect glutathione, spermidine, methionine or cysteine biosynthesis. Importantly, these metabolites have all previously been found to provide protection of *E. coli* cells by acting as antioxidants (92, 93, 101).

Methods – Lines 1520-1227: We identified metabolic pathways that were enriched in each cluster of the bipartite networks using hypergeometric enrichment tests using the *scipy* function *hypergeom* (132). We considered a pathway as significantly enriched in a cluster if the false discovery rate (FDR) was less than 1% and used the Benjamini-Hochberg method for correction against multiple testing. We considered two sets of pathway lists for the enrichment. The first used the 40 subsystems as defined in the iML1515 GSM. A second list of pathways was downloaded from the BioCyc database using the SMART tables for *E. coli* (52), which provided a more extensive list of specific metabolic pathways.

Figure caption for new supplementary Figure 4 - S4 Fig. a) Network diagram showing the significantly enriched pathways in each cluster in Figure 5a. Genes are connected to a metabolic pathway, if the gene affected the production of at least 1 metabolite in the pathway. Note that the colours of the gene clusters are co-ordinated with the clusters in Figure 5a. b) Heatmap showing the number of genes in each cluster that is connected, by at least 1 metabolite, to the significant metabolic pathway. c) Heatmap showing the number genes in each antibiotic ML model that affected at least 1 metabolite in the significant pathway.

Figure caption for new supplementary Figure 6: S6 Fig. a) Network diagram showing the significantly enriched pathways in each cluster in Figure 6a. Genes are connected to a metabolic pathway, if the gene affected the flux span of at least 1 reaction in the pathway. Note that the colours of the gene clusters are co-ordinated with the clusters in Figure 6a. b) Heatmap showing the number of genes in each cluster that is connected, by at least 1 reaction, to the significant metabolic pathway. c) Heatmap showing the number genes in each antibiotic ML model that affected at least 1 reaction in the significant pathway.

52. Karp PD, Riley M, Saier M, Paulsen IT, Collado-Vides J, Paley SM, Pellegrini-Toole A, Bonavides C, Gama-Castro S. 2002. The ecocyc database. *Nucleic acids research* 30:56-58.
53. Christgen SL, Becker DF. 2019. Role of proline in pathogen and host interactions. *Antioxidants & redox signaling* 30:683-709.
54. Nagao T, Nakayama-Imaohji H, Elahi M, Tada A, Toyonaga E, Yamasaki H, Okazaki K, Miyoshi H, Tsuchiya K, Kuwahara T. 2018. L-histidine augments the oxidative damage against Gram-negative bacteria by hydrogen peroxide. *Int J Mol Med* 41:2847-2854.
55. Tkachenko AG, Akhova AV, Shumkov MS, Nesterova LY. 2012. Polyamines reduce oxidative stress in *Escherichia coli* cells exposed to bactericidal antibiotics. *Res Microbiol* 163:83-91.
56. Garavaglia M, Rossi E, Landini P. 2012. The Pyrimidine Nucleotide Biosynthetic Pathway Modulates Production of Biofilm Determinants in *Escherichia coli*. *PLOS ONE* 7:e31252
57. Hansen S, Lewis K, Vulić M. 2008. Role of Global Regulators and Nucleotide Metabolism in Antibiotic Tolerance in *Escherichia coli*. *Antimicrobial Agents and Chemotherapy* 52:2718-2726.
58. Bolhuis H, van Veen HW, Poolman B, Driessen AJ, Konings WN. 1997. Mechanisms of multidrug transporters. *FEMS microbiology reviews* 21:55-84.
59. Levy S. 2002. Active efflux, a common mechanism for biocide and antibiotic resistance. *Journal of applied microbiology* 92:65S-71S.
60. Pontes MH, Groisman EA. 2019. Slow growth determines nonheritable antibiotic resistance in *Salmonella enterica*. *Sci Signal* 12.
61. Brauner A, Fridman O, Gefen O, Balaban NQ. 2016. Distinguishing between resistance, tolerance and persistence to antibiotic treatment. *Nat Rev Microbiol* 14:320-30.
62. Greulich P, Scott M, Evans MR, Allen RJ. 2015. Growth-dependent bacterial susceptibility to ribosome-targeting antibiotics. *Mol Syst Biol* 11:796.
70. Tan Y, Kagan JC. 2014. A cross-disciplinary perspective on the innate immune responses to bacterial lipopolysaccharide. *Mol Cell* 54:212-23.
71. Matsuura M. 2013. Structural Modifications of Bacterial Lipopolysaccharide that Facilitate Gram-Negative Bacteria Evasion of Host Innate Immunity. *Front Immunol* 4:109.
72. Irazoki O, Hernandez SB, Cava F. 2019. Peptidoglycan Muropeptides: Release, Perception, and Functions as Signaling Molecules. *Front Microbiol* 10:500.

73. Huang KC, Mukhopadhyay R, Wen B, Gitai Z, Wingreen NS. 2008. Cell shape and cell-wall organization in Gram-negative bacteria. *Proc Natl Acad Sci U S A* 105:19282-7.
74. Zhang Y-M, Rock CO. 2008. Membrane lipid homeostasis in bacteria. *Nature Reviews Microbiology* 6:222-233.
75. Shaffer CL, Zhang EW, Dudley AG, Dixon BREA, Guckes KR, Breland EJ, Floyd KA, Casella DP, Algood HMS, Clayton DB, Hadjifrangiskou M. 2017. Purine Biosynthesis Metabolically Constrains Intracellular Survival of Uropathogenic *Escherichia coli*. *Infection and Immunity* 85:e00471-16.
76. Lin Y, Li W, Sun L, Lin Z, Jiang Y, Ling Y, Lin X. 2019. Comparative metabolomics shows the metabolic profiles fluctuate in multi-drug resistant *Escherichia coli* strains. *Journal of Proteomics* 207:103468.
77. Switzer A, Burchell L, McQuail J, Wigneshweraraj S. 2020. The Adaptive Response to Long-Term Nitrogen Starvation in *Escherichia coli* Requires the Breakdown of Allantoin. *Journal of Bacteriology* 202:e00172-20.
78. Brown DR. 2019. Nitrogen Starvation Induces Persister Cell Formation in *Escherichia coli*. *Journal of bacteriology* 201:e00622-18.
79. Poole K. 2012. Bacterial stress responses as determinants of antimicrobial resistance. *Journal of Antimicrobial Chemotherapy* 67:2069-2089.
80. Yee R, Cui P, Shi W, Feng J, Zhang Y. 2015. Genetic Screen Reveals the Role of Purine Metabolism in *Staphylococcus aureus* Persistence to Rifampicin. *Antibiotics (Basel)* 4:627-42.
81. Pedley AM, Benkovic SJ. 2017. A New View into the Regulation of Purine Metabolism: The Purinosome. *Trends Biochem Sci* 42:141-154.
82. Vedantam G, Guay GG, Austria NE, Doktor SZ, Nichols BP. 1998. Characterization of mutations contributing to sulfathiazole resistance in *Escherichia coli*. *Antimicrob Agents Chemother* 42:88-93.
83. Morgan J, Smith M, Mc Auley MT, Salcedo-Sora JE. 2018. Disrupting folate metabolism alters the capacity of bacteria in exponential growth to develop persisters to antibiotics. *bioRxiv* doi:10.1101/335505:335505.
84. Shi L, Tu BP. 2015. Acetyl-CoA and the regulation of metabolism: mechanisms and consequences. *Curr Opin Cell Biol* 33:125-31.
85. Anzaldi LL, Skaar EP. 2010. Overcoming the Heme Paradox: Heme Toxicity and Tolerance in Bacterial Pathogens. *Infection and Immunity* 78:4977-4989.

86. Contreras H, Chim N, Credali A, Goulding CW. 2014. Heme uptake in bacterial pathogens. *Curr Opin Chem Biol* 19:34-41.
87. Nairz M, Schroll A, Sonnweber T, Weiss G. 2010. The struggle for iron - a metal at the host-pathogen interface. *Cell Microbiol* 12:1691-702.
88. Adler C, Corbalan NS, Peralta DR, Pomares MF, de Cristóbal RE, Vincent PA. 2014. The Alternative Role of Enterobactin as an Oxidative Stress Protector Allows *Escherichia coli* Colony Development. *PLOS ONE* 9:e84734.
89. Cassat JE, Skaar EP. 2013. Iron in infection and immunity. *Cell host & microbe* 13:509-519.
90. Chen GY, Ayres JS. 2020. Beyond tug-of-war: Iron metabolism in cooperative host-microbe interactions. *PLOS Pathogens* 16:e1008698.
91. Lobritz MA, Belenky P, Porter CBM, Gutierrez A, Yang JH, Schwarz EG, Dwyer DJ, Khalil AS, Collins JJ. 2015. Antibiotic efficacy is linked to bacterial cellular respiration. *Proceedings of the National Academy of Sciences of the United States of America* 112:8173-8180.
92. Ezraty B, Gennaris A, Barras F, Collet JF. 2017. Oxidative stress, protein damage and repair in bacteria. *Nat Rev Microbiol* 15:385-396.
93. Dwyer DJ, Belenky PA, Yang JH, MacDonald IC, Martell JD, Takahashi N, Chan CT, Lobritz MA, Braff D, Schwarz EG, Ye JD, Pati M, Vercruyse M, Ralifo PS, Allison KR, Khalil AS, Ting AY, Walker GC, Collins JJ. 2014. Antibiotics induce redox-related physiological alterations as part of their lethality. *Proc Natl Acad Sci U S A* 111:E2100-9.
94. Rider JE, Hacker A, Mackintosh CA, Pegg AE, Woster PM, Casero RA, Jr. 2007. Spermine and spermidine mediate protection against oxidative damage caused by hydrogen peroxide. *Amino Acids* 33:231-40.
95. Furniss RCD, Clements A. 2017. Regulation of the Locus of Enterocyte Effacement in Attaching and Effacing Pathogens. *Journal of bacteriology* 200:e00336-17.
96. Connolly JPR, Finlay BB, Roe AJ. 2015. From ingestion to colonization: the influence of the host environment on regulation of the LEE encoded type III secretion system in enterohaemorrhagic *Escherichia coli*. *Frontiers in Microbiology* 6.
97. Keren I, Shah D, Spoering A, Kaldalu N, Lewis K. 2004. Specialized persister cells and the mechanism of multidrug tolerance in *Escherichia coli*. *Journal of bacteriology* 186:8172-8180.

98. Koutsolioutsou A, Peña-Llopis S, Demple B. 2005. Constitutive *soxR* mutations contribute to multiple-antibiotic resistance in clinical *Escherichia coli* isolates. *Antimicrobial agents and chemotherapy* 49:2746-2752.
99. Maisonneuve E, Gerdes K. 2014. Molecular Mechanisms Underlying Bacterial Persisters. *Cell* 157:539-548.
100. Lerouge I, Vanderleyden J. 2002. O-antigen structural variation: mechanisms and possible roles in animal/plant-microbe interactions. *FEMS Microbiol Rev* 26:17-47.
101. Bin P, Huang R, Zhou X. 2017. Oxidation Resistance of the Sulfur Amino Acids: Methionine and Cysteine. *Biomed Res Int* 2017:9584932.
107. Ramadurai S, Sarangi NK, Maher S, MacConnell N, Bond AM, McDaid D, Flynn D, Keyes TE. 2019. Microcavity-Supported Lipid Bilayers; Evaluation of Drug-Lipid Membrane Interactions by Electrochemical Impedance and Fluorescence Correlation Spectroscopy. *Langmuir* 35:8095-8109.
108. Oexle H, Gnaiger E, Weiss G. 1999. Iron-dependent changes in cellular energy metabolism: influence on citric acid cycle and oxidative phosphorylation. *Biochim Biophys Acta* 1413:99-107.
109. Schrettl M, Haas H. 2011. Iron homeostasis--Achilles' heel of *Aspergillus fumigatus*? *Curr Opin Microbiol* 14:400-5.
110. Taber HW, Mueller JP, Miller PF, Arrow AS. 1987. Bacterial uptake of aminoglycoside antibiotics. *Microbiol Rev* 51:439-57.
111. Brynildsen MP, Winkler JA, Spina CS, MacDonald IC, Collins JJ. 2013. Potentiating antibacterial activity by predictably enhancing endogenous microbial ROS production. *Nature biotechnology* 31:160-165.
132. Oliphant TE. 2007. Python for scientific computing. *Computing in Science & Engineering* 9:10-20.

- (Line 267/Figure 3d) For the hierarchical clustering, what data is being clustered and which distance metric and linkage were used? Please specify.

The seaborn python package was used to carry out the hierarchical clustering in Figure 3d. This package calls the scipy python package with the default parameters set as: single linkage and Euclidean distance as the metric. We have added this information to the legend of Figure 3d, see below. Note also that the number of genes presented in the matrix has undergone columns standardisation. That is, for each column the minimum is subtracted and then divided by the maximum value.

Figure 3 (legend): (d) Heatmap showing the normalised number of genes associated to each metabolic system. Note that the number of genes was normalised via column standardisation. Note that hierarchical clustering was applied to both rows (metabolic systems) and columns (antibiotic classes) using the single linkage method and Euclidean distance as the metric. Each subplot shows the results for the top 10% of genes identified in each AMR classifier. Subplot b, c, and d show the results for the 289 genes found by combining the genes that correspond to the features in the top 10% of the k-mer and SNPs classifications.

- (Line 434) "first computational pipeline that combined machine learning and genome-scale metabolic model analysis." Update this section with a comparison to another recent approach to integrating FBA with ML for AMR: <https://doi.org/10.1038/s41467-020-16310-9>

Thank you for highlighting this paper, their approach is very interesting. We have modified the discussion lines 898-957, see also below:

Discussion - Lines 898-957: Kavas (30) have recently developed the first computational pipeline that combines machine learning with genome scale metabolic models to enable biochemical interpretation of genetic determinants. In their pipeline, the effect of alleles on the flux solution space was used to successfully classify AMR phenotypes of *Mycobacterium tuberculosis* strains. In our work, we take an alternative two-step approach. First, a combination of a k-mer and SNP-based machine learning approach is used to identify genetic determinants. Second, a genome scale metabolic model is used to assess the effect of genetic determinants on metabolite producibility and biochemical fluxes to elucidate possible metabolic adaptive mechanisms. Our approach produced AMR models of *E. coli* that achieved performance accuracies competitive with the current approaches. Moreover, we were able to reveal novel biomarkers based on the systemic effect the genetic determinants have on growth, metabolite yields and metabolic fluxes.

30. Kavas ES, Yang L, Monk JM, Heckmann D, Palsson BO. 2020. A biochemically-interpretable machine learning classifier for microbial GWAS. *Nature Communications* 11:2580.

- (Line 439) "first time by integrating the GSM with the ML" see above
See above.

- (Line 461) The discussion of GSM-related results does a good job of highlighting metabolic systems that are associated with multi-drug AMR through broad mechanisms. Drug-specific associations discovered here should also be emphasized.

As before with the old analysis, genes linked to the primary antibiotic resistance mechanisms for each antibiotic were identified and presented in Table 1 and 2, validating that the

machine learning models are good classifiers of AMR. The genes involved in primary mechanisms, however, are often acquired genes, such as the beta-lactamase, and for that reason not included in the iML1515 GSM. Changes (i.e. absence/presence or mutations) to the 289 genes identified in the metabolic model iML1515 appear to be responsible for secondary adaptation mechanisms, such as adapting to the host immune response, persistence and reduction of reactive oxidative stress. We have analysed whether any metabolic processes could be linked to single antibiotics or antibiotic classes, but could not find any correlations, instead we expect these metabolic adaptations are a more generic response to antibiotic exposure, and in that way may be useful for developing new treatments that induce antibiotic efficacy by reducing the adaptation capability that provides the bacteria resistance. We have modified the discussion lines 1199-1203 and lines 1211-1237 to highlight this point:

Discussion - Lines 1199-1203: Importantly, the genetic determinants associated with the metabolic adaptation mechanisms described here, were identified in the ML-models for diverse antibiotic classes. Changes in these genes are therefore suggested to be contributing to secondary resistance mechanisms via a generic response against toxicity and stress, which is nonetheless essential for their survival (97-99).

Discussion – Lines 1211-1237: Targeting the most important genetic determinants with the highest effect on these secondary adaptation mechanisms, whilst simultaneously targeting essential metabolic processes however, may provide novel new treatments that increase antibiotic efficacy (112).

97. Keren I, Shah D, Spoering A, Kaldalu N, Lewis K. 2004. Specialized persister cells and the mechanism of multidrug tolerance in *Escherichia coli*. *Journal of bacteriology* 186:8172-8180.
98. Koutsolioutsou A, Peña-Llopis S, Demple B. 2005. Constitutive *soxR* mutations contribute to multiple-antibiotic resistance in clinical *Escherichia coli* isolates. *Antimicrobial agents and chemotherapy* 49:2746-2752.
99. Maisonneuve E, Gerdes K. 2014. Molecular Mechanisms Underlying Bacterial Persisters. *Cell* 157:539-548.
112. Martin II JK, Sheehan JP, Bratton BP, Moore GM, Mateus A, Li SH-J, Kim H, Rabinowitz JD, Typas A, Savitski MM. 2020. A dual-mechanism antibiotic kills gram-negative bacteria and avoids drug resistance. *Cell* 181:1518-1532. e14.

- (Line 559) Assessing the metabolic effects of a given gene using the methods here appear to depend only on a metabolic model and not on the sequence or AMR data. A near-term extension of this work that may be worth discussing is the prospect of precomputing all of these deeper metabolic effects for each gene in a given GSM(s), so that future AMR GWAS

studies can readily draw insights on potential AMR gene metabolic effects as predicted by these methods without needing to setup and solve all the GSM problems independently.

We would like to thank the reviewer for this interesting suggestion, which we had not thought about previously. We shall explore ways in which we can develop a new tool for storing this data and analysing/visualising the results, which would be easy for other researchers to use, without the need for genome scale modelling experience. We have included the following sentence in the discussion lines 1239-1249, see also below:

Discussion - Lines 1239-1249: Our new approach can be applied to study genetic determinants of any pathogen of interest, providing a large cohort of AMR phenotypes are available and a genome scale metabolic model exists for a reference genome. The second step of our approach depends only on the GSMM and therefore precomputing the metabolic changes (e.g. effects on metabolite yields or metabolic fluxes) for the entire set of genes in the model is possible, which could be readily available for future AMR studies to draw insights on potential new AMR genes. Future efforts may precompute all of these deeper metabolic effects for each gene in a given GSM(s). Such future endeavours will offer the possibility to future AMR GWAS studies to readily draw insights on potential AMR gene metabolic effects as predicted by these methods without needing to setup and solve all the GSM problems independently.

- (Line 615-616) What are "variants having $\leq 95\%$ constant nucleotides"?

The features in the SNPs based approach were selected based on this criterion. The features relate to the aligned position of single nucleotide variants. An allele (with a specific nucleotide variant) was included in the analysis if it was frequent in more than 5% of the population of strains. As an example, let's say we have 100 strains in the population set each with either the nucleotide A or G in position 1 of gene X. If 95 of these strains have nucleotide A in position 1 and 5 of the strains have nucleotide G in position 1, then position 1 of gene X would be selected as a feature for the classifier. If, however, the number of strains with A in position 1 was 96 or greater, then position 1 would not be included. We have added some additional information to the methods lines 1378-1381 to hopefully make this clearer.

Methods- Lines 1378-1381: Each core gene nucleotide sequence was further aligned, and single nucleotide variants were identified. The position of a SNP in a gene was selected as a feature in the machine learning if the nucleotide varied in more than 5% of strains (i.e., was constant in less than 95% of strains).

- (Line 699-700) 100 carbon atom restriction could be made more clear with a formal description of the corresponding linear constraint(s) added to the model. Can the authors explain the rationale behind allowing the model to use any carbon sources in the FVA analysis?

We agree that the reasoning behind using a 100 carbon atom restriction for the FVA analysis and not for the metabolite yields was confusing and inconsistent. Our original reasoning was

to allow for a rich media to be considered, and alternative carbon sources, to be taken into account in flux changes. We have carried out the analysis for the FVA on the new set of 289 genes using rich media (i.e. 100 carbon atom restriction with all carbon sources open) and also on minimal media with glucose as the carbon source. We found a difference of only 27 genes, all of which related to alternative carbon metabolism. In the previous section of the manuscript titled "GSM knockout analysis reveals genes related to growth-limitation, auxotrophic behaviour and alternative carbon source utilisation", we already investigated which genes were affecting growth on alternative carbon metabolism. In this analysis, we identified the majority (22/27) of these genes that differed between the two environmental conditions. For that reason, we used glucose minimal media conditions for both metabolite and FVA analysis, to ensure consistency. The approach was re-ran on a standard laptop and so we also removed the sentence referring to using the University of Nottingham's HPC. The methods lines 1494-1496 has been changed to the following:

Methods - Lines 1494-1496: FVA was simulated using glucose as the only carbon source in aerobic minimal M9 media conditions. Note that reaction loops in the solution were not allowed.

- **Minor edits**

- (Line 52) Current wording suggests that 52 known AMR genes were recovered, but Table 1+2 reports that only 22+3 known AMR genes were matched to the correct drug. Update to report separate counts for correctly associated genes and other AMR genes detected.

The number of known AMR genes cross-matched with CARD and mutationDB databses was a lot higher in the new results. We found 225 genes that were found in the machine learning models in the top ranked 10% of features, which are also a known AMR gene in the database for any antibiotic class. The number of AMR genes that have been found specifically for the matching antibiotic is 35. Note however, that some genes will not have been tested for that specific antibiotic or may not be listed in CARD and mutationDB. The gene *folP* is one example where it is listed in mutationDB only for sulfonamide, but evidence in the literature has reported the resistance to ampicillin from changes to this gene. Also, note that for mutationDB there are no entries for levofloxacin or meropenem, hence this is why the 'Known AMR genes to the antibiotic' column in Table 2 is empty for these antibiotics. We have updated the abstract lines 39-40, the introduction lines 152-156, the results lines 330-333 and lines 381-386 and the discussion lines 1002-1003 to the new values and to make it more clear. Tables 1 and 2 have also been updated to the new list of AMR genes, see below also. In the previous manuscript, we don't think we made it clear what the AMR genes in the table 1 and 2 corresponded to. We have now listed in the Tables the AMR genes found in the top 10% of features, since this matches what was being used for the GSM analysis.

Abstract – Lines 39-40: First, our approach corroborates 225 known AMR-conferring genes, 35 of which are known for the specific antibiotic.

Introduction – Lines 152-156: Using our approach we firstly were able to accurately predict AMR resistant and susceptible phenotypes against 12 different antibiotics, as well as identifying 225 (35 of which were matching to the specific antibiotic class reported in AMR-related databases) known AMR-conferring genes in 3616 *E. coli* isolates.

Results – Lines 330-333: When mapped to the CARD (48) and MutationDB databases (49), 84 unique AMR genes were identified in the top 10% of features (ranked according to the maximum weight found in the 50 runs), 25 of which had evidence of the AMR gene for the specific antibiotic class (Table 1).

Results – Lines 381-386: By comparisons with the CARD and mutationDB databases, we identified 146 unique AMR genes associated to at least 1 antibiotic (Table 2) that were in the top 10% of features (ranked according the maximum feature importance in the 50 runs). Out of these 146 genes, 8 had evidence in the database of the AMR gene for the specific antibiotic class (Table 2). Note however, that mutationDB does not include entries for AMR genes for the levofloxacin and meropenem antibiotics.

Discussion – Lines 1002-1003: The combined approaches identified 225 known AMR genes corresponding to the top 10% of ranked features recognised as discriminant by the AMR classifiers. Out of these 225 genes, 35 were matching to the specific antibiotic class that has been reported in the databases.

44. Alcock BP, Raphenya AR, Lau TTY, Tsang KK, Bouchard M, Edalatmand A, Huynh W, Nguyen AV, Cheng AA, Liu S, Min SY, Miroshnichenko A, Tran HK, Werfalli RE, Nasir JA, Oloni M, Speicher DJ, Florescu A, Singh B, Faltyn M, Hernandez-Koutoucheva A, Sharma AN, Bordeleau E, Pawlowski AC, Zubyk HL, Dooley D, Griffiths E, Maguire F, Winsor GL, Beiko RG, Brinkman FSL, Hsiao WWL, Domselaar GV, McArthur AG. 2020. CARD 2020: antibiotic resistome surveillance with the comprehensive antibiotic resistance database. *Nucleic Acids Res* 48:D517-D525.
45. Wang X, Zorraquino V, Kim M, Tsoukalas A, Tagkopoulos I. 2018. Predicting the evolution of *Escherichia coli* by a data-driven approach. *Nat Commun* 9:3562.

Table 1: Known AMR genes identified by the k-mer-based AMR classifiers

Antibiotic	Drug Class	Known AMR genes to the antibiotic	Known AMR genes associated to other antibiotics
Ampicillin	beta-lactam	TEM-1**, CTX-M-15, yicJ*	sul1**, folP**, APH(3'')-Ib, katE*, yadV*, arnC, fsr, nmpC, pepT,

			yeeJ, yhdJ
Aztreonam	beta-lactam	CTX-M-55*	AAC(6')-Ib-cr, acrD, catIII, nmpC, pitA, yicl, cpdB, yoaE, rapA, dinG, yeeJ, oppA, arnC
Cefepime	beta-lactam	CTX-M-1**, CTX-M-15, CTX-M-55	dfrA25*, AAC(6')-Ib10*, AAC(3)-IId, catB3, AAC(6')-Ib-cr, folA*, yadV*, citF, yeeJ, ftsI
Cefoxitin	beta-lactam	CMY-2*, ybiW*, betT, chiP, cra, envZ, htrE, lyxK, mdIA, yeeJ, yghA	dfrA25, AAC(3)-IId, catIII, blc, yaiY, folA, putA, lpoA
Ciprofloxacin	fluoroquinolone	gyrA**,	OXA-1*, CTX-M-15*, arnC, nmpC, htrE, cpdB, arcA, flu
Gentamicin	aminoglycoside	AAC(3)-IId**, AAC(6')-Ib7**, aadA13*, AAC(3)-Ile*, AAC(6')-Ib9*, aadA7, ANT(2'')-Ia	floR, CTX-M-15, dfrA17, mphA, intS*, fliC*, arnC, yicJ
Levofloxacin	fluoroquinolone		gyrA**, lacI*, yqiK, flu, arcA, fimC, phoE, ybiH, dadA
Tetracycline	tetracycline	tet(A)**, tet(B)**, mdfA	APH(6)-Id, sul2, yeeJ, folP, csiD
Tobramycin	aminoglycoside	AAC(3)-IId**, AAC(6')-Ib-cr**, AAC(3)-Ile, AAC(6')-Ib7	catB3*, CTX-M-55, dfrA17, OXA-1, fliC*, pinR, ydfU, dnaQ
Trimethoprim	diaminopyrimidine		ANT(2'')-Ia**, sul2*, aadA16*, aadA25*, APH(3'')-Ib*, TEM-1, tet(A), APH(6)-Id, mphA, TEM-150, sul1, folP*, dosP, valS,

nmpC, htrE, groL, putP

Genes in the top 10% features, ranked according to their maximum feature important assigned by the GBC classifier, are presented only.

***Gene was associated to feature in the top 10*

**Gene was associated to feature in the top 50*

Table 2: Known AMR genes identified by the SNP-based AMR classifiers

Antibiotic	Known AMR genes to the antibiotic	Known AMR genes associated to other antibiotics
Ciprofloxacin	gyrA**, parC**, parE*, typA*, hofN, valS, pnp, gyrB	speB**, yegU*, ugpB*, ampH*, fhuB*, poxB*, gss*, hybB*, phoE, speC, bgIX, ftnA, pphA, yjfF, yjaB, yjjV, hofQ, yidC, prmB, hisF, plaP, truC, gcvP, mltC, rstB, mtlD, folA, metH, rnd, waaA, upp, putP, yohK, aidB, yegQ, uvrB, trmH, ulaG, yqjG, cpxA, proC, uvrA, recJ, hflX, tamB, cysK, metC, nrdB, mutM, mpl, osmF, mrcA, dcd, ravA, pepD, yejA, ribC, cstA, yeiQ, nusA, hemA, yaiZ, hybF, mglA, ysaA, potA, hemY, yjjP, recG, yebY, aroC
Levofloxacin		parC**, gyrA**, hemF*, recG*, mysB*, metC*, tktA*, aceF*, yicR*, blgX*, fabD*, mutS*, chaA*, msyB*, rbsA*, gcvP, glnE, pcnB, mdtB, hisF, purT, menD, nikC, ftnA, frwB, yjiN, nadR, cyoB, fumC, mdtD, citG, glgX, valS, ldcC, yebQ, adiA
Meropenem		parC**, gyrA**, creC**, yrfF**, valS**, bgIX**, fucl*, hisF*, parE*, plaP*, nikA*, pykF*, aidB*, yjjG*, gcvP*, yjfF*, dsbD*, lepA*, thrA*, hybB, yccS, mdtB, murC, yegR, ravA, yjjV, yjjK, mscM, menD, mutS, metF, mglA, yjcD, nuoL, nadR, rplL, dusB, yegU, sufB, nudI, ulaG, ccmD, rnr, tamB, pdxA, dld, asd, ychO, soxR, yebK, nrdB, argD, baeS, glgX, osmF, trmI, yegS, dnaX, yejH, waaC, fhuE, aroP, folA, ycbZ, rbbA, polA, recJ, speC

Genes in the top 10% features, ranked according to their maximum contribution to the classifier, are presented only.

***Gene was associated to feature in the top 10*

**Gene was associated to feature in the top 50*

○ (Line 83) "bacterias" -> "bacteria's"

Corrected.

○ (Line 128) See above regarding known vs. correctly matched AMR genes.
See above.

○ (Line 200) "which strongly correlated to" -> "which are strongly correlated with"
Corrected.

○ (Line 244) See above regarding known vs. correctly matched AMR genes
See above.

○ (Line 273) Reference Supplement Table 1 for carbon sources tested
Corrected - we added a reference to Supplementary Table 3 see line 452.

○ (Line 457) See above regarding known vs. correctly matched AMR genes
See above.

○ (Line 528) "effects" -> "affects"
Corrected.

○ (Figure 1) Make plot arrangement and dimensions consistent with Figure 2
Corrected.

○ (Figure 3b) May be useful to also present ratios between shared vs. combined metabolic genes (i.e. Jaccard index for each pair)

We have edited Figure 3b to include the number of shared genes in the right triangular matrix, whilst the jaccard index for each pair are provided in the left triangular matrix, please see below.

Figure 3b (legend): (b) Heatmap showing the number of metabolic genes in the intersection (right triangular matrix) and the jaccard index (left triangular matrix) between pairs of antibiotic classes. The diagonal values correspond to the total number of metabolic genes in each class.

○ (Figure 3d) Update caption, currently reads "number of genes" but heatmap shows non-integer values
Corrected.

(d) Heatmap showing the normalised number of genes associated to each metabolic system. Note that the number of genes was normalised via column standardisation.

○ (Figure 5b-c) An alternative presentation of pie chart data that may better highlight which mechanisms/antibiotics are associated with which clusters is a heatmap or clustermap similar to Figure 3d, showing what fraction of each cluster is of a given mechanism/antibiotic.

Changed the new Figure 5b-c to heatmaps instead. Updated the Figure legend, please see below

Figure 5. Effect of genetic determinants on metabolite yields. (a) Bipartite network with genes and metabolites as nodes. A gene and metabolite are connected by an edge if the deletion of the gene blocks the metabolite production. Genes and metabolites are highlighted according to the cluster they were assigned to via the Networkx modularity algorithm. Note that the number of clusters in the figure was reduced by considering only those of size greater than 10. (b) Heatmap showing the metabolic systems associated to each of the 6 clusters. A gene was associated with a metabolic system, if at least 1 metabolite associated with the system could no longer be produced after the gene was deleted. (c) Heatmap showing the antibiotics associated with each cluster. Note that genes occurring in multiple antibiotics will be accounted for twice. Hierarchical clustering was applied to the rows of each heatmap (metabolic systems or antibiotic class) using the single linkage method and Euclidean distance as the metric.

O (Figure 6b-c) See comment on Figure 5b-c

Changed the new Figure 6b-c to heatmaps instead. Updated the Figure legend as below:

Figure 6. Effect of genetic determinants on reaction fluxes. (a) Bipartite network with genes and reactions as nodes. A gene and reaction are connected by an edge if the deletion of the gene reduces the reaction flux by at least 10%. Genes and reactions are highlighted according to the cluster they were assigned to via the Networkx modularity algorithm. Note that to reduce the initial size of the network, we only included clusters of size greater than 10. (b) Heatmap showing the metabolic systems associated to each of the 9 clusters. A gene was associated with a metabolic system, if the flux span of at least 1 reaction associated with the system was reduced after the gene was deleted. (c) Heatmap showing the antibiotics associated with each cluster. Note that genes occurring in multiple antibiotics will be accounted for twice. Hierarchical clustering was applied to the rows of each heatmap (metabolic systems or antibiotic class) using the single linkage method and Euclidean distance as the metric.

Reviewer #2 (Comments for the Author):

This article combines available knowledge, statistical inference and metabolic modeling to provide an increased understanding of specific metabolic processes that may contribute to confer resistance to specific antibiotics in *E. coli*. The work presented is overall creative and thought-provoking, and I feel it contributes an original and thorough analysis that could inform and inspire other researchers. However, I also think that there are multiple aspects of the writing that need substantial clarification and rephrasing:

1. A major point that I would like to bring up is that the rationale and hypothesis underlying the GSM analysis that is central to the paper is not clearly justified in the introduction and beginning of the result section, and barely justified later in the results. As a reader trying to follow the rationale of the approach, I found it hard to figure out how and why the ML and GSM analysis can inform each other. For example, at line 117, the authors mention that GSM "offers a way of mechanistically evaluating the genetic determinants identified using ML". It is not clear what it means to evaluate a genetic determinant. If I understand correctly, what you are evaluating is really the role of that gene (or even more precisely the presence/absence, or different variants of that gene) on the resistance phenotypes. Furthermore, it is not clear nor obvious why the deletion of a gene should inform the resistance phenotype. A mutation/k-mer pattern could be in principle associated with the increase of expression of a gene. So it is not clear (as somehow implied in the text) if and why the deletion of a gene whose variation is correlated with resistance would help inform the underlying mechanisms of resistance. It is entirely possible (and, apparently consistent with the findings) that gene deletions end up being informative, but - again - to me this was not self-evident at the start. I would expect the authors to revise their presentation of the underlying motivation and hypothesis with more details, and a more precise description of why they would expect GSM to be informative. A similar unsatisfactory description of this link is also appearing in the first section of the results, where, towards the end (lines 156-158) the notion that GSM lethality of AMR-related genes would yield interesting results is mentioned very briefly and without a real rationale. The first hint to a rationale (one of many possible) appears only at line 273-275.

Thank you for this comment. We agree that the original manuscript lacked motivation behind using the GSM and didn't clearly explain how knocking out a gene related to the SNPs or k-mers. To answer the question related to the rationale and hypothesis underlying the GSM analysis, please consider that the machine learning output lacks any biological interpretation for how the genetic determinants are related to resistance or susceptibility. Some cases, such as the beta-lactamases and drug efflux pumps, which are found via ML-classifiers can be simply linked to AMR since they allow for drug degradation and excretion of the drug. However, the majority of ML approaches lack the ability to investigate alleles that are associated with metabolic changes, which are advantageous for the bacteria to adapt to the antibiotic exposure and therefore important for revealing novel insights which could help develop drugs with higher efficacy. This limitation has also been addressed by Kavas et al. (2018) (<https://doi.org/10.1038/s41467-018-06634-y>). Genome scale models provide a way of predicting genotype-phenotype relationships at the system-level. These models include the information regarding the enzyme that is associated to the gene of interest and so by blocking the flux through this enzyme, we can predict how changes to the

genetic determinants may affect the entire metabolic network. A limitation of this approach, however, is that we do not know whether the highly discriminant SNPs and k-mers are related to an increase or decrease in activity, or complete loss function of the corresponding enzyme. A SNP may have caused changes in enzyme activity via changes to the ligand binding active site or may have resulted in a complete loss of function if the SNP resulted in unfolding the 3-D protein structure, for example. A k-mer may result in the same changes, whilst additionally it may be due to the absence/presence of a gene in resistant strains. Without this information however, we are just assuming the enzymatic activity changed, and investigate which the metabolic pathways are being affected as a result. (Note that we address this limitation in the discussion Lines 1270-1274, in response to Reviewer 1, also see below.) We therefore use gene knockout simulations, not to assume the enzymatic activity has been completely lost, but to identify which metabolic pathways are being disrupted if the genetic determinant was causing the associated reaction to change (simulated here by blocking the flux completely). This provides a means for linking the genetic variants found as highly discriminant of AMR phenotype by the ML-classifiers to biochemical pathways that are potentially being disrupted (they have potentially increased, decreased or been complete blocked). From this analysis, we can identify which genetic determinants are having the largest system level effect (i.e. effect the largest number of pathways and therefore targeting them for new drugs may have the largest disruption to the bacteria's metabolism), whilst we can also infer whether any genetic determinants are effecting similar metabolic processes, and therefore providing alternative routes for the bacteria to adapt for increasing resistance. In our results, for example, we found a number of genes that are suggested to effect iron metabolism, suggesting iron is particularly important for resistance. As discussed above, we can infer that iron metabolism is being affected but we can't determine if the activity of iron metabolism has increased, decreased or been completely blocked. By using more advanced GSMs, such as GEM-PRO, which take into account changes to protein structure, we could improve our method to infer this information regarding an increase or decrease in activity, which we are currently working on for our future work. We have updated the introduction lines 74-161, results lines 188-203 and lines 464-469 and the discussion lines 1017-1026 and 1269-1276 to make clearer the motivation and rationale behind the GSM-ML integration, also see below

Introduction – Lines 74-161: Antimicrobial resistance is a major threat to global health. Worryingly, a growing number of pathogens exhibit an extraordinary capacity for acquiring new antibiotic resistance traits in the bacteria population worldwide (1). New multi-drug resistance mechanisms have emerged and spread globally, resulting in current treatments becoming less effective against common bacteria, which cause severe and often deadly infections. Consequently, the development of new drugs and novel treatment strategies are urgently needed (2, 3).

The opportunistic pathogen *Escherchia coli* plays a major role in the AMR global health crisis. Firstly, the ability of *E. coli* to acquire resistance via single nucleotide polymorphisms (SNPs) in its existing genome (4-7) and via acquisition of resistance genes through horizontal gene transfer (HGT) from surrounding species (8-10) has resulted in increased levels of resistance

to many antibiotic classes, including penicillins, carbapenems, cephalosporins, fluoroquinolones, aminoglycosides and tetracyclines (11-15). Secondly, the ease of its transmission from humans and environmental sources has resulted in alarming numbers of multidrug resistant *E. coli* strains being reported worldwide (16, 17). Thirdly, the ease by which the bacteria can transfer genetic material via HGT, combined with the bacteria's ability to colonise different environments, including the gut where it has particularly close interaction with many other species, allows *E. coli* to act as a reservoir of AMR genes for other opportunistic pathogens, whilst also acquiring further resistance (18-21). For these reasons, the World Health Organization (WHO) have recently classified *E. coli* as a critical priority pathogen whereby the development of a new treatment is of high priority (22).

Recent advances in data generation and data mining, combined with machine learning (ML), have led to invaluable results in the identification of specific genomic markers which could be used to effectively predict resistant strains and to detect AMR genes (23-30). Most of these methods work to identify known AMR mutations giving rise to the phenotypic resistance. This has great potential for fast diagnostic evaluation of bacteria compared to laboratory methods. However, ML-based approaches offer further powerful opportunities compared to conventional methods as they allow for the genome-wide identification of truly novel features (i.e. k-mers and SNPs) ranked on strength of correlation with the resistance phenotype. Recently several studies have used these approaches (29, 30), which not only allow the identification of genes with known functional relationship with the resistance phenotype, but also allow the identification of genes which have no prior association to a specific resistance phenotype. This creates a path for generating non-intuitive testable hypotheses about the association of antibiotic resistance to a wider repertoire of genes, including deletions and functional mutations altering metabolism, and therefore provides a significant advantage in comparison with the conventional use of annotated gene databases.

Recent findings have shown the interconnectivity of antibiotic resistance with metabolism and emphasize the importance of considering this relationship in the design of new antibiotic regimens (31-33). Through its ease of HGT, *E. coli* has been able to adopt a highly flexible carbon and energy metabolism for adaption against stresses in niche environments (34, 35). For this reason, the bacteria is an ideal organism for investigating the interplay between AMR and metabolic adaption mechanisms. Connecting antimicrobial genes, specific mutations and alleles to metabolic phenotypes, however, still remains a significant challenge (36, 37). Black-box ML predictions lack biological interpretation of the genetic determinants (30), and therefore previous methods have often not accounted for the characterisation of new advantageous genetic variants occurring in targets beyond annotated drug resistant genes (29, 38), therefore neglecting important metabolic adaptations that allow resistance and tolerance to antibiotic stress (39-41).

A genome-scale metabolic model (GSM) offers a way of mechanistically evaluating the genetic determinants identified using ML. A GSM is a computational model of metabolism, which includes all known biochemical reactions and their corresponding gene-protein-reaction (GPR) rules. The GPR rules provide the important information linking genes to the reactions that are catalysed by the enzymes they encode and provides a means of simulating the metabolic system-level behaviour of the bacteria to perturbations in the gene. Whilst GSMs have proven invaluable tools for predicting genotype-phenotype relationships (42), they lack the power of machine learning algorithms (30). Recent studies have therefore been developing new approaches that integrate the power of ML with GSMs to allow for a mechanistic interpretation of the genetic associations discovered by machine learning, which offers a significant advantage over ML approaches alone (30, 43).

In this study, we developed a computational solution integrating the discriminant power of ML with GSM models to reveal the systemic relationships connecting the genetic determinants of AMR to important metabolic evolutionary adaptations in *E. coli*. Using our approach, we firstly were able to accurately predict AMR resistant and susceptible phenotypes against 12 different antibiotics, as well as identifying 225 (35 of which were matching to the specific antibiotic class reported in AMR-related databases) known AMR-conferring genes in 3616 *E. coli* strains. Secondly, by elucidating the effect of genetic discriminants on bacterial growth, metabolite yields and biochemical fluxes using the GSM, we were able to relate genetic determinants to a number of metabolic adaptation mechanisms, including reduced growth, alternative carbon source utilisation, changes to energy metabolism, iron metabolism, nucleotides metabolism and modifications to cell wall metabolism.

Results – Lines 188-203: The interconnectivity of antibiotic resistance, antimicrobial genes, specific mutations and alleles to metabolic phenotypes, as well as the identification of new advantageous genetic variants occurring in targets beyond annotated drug resistant genes was determined using the GSM (Supplementary Figure 1). Specifically, flux balance analysis (FBA), a constraint-based approach, was used to predict the effects of the genetic determinants on the metabolic network. Importantly, we considered the protein-coding regions only in the ML-classifiers, and therefore the genetic variants are potentially increasing or decreasing enzymatic activity, or in some cases completely block the function of the gene. Here, we evaluate the effect of each genetic determinant by blocking the flux through its corresponding enzyme, and assessed the propagation of this 'loss of function' through the entire metabolic network. Specifically, we used the GSM to predict the effect of each genetic determinant on bacterial growth, the production of individual metabolites and the feasible flux range through individual reactions. Changes to metabolic phenotype capabilities in each gene KO model (i.e. reduction in growth rate, reduced metabolite

production or reduction in flux span through a reaction) were assed using the wild type model of *E. coli* K-12 MG1655.

Results – lines 464-469-Y: Next, using the GSM, we investigated the system-level effect of each important gene on metabolism, beyond the pathways they are encoded for. To this aim, we blocked the flux through reactions associated with an important gene (gene knockout) and evaluated the metabolic processes that were being affected. In doing so, we can infer potential metabolic adaption mechanisms that can be linked to a change in gene function (i.e. down-regulation, over-expression or deletion).

Discussion – lines 1017-1026: To understand the mechanistic effect of these 289 genes, we used flux balance analysis (FBA) to predict the system-level metabolic changes that result from genetic variants of the genes (i.e. mutations or absense). More specifically, we predicted metabolic phenotypes of genetic variants via gene knockouts and identified the metabolic processes that were being affected. Importantly, using our new ML-FBA integrated approach, we could reveal interesting links between genes and potential metabolic adaption mechanisms that, importantly, were not identified using standard gene pathway enrichment analysis.

Discussion – lines 1269-1274: The characterisation of the AMR-associated SNPs, in respect to a reference genome such as K12 MG1655, and would allow us to link the specific amino acid substitutions or deletions to antibiotic resistance. 1D-3D Structure-function prediction analysis may then enable us to determine whether the SNPs result in a loss or gain of function which is directly integrated as constraints into models such as GEM-PRO. The effects of the SNPs on the genes (i.e. loss of function or gain of function) is not determined in our approach and if considered would allow further insights into the biological interpretation.

1. Partridge SR, Kwong SM, Firth N, Jensen SO. 2018. Mobile Genetic Elements Associated with Antimicrobial Resistance. *Clin Microbiol Rev* 31.
2. Alanis AJ. 2005. Resistance to antibiotics: are we in the post-antibiotic era? *Arch Med Res* 36:697-705.
3. Falagas ME, Bliziotis IA. 2007. Pandrug-resistant Gram-negative bacteria: the dawn of the post-antibiotic era? *Int J Antimicrob Agents* 29:630-6.
4. Yang Y, Mi J, Liang J, Liao X, Ma B, Zou Y, Wang Y, Liang J, Wu Y. 2019. Changes in the Carbon Metabolism of *Escherichia coli* During the Evolution of Doxycycline Resistance. *Front Microbiol* 10:2506.
5. Truong QC, Nguyen Van JC, Shlaes D, Gutmann L, Moreau NJ. 1997. A novel, double mutation in DNA gyrase A of *Escherichia coli* conferring resistance to quinolone antibiotics. *Antimicrob Agents Chemother* 41:85-90.

6. Toprak E, Veres A, Michel JB, Chait R, Hartl DL, Kishony R. 2011. Evolutionary paths to antibiotic resistance under dynamically sustained drug selection. *Nat Genet* 44:101-5.
7. Melnyk AH, Wong A, Kassen R. 2015. The fitness costs of antibiotic resistance mutations. *Evol Appl* 8:273-83.
8. Doi Y, Adams-Haduch JM, Peleg AY, D'Agata EM. 2012. The role of horizontal gene transfer in the dissemination of extended-spectrum beta-lactamase-producing *Escherichia coli* and *Klebsiella pneumoniae* isolates in an endemic setting. *Diagn Microbiol Infect Dis* 74:34-8.
9. Huddleston JR. 2014. Horizontal gene transfer in the human gastrointestinal tract: potential spread of antibiotic resistance genes. *Infect Drug Resist* 7:167-76.
10. Bajaj P, Singh NS, Viridi JS. 2016. *Escherichia coli* beta-Lactamases: What Really Matters. *Front Microbiol* 7:417.
11. Ramirez-Castillo FY, Moreno-Flores AC, Avelar-Gonzalez FJ, Marquez-Diaz F, Harel J, Guerrero-Barrera AL. 2018. An evaluation of multidrug-resistant *Escherichia coli* isolates in urinary tract infections from Aguascalientes, Mexico: cross-sectional study. *Ann Clin Microbiol Antimicrob* 17:34.
12. Sanchez S, McCrackin Stevenson MA, Hudson CR, Maier M, Buffington T, Dam Q, Maurer JJ. 2002. Characterization of multidrug-resistant *Escherichia coli* isolates associated with nosocomial infections in dogs. *J Clin Microbiol* 40:3586-95.
13. Saenz Y, Brinas L, Dominguez E, Ruiz J, Zarazaga M, Vila J, Torres C. 2004. Mechanisms of resistance in multiple-antibiotic-resistant *Escherichia coli* strains of human, animal, and food origins. *Antimicrob Agents Chemother* 48:3996-4001.
14. Aworh MK, Kwaga J, Okolocha E, Mba N, Thakur S. 2019. Prevalence and risk factors for multi-drug resistant *Escherichia coli* among poultry workers in the Federal Capital Territory, Abuja, Nigeria. *PLoS One* 14:e0225379.
15. Hassan R, Tantawy M, Gouda NA, Elzayat MG, Gabra S, Nabih A, Diab AA, El-Hadidi M, Bakry U, Shoeb MR, Elanany M, Shalaby L, Sayed AA. 2020. Genotypic characterization of multiple drug resistant *Escherichia coli* isolates from a pediatric cancer hospital in Egypt. *Sci Rep* 10:4165.
16. Aslam B, Wang W, Arshad MI, Khurshid M, Muzammil S, Rasool MH, Nisar MA, Alvi RF, Aslam MA, Qamar MU, Salamat MKF, Baloch Z. 2018. Antibiotic resistance: a rundown of a global crisis. *Infect Drug Resist* 11:1645-1658.
17. Organization) WWH. 2020. Global Antimicrobial Resistance and Use Surveillance System (GLASS) Report.

18. Rasheed MU, Thajuddin N, Ahamed P, Teklemariam Z, Jamil K. 2014. Antimicrobial drug resistance in strains of *Escherichia coli* isolated from food sources. *Rev Inst Med Trop Sao Paulo* 56:341-6.
19. Argudin MA, Deplano A, Meghraoui A, Dodemont M, Heinrichs A, Denis O, Nonhoff C, Roisin S. 2017. Bacteria from Animals as a Pool of Antimicrobial Resistance Genes. *Antibiotics (Basel)* 6.
20. Penders J, Stobberingh EE, Savelkoul PH, Wolfs PF. 2013. The human microbiome as a reservoir of antimicrobial resistance. *Front Microbiol* 4:87.
21. Sommer MOA, Dantas G, Church GM. 2009. Functional characterization of the antibiotic resistance reservoir in the human microflora. *Science* 325:1128-1131.
22. Organization) WWH. 2017. WHO publishes list of bacteria for which new antibiotics are urgently needed.
23. Boolchandani M, D'Souza AW, Dantas G. 2019. Sequencing-based methods and resources to study antimicrobial resistance. *Nat Rev Genet* 20:356-370.
24. Nguyen M, Long SW, McDermott PF, Olsen RJ, Olson R, Stevens RL, Tyson GH, Zhao S, Davis JJ. 2019. Using Machine Learning To Predict Antimicrobial MICs and Associated Genomic Features for Nontyphoidal *Salmonella*. *J Clin Microbiol* 57.
25. Hyun JC, Kavvas ES, Monk JM, Palsson BO. 2020. Machine learning with random subspace ensembles identifies antimicrobial resistance determinants from pan-genomes of three pathogens. *PLoS Comput Biol* 16:e1007608.
26. Moradigaravand D, Palm M, Farewell A, Mustonen V, Warringer J, Parts L. 2018. Prediction of antibiotic resistance in *Escherichia coli* from large-scale pan-genome data. *PLoS Comput Biol* 14:e1006258.
27. Davis JJ, Boisvert S, Brettin T, Kenyon RW, Mao C, Olson R, Overbeek R, Santerre J, Shukla M, Wattam AR, Will R, Xia F, Stevens R. 2016. Antimicrobial Resistance Prediction in PATRIC and RAST. *Sci Rep* 6:27930.
28. Nguyen M, Brettin T, Long SW, Musser JM, Olsen RJ, Olson R, Shukla M, Stevens RL, Xia F, Yoo H, Davis JJ. 2018. Developing an in silico minimum inhibitory concentration panel test for *Klebsiella pneumoniae*. *Sci Rep* 8:421.
29. Kavvas ES, Catoiu E, Mih N, Yurkovich JT, Seif Y, Dillon N, Heckmann D, Anand A, Yang L, Nizet V, Monk JM, Palsson BO. 2018. Machine learning and structural analysis of *Mycobacterium tuberculosis* pan-genome identifies genetic signatures of antibiotic resistance. *Nat Commun* 9:4306.

30. Kavas ES, Yang L, Monk JM, Heckmann D, Palsson BO. 2020. A biochemically-Interpretable machine learning classifier for microbial GWAS. *Nature Communications* 11:2580.
31. Dunphy LJ, Yen P, Papin JA. 2019. Integrated Experimental and Computational Analyses Reveal Differential Metabolic Functionality in Antibiotic-Resistant *Pseudomonas aeruginosa*. *Cell Syst* 8:3-14 e3.
32. Lopatkin AJ, Stokes JM, Zheng EJ, Yang JH, Takahashi MK, You L, Collins JJ. 2019. Bacterial metabolic state more accurately predicts antibiotic lethality than growth rate. *Nat Microbiol* 4:2109-2117.
33. Zampieri M, Zimmermann M, Claassen M, Sauer U. 2017. Nontargeted Metabolomics Reveals the Multilevel Response to Antibiotic Perturbations. *Cell Rep* 19:1214-1228.
34. Dobrindt U. 2005. (Patho-)Genomics of *Escherichia coli*. *Int J Med Microbiol* 295:357-71.
35. Alteri CJ, Mobley HL. 2012. *Escherichia coli* physiology and metabolism dictates adaptation to diverse host microenvironments. *Curr Opin Microbiol* 15:3-9.
36. Palmer AC, Kishony R. 2013. Understanding, predicting and manipulating the genotypic evolution of antibiotic resistance. *Nat Rev Genet* 14:243-8.
37. de Visser JA, Krug J. 2014. Empirical fitness landscapes and the predictability of evolution. *Nat Rev Genet* 15:480-90.
38. Abdollahi H, Mofid B, Shiri I, Razzaghdoust A, Saadipoor A, Mahdavi A, Galandooz HM, Mahdavi SR. 2019. Machine learning-based radiomic models to predict intensity-modulated radiation therapy response, Gleason score and stage in prostate cancer. *Radiol Med* 124:555-567.
39. Martinez JL, Rojo F. 2011. Metabolic regulation of antibiotic resistance. *FEMS Microbiol Rev* 35:768-89.
40. Cabral DJ, Wurster JI, Belenky P. 2018. Antibiotic Persistence as a Metabolic Adaptation: Stress, Metabolism, the Host, and New Directions. *Pharmaceuticals (Basel)* 11.
41. Bhargava P, Collins JJ. 2015. Boosting bacterial metabolism to combat antibiotic resistance. *Cell Metab* 21:154-155.
42. O'Brien EJ, Monk JM, Palsson BO. 2015. Using genome-scale models to predict biological capabilities. *Cell* 161:971-987.
43. Yang JH, Wright SN, Hamblin M, McCloskey D, Alcantar MA, Schrübbers L, Lopatkin AJ, Satish S, Nili A, Palsson BO, Walker GC, Collins JJ. 2019. A White-Box Machine

2. I am curious whether the authors considered using experimental gene deletion data to cross-validate some of the predictions used in the analysis. I understand that such data may not be available for all strains and conditions, but it may be nice to at least see a mention of how the availability of such data could impact your analysis or help refine it.

We would like to thank the reviewer for this suggestion. From this point, we realised that some genes did not always agree with the experimental data, but this was because the experimental data was carried out in rich media. We have therefore improved this section by running FBA knockout simulations under both rich and minimal media conditions. We validated the gene knockouts using KEIO collection for *E. coli* (63) for rich and using the study (4) for minimal media conditions. Our results are in good agreement with these results. 20 of the 289 genes were found in the GSM that were essential under rich media, 20 of which were also essential in the experimental data (true positives). Just 3 of the essential genes predicted by the GSM were not essential in the experimental results (false positives). A further 26 genes were identified that were essential under minimal media conditions. These were compared to the experimental results on minimal media for *E. coli* in (64). 25 of the 26 essential genes in the model were also essential in these *in vivo* results. We have updated Table 3 to include two columns. One for the essential genes in rich media, and another column for the essential genes in minimal media. Validated genes are highlighted in blue (true positives) and red (false positives). From this change in analysis, we also thought it would be interesting to identify whether there were specific substrates in the rich media that were restoring growth (i.e. were any of the gene KOs potentially leading to auxotrophic growth). To do this, we fixed glucose as the carbon source and looped through to allow for an additional carbon source to be available. We have therefore also added the following to the results lines 474-475 (updated title), results lines (480-539) the discussion lines 1028-1102, the methods lines 1452-1459 and a new Table 4 presenting the results for any gene in each antibiotic that restored growth using one additional carbon source, see also below. These results also show the new results based on the updated dataset of isolates. Table 3 is below, as well as a new Table 4 that shows the auxotrophy results.

Results (Title) – Lines 474-475: GSM knockout analysis reveals genes related to growth-limitation, auxotrophic behaviour and alternative carbon source utilisation

Results – lines 480-539: Identifying those that are essential for growth, whilst also being highly important in the ML models, may therefore provide a novel opportunity to selecting targets with dual-mechanism.

To this aim, the GSM was used to simulate the behaviour of *E. coli* with mutations in the 289 genes. Single gene deletions under rich environments conditions were carried out in iML1515 to mimic the effect of a 'loss of function' mutation on the entire system (see Material and Methods). Importantly, we found a total of 20 genes that were lethal to the

bacteria. These genes show a high level of agreement with *in vivo* gene essentiality study (63), as shown in Table 3. The lethal genes with the highest contribution (i.e. associated to the top 50 features) to the ML models, and therefore of greatest interest, included: *accA* and *metK* for ciprofloxacin, *fabD* and *fabG* for levofloxacin, *murG*, *lptG* and *mraY* for meropenem, *folP* for ampicillin and trimethoprim and *glmM* for gentamicin. These genes play essential roles in fatty acid elongation (*fabD*, *fabG* and *accA*), peptidoglycan metabolism (*murG*, *mraY* and *glmM*), lipopolysaccharides biosynthesis (*lptG*), S-adenosyl-L-methionine metabolism and folate metabolism (*folP*) (Figure 4). Importantly, *folP*, *lptG*, *fabG* and *murG* are already known AMR-conferring genes, as shown by Tables 1 and 2.

Next, we considered genes that were growth limiting under minimal media with glucose as the carbon source. We found an additional 26 genes that were essential under these conditions (Table 3), which again showed high agreement with the *in vivo* results (64). Under poor nutrient conditions of the host, changes in the function of these genes may contribute to slowing the growth rate, as before. However, if the environment is rich in nutrients, then a loss of function of these genes may have lead to advantageous auxotrophic behaviour. To test this hypothesis, we re-ran the KO simulations for growth on glucose, whilst also allowing for individual metabolites to be utilised. Importantly, we found that 17 of these genes could be linked to auxotrophic behaviour to the amino acids, including cysteine (meropenem, gentamicin), histidine (levofloxacin, ciprofloxacin, meropenem), phenyl-alanine (ciprofloxacin) and proline (ciprofloxacin). Auxotrophy for the vitamins thiamine (levofloxacin, tobramycin, meropenem), and panthanoate (ciprofloxacin) was also found. Auxotrophy to peptidoglycan precursors was also found for the antibiotics ciprofloxacin and meropenem, whilst purine and pyrimidine precursors were found for ciprofloxacin and cefepime. Importantly, auxotrophy for histidine and thiamine have previously been found to elevate fitness (65).

Discussion – Lines 1028-1102: Using the GSM, we found 20 genes essential for growth under rich environmental conditions. The essential genes with the highest importance in the ML-models may be promising targets for generating dual-mechanism antibiotics. That is, the antibiotic targets pathways that would lead to inhibition of an essential metabolic process, whilst simultaneously reducing the ability of the pathogen to adapt. The most promising new candidates as targets included: *accA* and *metK* for ciprofloxacin, *fabD* and *fabG* for levofloxacin, *murG*, *lptG* and *mraY* for meropenem, *folP* for ampicillin and *glmM* for gentamicin. Modifications to these genes may result in slower growth, which has previously been found advantageous to pathogenic bacteria, including *E. coli* and *Salmonella*, for reducing the damage that occurs as a result of being the primary target of antibiotics (60-62, 103). Alternatively, however, the genes *accA*, *fabG*, *fabD*, *lptG*, *murG*, *mraY* and *glmM* affect biosynthesis of cell wall components and therefore may have had an effect on membrane properties for antibiotic uptake or manipulation of the hosts immune response (73, 74). The gene *folP*, which is involved in folate metabolism, has previously been identified to prevent sulfonamide drugs from inhibiting folate metabolism (82). Importantly, however, we

identified *folP* in the trimethoprim, tetracycline and ampicillin ML models. Folate metabolism, including THF, however, are again important for nucleotides biosynthesis and have in fact been found important for persistence in *E. coli* cells exposed to ampicillin (83). Importantly, a number of additional genes affecting folate metabolism were also identified in the metabolite reproducibility analysis and flux variability analysis.

Interestingly, we also found a number of gene KOs which resulted in auxotrophic behaviour to a number of amino acids and vitamins, as well as peptidoglycan precursors and purine and pyrimidine precursors. The production of these metabolites are particularly energy intensive, and therefore their acquisition from the host may provide pathogens with a competitive fitness advantage against commensal bacteria (104). Alternatively, auxotrophy may have developed due to the critical role the metabolite plays in host-pathogen interactions. Using these genes as new drug targets has the disadvantage that the pathogen may be able to utilise exogeneous nutrients from the host environment.

Methods – Lines 1452-1459: We considered the essentiality of a gene under both rich media conditions and m9 minimal media conditions. To mimic rich media conditions, the model was constrained to allow all carbon sources into the system, with a fixed uptake rate of 1 mmol/gDCW/h. If a feasible solution exists, whilst maximising the biomass equation as the objective function, then the KO of the gene was not essential. To mimic m9 minimal media conditions, the model was constrained so one individual carbon source had a maximum uptake of 10 mmol/gDCW/h. This simulation (minimal media condition) was repeated for each carbon source in the model.

60. Pontes MH, Groisman EA. 2019. Slow growth determines nonheritable antibiotic resistance in *Salmonella enterica*. *Sci Signal* 12.
61. Brauner A, Fridman O, Gefen O, Balaban NQ. 2016. Distinguishing between resistance, tolerance and persistence to antibiotic treatment. *Nat Rev Microbiol* 14:320-30.
62. Greulich P, Scott M, Evans MR, Allen RJ. 2015. Growth-dependent bacterial susceptibility to ribosome-targeting antibiotics. *Mol Syst Biol* 11:796.
63. Baba T, Ara T, Hasegawa M, Takai Y, Okumura Y, Baba M, Datsenko KA, Tomita M, Wanner BL, Mori H. 2006. Construction of *Escherichia coli* K-12 in-frame, single-gene knockout mutants: the Keio collection. *Molecular systems biology* 2:2006.0008.
64. Joyce AR, Reed JL, White A, Edwards R, Osterman A, Baba T, Mori H, Lesely SA, Palsson BO, Agarwalla S. 2006. Experimental and computational assessment of conditionally essential genes in *Escherichia coli*. *J Bacteriol* 188:8259-71.

65. Seif Y, Choudhary KS, Hefner Y, Anand A, Yang L, Palsson BO. 2020. Metabolic and genetic basis for auxotrophies in Gram-negative species. *Proceedings of the National Academy of Sciences* 117:6264-6273.
82. Vedantam G, Guay GG, Austria NE, Doktor SZ, Nichols BP. 1998. Characterization of mutations contributing to sulfathiazole resistance in *Escherichia coli*. *Antimicrob Agents Chemother* 42:88-93.
83. Morgan J, Smith M, Mc Auley MT, Salcedo-Sora JE. 2018. Disrupting folate metabolism alters the capacity of bacteria in exponential growth to develop persisters to antibiotics. *bioRxiv* doi:10.1101/335505:335505.
103. Lee AJ, Wang S, Meredith HR, Zhuang B, Dai Z, You L. 2018. Robust, linear correlations between growth rates and beta-lactam-mediated lysis rates. *Proc Natl Acad Sci U S A* 115:4069-4074.
104. Juliao PC, Marrs CF, Xie J, Gilsdorf JR. 2007. Histidine auxotrophy in commensal and disease-causing nontypeable *Haemophilus influenzae*. *Journal of bacteriology* 189:4994-5001.

Table 3. *In silico* predicted gene lethality from the top-ranked discriminant genes in k-mer-based and SNP-based classifiers listed for each antibiotic.

Antibiotic	Essential genes (rich media)	Essential genes (glucose minimal media only)
Ampicillin	folP *	
Aztreonam		asd, purL
Cefepime		pyrF
Cefoxitin		
Ciprofloxacin	murJ *, lptG , hemG , ribC , accA *, cysG , aroC , waaA , hemA , metK *, lptF	purA , pheA , hisD , hisG , purL *, hisF , dapE , panD , purM , hisI *, ilvD , iscS , thiD , hisA , hisB *, hisH *, proC , purD
Levofloxacin	fabG *, fabD *	hisF *, purL *, thiD *
Gentamicin	glmM *, cysG	cysH

Meropenem	lptG* , mraY* , murG* , ispA	cysJ , hisD* , pdxA , hisC* , asd , hisF* , metF , murC , iscS , hisA , hisB , hisH* , hisG
Tetracycline	folP , murB	
Tobramycin		iscS
Trimethoprim	folP* , ftsI	

* Genes are associated with top 50 ranked features of antibiotic AMR model
 Genes highlighted in red were have not been found essential in experimental studies.
 Genes highlighted in blue did not have the experimental phenotype available for that environmental condition.

Table 4. *In silico* predicted genes knockouts that lead to auxotrophy from the top-ranked discriminant genes in k-mer-based and SNP-based classifiers listed for each antibiotic.

Antibiotic	Genes leading to specific auxotrophy
Ampicillin	
Aztreonam	
Cefepime	Pyrimidine compounds (pyrF)
Cefoxitin	
Ciprofloxacin	Phenyl-alanine (pheA), histidine (hisA , hisB* , hisD , hisF , hisG , hisI* , hisH*), pantothenate (panD), thiamine (iscS , thiD), proline (proC), nucleosides (purA), peptidoglycan precursors (dapE)
Levofloxacin	Histidine (hisF), thiamine (thiD*)
Gentamicin	Cysteine-derived compounds (cysH)
Meropenem	Histidine (hisA , hisB , hisD* , hisC* , hisF* , hisH*), S-Methyl-L-methionine (metF), thiamine (iscS), pyridoxine (pdxA), cysteine (cysJ), peptidoglycan precursors (murC)
Tetracycline	

Tobramycin Thiamine (*iscS*)

Trimethoprim

Update to Figure 4 legend: **Figure 4. An overview of the metabolic pathways involving potential gene targets for *E. coli*.** The genes *accA*, *lptG*, *fabD*, *fabG*, *murG*, *mraY*, *folP* and *metK* were all found as essential in the GSM of *E. coli*, whereas knockout of the genes *hisA* and *thiD* all resulted in auxotrophic behaviour. The genes *fucK*, *fucI*, *nupG*, *speB*, *uxaA*, *uxaB*, *dgoD*, *uidB* and *ttdB* were all found as essential to the growth on alternative carbon sources.

3. The phylogenetic analysis (starting at line 160) is potentially interesting but fairly disconnected from the rest of the manuscript. At first I was confused in trying to figure out whether and how that analysis informed the subsequent ML inference. I would suggest that the authors consider embedding that section differently in the flow of the manuscript, or at least make it very clear how it connects (or doesn't connect) to other portions.

We have added the phylogenetic analysis to only give a descriptive overview of the data used in this study. In particular, to show the genetic relatedness featuring the data to better appreciate the performance of the learners. However, we agree that the detail here from the phylogenetic tree analysis did not connect with the rest of the manuscript. We have therefore removed this section. The title of the results section starting line 205 and methods lines 1327-1328 have therefore changed, see all below

Results (Title) – Line 205: Genomic and metadata characteristics of the *E. coli* cohort

Methods (Title) – Lines 1327-1328: Genome assembly and annotation, in silico subtyping identification, pangenome construction and core genome alignment

4. Lines 256-259: It would be important to make portion clearer. If I understand correctly, SNPs are in (annotated) genes only, whereas k-mers could be anywhere in the genome. However it is not clear whether SNPs are only in the coding regions of the genes (it doesn't have to be the case). Also, I expect many genes to be non-metabolic and therefore not in the GSM, because only a portion (~1/4th?) of genes in *E. coli* are metabolic enzymes. So it is not clear what are the genes that the authors call "accessory genes": all the non-metabolic ones? In short, this whole part seems either misinformed about the connection between SNPs, k-mers and GSM, or just poorly explained.

As correctly pointed out by the Reviewer this part was poorly explained. The reviewer is correct that the SNPs are in (annotated) genes only, whereas k-mers could be anywhere in the genome. Please consider that we considered the core genome as conventionally defined as those genes present in all isolates, and an accessory genome, which includes the genes

absent from one or more isolates or unique to a given isolate (<https://doi.org/10.1371/journal.pcbi.1003788>).

The variable or accessory genome (also: flexible, dispensable genome) refers to genes not present in all strains of a species. So, our definition of accessory genes is not related to metabolic or not metabolic but to the way their presence/absence in the strains considered in our cohort. The pan-genome we considered in this study, consists of a core genome as conventionally defined as those genes present in all isolates and accessory genome as defined above. For extracting the complete pan-genome in this study we classified the annotated gene catalogue of the strain sets for each antibiotic into core (99% of the strains have this gene present in the genome) and accessory genes (<99% of the strains have this gene present in the genome) by using the default parameters in Roary version 3.13.0. Note that when we generated the pan-genome we used the protein-coding sequences, this is because we were interested in providing a global picture of the potentially functional alleles implicated in the AMR phenotypes and more importantly because the SNPs and k-mers identified as correlated to the AMR phenotypes by means of the machine learning approaches were then matched and tested for their functional role to the gene catalogue, which are associated to reactions in the iML1515 genome scale model of *E. coli*. Hence, we could only consider the coding regions and not the non-coding regions of the genomes to build up the pan-genome. We recognize that not accounting for the non-protein coding genes represent a limitation of our study as there are evidence in literature, such as *eis* and *rrs*, showing how non-protein coding genes can confer resistance. Likewise, we are not considering synonymous changes in the protein that also have been related to resistance (doi:[10.1038/ng.2743](https://doi.org/10.1038/ng.2743)). However, as previously explained our aim was to integrate the reactions in the iML1515 genome scale model of *E. coli*, we opted for only studying the protein-coding genes as also previously done by Kavvas et al., (2018) (<https://doi.org/10.1038/s41467-018-06634-y>). However, these types of computational platforms are open to account for non-protein coding genes and synonymous SNPs (which was also pointed out by Kavvas et al., (2018) in <https://doi.org/10.1038/s41467-018-06634-y>), and are actually the target of our future works. We have acknowledged this in the discussion.

Concerning the SNPs analysis done in this study, the SNPs were only searched in the coding regions of the pan-genome for the reasons explained above. In addition, the SNPs were only searched in the core genome (defined as those genes present in all isolates core genome) of the pan-genome because the SNPs variants across the strains could only be extracted via multiple alignment across the strains. In particular, to extract the SNPs variants each core gene nucleotide sequence had to be aligned, and single nucleotide variants were identified. The position of a SNP in a gene was selected as a feature in the machine learning if the nucleotide varied in more than 5% of isolates (i.e., was constant in less than 95% of strains). The features relate to the aligned position of single nucleotide variants. An allele (with a specific nucleotide variant) was included in the analysis if it was frequent in more than 5% of the population of isolates. Hence, such variation could only be determined if only including genes present in 100% of isolates (core genome) within the study population and by aligning the core genome (the genes present in all isolates). Since the accessory genome represents the genes not present in all strains, the 95%-5% variation criteria cannot be applied if using the accessory genome. For the k-mers analysis, these are lists of k-mers of length 13 bp which occurred in at least one of the genome files, which were generated for each antibiotic

using all the genomes associated to that antibiotic. Hence, for the k-mers we had less restrictions in how to capture them. The k-mers differently from the SNPs are 13 bp strings present in the genomes so their identification is done by estimating their presence/absence and consequently the presence/absence of the genes containing that string in each genome independently from all the other genomes in the cohort, while the SNPs are captured by their variation (i.e. if the nucleotide varied in more than 5% of strains). Hence, while for the SNPs we had to have the same gene content (core genome) in all strains to be able to make a comparative analysis of their variation across strains, the k-mers could be mapped anywhere in the genome (coding, non-coding, core or accessory) and each genome was independent from the others. This is why for the k-mers we could also incorporate the variable or accessory genome (also: flexible, dispensable genome) that includes genes not present in all strains of a species.

The important features (i.e. those in the top 10% of the features, ranked according to the maximum importance in the 50 runs of the GBC) from both the SNP and k-mer-based approaches were then matched to the gene catalogue, which are associated to reactions in the iML1515 genome scale model of *E. coli*. We agree that this is confusing as we don't properly explain this until the methods section. We have added some additional detail to the results lines 231-234, lines 340-342 and lines 407-425 and to the discussion lines 1005-1014 and lines 1258-1269 and to the methods lines 1334-1341 see also below

Results- Lines 231-234: Next, the pan-genome was extracted for the selected strains using the default parameters in Roary version 3.13.0 (47), which classified the catalogue of annotated genes as either core (i.e. occurring in >99% of strains) or accessory (i.e. missing from >99% of strains).

Results- Lines 340-342: The variant sites (SNPs) in the protein-coding genes of the core genome of the pan-genome were identified using SNPsites tool (www.github.com/sanger-pathogens/snp-sites) and used as the features in the GBC model for fitting AMR labels.

Results- Lines 407-425: The ratio of metabolic genes to total genes corresponding to the top ranked 10% of features was considerably higher for the SNPs-based models than the k-mer based models (Figure 3a, Supplementary Figure 3). The percentage of metabolic genes accounted for in iML1515 from the top features, for example, ranged from 43% (ciprofloxacin) to 48% (levofloxacin) in SNP-based AMR models. The percentage of metabolic genes identified by the k-mer based approach and present in iML1515, were considerably lower, ranging from 5% (tobramycin) to 19% (levofloxacin). A large number of genes identified by the k-mer approach, however, were from the accessory genome, which currently lack many functional annotations, as shown in Supplementary Figure 4 (see the decrease in cyan bars to yellow bars). Additionally, since the genome scale model is based on the K-12 strain, accessory genes missing from this reference genome will not be included in the analysis. Nevertheless, a total of 289 genes present in iML1515 were identified by combining the genes that were associated to the top ranked 10% of genes in the two machine learning approaches, which motivates the integration with GSMM analysis.

The contribution of genes from each AMR classifier ranged between 1 (tobramycin) and 123 (ciprofloxacin), with a small number of genes overlapping between antibiotic AMR models (Figure 3b).

Discussion – Lines 1005-1014: Importantly, a number of the genes identified in both the k-mer and SNP-based models were associated to metabolic reactions. Using the GSM iML1515, we found a total of 289 metabolic genes from the top 10% of features from both the k-mer and SNP-based models. The number of metabolic genes from the SNP-based models was considerably higher than the number of metabolic genes from the k-mer-based models. This is not too surprising however, since the k-mer-based approach included the important accessory genes responsible for drug target modifications, drug efflux and enzymatic inhibition. Metabolic-specific mutations provide a secondary adaptation mechanism to reduce the antibiotic efficacy. Importantly, previous studies have also found that metabolic-specific mutations are present in the core genes of *E. coli* (102).

Discussion: 1258-1269: Furthermore, our approach was limited to protein-coding genes only, and therefore lacks the ability to identify important non-protein coding regions, which have previously been found to confer resistance, such as *eis* and *rrs* (113). Likewise, we are not considering synonymous changes in the protein that also have been related to resistance (114). However, as also pointed out by Kavas (2018) (29) these types of computational platforms are open to account for non-protein coding genes and synonymous SNPs in future work. Using more advanced GSM frameworks, such as regulatory FBA (115) and GEM-PRO (116), for example, would allow us to investigate the effect of genetic determinants on metabolic phenotypes via changes to gene regulation and protein structure.

Methods- Lines 1334-1341: Each core gene nucleotide sequence was further aligned, and single nucleotide variants were identified. The position of a SNP in a gene was selected as a feature in the machine learning if the nucleotide varied in more than 5% of strains (i.e., was constant in less than 95% of strains). Such variation could only be determined if only including genes present in 100% of isolates (core genome) within the study population and by aligning the core genome (the genes present in all isolates). This is why we only considered the core genome of the pan-genome for this analysis.

29. Kavas ES, Catoiu E, Mih N, Yurkovich JT, Seif Y, Dillon N, Heckmann D, Anand A, Yang L, Nizet V, Monk JM, Palsson BO. 2018. Machine learning and structural analysis of *Mycobacterium tuberculosis* pan-genome identifies genetic signatures of antibiotic resistance. *Nat Commun* 9:4306.

47. Page AJ, Cummins CA, Hunt M, Wong VK, Reuter S, Holden MT, Fookes M, Falush D, Keane JA, Parkhill J. 2015. Roary: rapid large-scale prokaryote pan genome analysis. *Bioinformatics* 31:3691-3.
102. Lopatkin AJ, Bening SC, Manson AL, Stokes JM, Kohanski MA, Badran AH, Earl AM, Cheney NJ, Yang JH, Collins JJ. 2021. Clinically relevant mutations in core metabolic genes confer antibiotic resistance. *Science* 371.
113. Kambli P, Ajbani K, Nikam C, Sadani M, Shetty A, Udwadia Z, Georghiou SB, Rodwell TC, Catanzaro A, Rodrigues C. 2016. Correlating rrs and eis promoter mutations in clinical isolates of *Mycobacterium tuberculosis* with phenotypic susceptibility levels to the second-line injectables. *International journal of mycobacteriology* 5:1-6.
114. Safi H, Lingaraju S, Amin A, Kim S, Jones M, Holmes M, McNeil M, Peterson SN, Chatterjee D, Fleischmann R, Alland D. 2013. Evolution of high-level ethambutol-resistant tuberculosis through interacting mutations in decaprenylphosphoryl- β -D-arabinose biosynthetic and utilization pathway genes. *Nature genetics* 45:1190-1197.
115. Covert MW, Schilling CH, Palsson B. 2001. Regulation of Gene Expression in Flux Balance Models of Metabolism. *Journal of Theoretical Biology* 213:73-88.
116. Lu H, Li F, Sánchez BJ, Zhu Z, Li G, Domenzain I, Marcišauskas S, Anton PM, Lappa D, Lieven C, Beber ME, Sonnenschein N, Kerkhoven EJ, Nielsen J. 2019. A consensus *S. cerevisiae* metabolic model Yeast8 and its ecosystem for comprehensively probing cellular metabolism. *Nature Communications* 10:3586.

5. Lines 289-290: The opening sentence of this paragraph doesn't make sense to me, making the whole paragraph a bit shaky.

We agree that the above sentence is very unclear. We have revised the results lines 541-553 to make motivation behind the analysis clearer and also to show the changes in the results using the new updated dataset. Similarly, we have also updated the discussion lines 1095-1102 and the new Table 4.

Results - Lines 541 – 553: Additionally, gene modifications that affect the utilisation of alternative carbon sources was also investigated. Alternative carbon source utilisation has been found advantageous for pathogenic survival of bacteria including *E. coli*, *Salmonella*, *Vibrio cholerae* and *Campylobacter jejuni* (66-68). To this aim, we used the GSM to test the effect of the 289 genes on the 297 different carbon sources in the iML1515 model. Single gene knockouts were repeated for each individual carbon source, under minimal media conditions. We found 39 genes whose deletion blocked growth on a variety of alternative carbon source (Table 5). The carbon sources that were blocked by the genes with the highest importance (i.e. associated to the top 50 features) in the ML models included: fucose (cefoxitin and meropenem), galactonate (cefoxitin), tartrate (levofloxacin), agmatine

(ciprofloxacin), galacturonate (ciprofloxacin and levofloxacin), methyl-beta-D-glucuronate (cefoxitin) and a variety of nucleosides (ciprofloxacin).

Discussion – Lines 1095-1102: Additionally, we identified 39 genes whose KO affected the growth of *E. coli* on for growth on alternative carbon sources. The genetic determinants with the highest importance in the ML-models affected growth on various carbohydrates. Interestingly, a previous study found that various carbohydrates, including fucose, promote natural transformation of *E. coli*, therefore potentially contributing to the acquisition of antibiotic resistance and virulence (105). Fucose is particularly interesting as it has also been found to positively regulate microbiome bacterial colonisation and host immune activation (106).

66. Hibbing ME, Fuqua C, Parsek MR, Peterson SB. 2010. Bacterial competition: surviving and thriving in the microbial jungle. *Nat Rev Microbiol* 8:15-25.
67. Sorbara MT, Pamer EG. 2019. Interbacterial mechanisms of colonization resistance and the strategies pathogens use to overcome them. *Mucosal Immunol* 12:1-9.
68. Fabich AJ, Jones SA, Chowdhury FZ, Cernosek A, Anderson A, Smalley D, McHargue JW, Hightower GA, Smith JT, Autieri SM, Leatham MP, Lins JJ, Allen RL, Laux DC, Cohen PS, Conway T. 2008. Comparison of carbon nutrition for pathogenic and commensal *Escherichia coli* strains in the mouse intestine. *Infect Immun* 76:1143-52.
105. Guo M, Wang H, Xie N, Xie Z. 2015. Positive Effect of Carbon Sources on Natural Transformation in *Escherichia coli*: Role of Low-Level Cyclic AMP (cAMP)-cAMP Receptor Protein in the Derepression of *rpoS*. *Journal of Bacteriology* 197:3317-3328.
106. Pickard JM, Chervonsky AV. 2015. Intestinal fucose as a mediator of host–microbe symbiosis. *The Journal of Immunology* 194:5588-5593.

Table 5. *In silico* predicted essential genes on specific carbon sources from the top-ranked discriminant genes in k-mer-based and SNP-based classifiers listed for each antibiotic.

Antibiotic	Lethal genes for growth on specific carbon sources important in AMR model
------------	---

Ampicillin	gatC, mhpB
Aztreonam	adiC, yihP, cpdB, garD, mngB, paaK
Cefepime	
Cefoxitin	garD, kgtP, fucK*, ulaC, putA, fecA, mngB, uidB*, dgoD*
Ciprofloxacin	malF, ulaG, nupG*, nanE, deoA, pepD, deoC, tonB, nanA, mtlD, xylA, uxaA*, putP, speB*, mngB, cpdB, lamB
Levofloxacin	adiC, ttdT, uxuB, uxuA*, ttdB*
Gentamicin	hcaB
Meropenem	manZ, adiC, ulaG, exuT, fucI*
Tetracycline	
Tobramycin	
Trimethoprim	putP, emrE

6. Lines 307-308: There is no effect of a gene on a metabolite. Again, the setup of this portion could be much clearer with a sharper and more rigorous opening sentence. I gather from reading on (especially the methods) that the author meant effect of a gene on the producibility of a metabolite. Also: what metabolites? All metabolites or biomass components only? What is the rationale for either choice? These points can be clarified with minor sentence tweaks, but they can greatly enhance clarity.

We again agree that this paragraph lacked clarity. Here, we apply the analysis to compare the difference between the maximum production (or maximum theoretical yield) of a metabolite in the wild type model against the knockout model. We then construct a bipartite network, which connects a gene to a metabolite, if the metabolite is produced (i.e. $MTY > 0$) in the wild type model but its production is blocked when the gene is knocked out ($MTY = 0$). We carried out this analysis on all metabolites in the model, which included biomass metabolites. We have edited/extended the results lines 559-605 to hopefully make this clearer and added some rationale for considering all metabolites and reactions for the analysis.

Results - Lines 559-605: Next, the GSM was used to investigate whether the genetic determinants could be linked to additional metabolic adaption mechanisms, beyond those affecting the growth rate and alternative carbon utilisation. For this analysis, we examined the effect of each gene on metabolite reproducibility and reaction fluxes. More specifically,

we simulated single gene knockouts as before, however this time, we captured the effect on metabolite yields and flux spans (i.e. the variation of possible flux values for a given reaction) for all metabolites and reactions in the iML1515 model. The output of this analysis is twofold: i) identify clusters of genes that have similar metabolic phenotypes and ii) elucidate the metabolic adaptations that are important in providing bacteria with possible resistance to antibiotic stress. Genes that have a similar phenotype could give rise to higher variation of strains, whilst providing similar advantages for resistance (69). Determining the most important metabolic adjustments that provide resistance to antibiotic stress may provide useful information for the development of novel treatments. The genetic determinants that have the largest system-level impact, i.e. an increase or decrease in their functionality (modelled here via gene knockouts) disrupts the largest number of metabolite yields and/or reaction fluxes, could provide promising new targets.

69. Mayers DL, Lerner SA, Ouellette M, Sobel JD. 2009. Antimicrobial drug resistance. Humana Press, Totowa, N.J.

7. Prior work (Brynildsen et al., Nature Biotechnology volume 31, pages 160-165) had used extended FBA models that included the production of ROS to study in detail the metabolic processes associated with cell death upon antibiotic killing. I would expect the authors to comment on the relevance of this prior work to their study. Is there an overlap of emerging pathways, despite the distinct approaches? Would it be beneficial to extend the new methodology to an FBA model that explicitly includes ROS production?

We would like to thank the reviewer for this suggestion. To reflect the reviewers comment we have compared the results found in (Brynildsen et al., Nature Biotechnology volume 31, pages 160-165) to the significant pathways identified in our approach. Interestingly, both our work and their work identified genes involved in the electron transport chain as targets for improving antibiotic killing by increased ROS. We have therefore updated the results lines 1173-1194. In addition, we agree with the reviewer that It would be very interesting to use the iML1515-ROS model with the genetic determinants from the machine learning, to investigate further the role of resistance via reduction of damage via ROS. We think that would make a great future study for potentially identifying the most important genetic determinants that would increase antibiotic efficacy by targeting genes that induce ROS. We have edited the discussion lines 1280-1284 to acknowledge this in future work:

Results - Lines 1173-1194: Additionally, these genes also affect the production of important cofactors of energy metabolism, such as ATP, NAD and NADPH, which are important for the electron transport chain. Other ETC metabolites, including ubiquinone, menaquinone and flavin, were also being affected by the genes. Changes in the flux through ETC may contribute to antibiotic resistance via reduction of aminoglycoside uptake (110), increase in persister cells by reduced growth rate (60-62) and/or reduction in ROS. Importantly, a related study that applied gene KO simulations on an extended GSM of *E. coli*, which included

specific ROS-producing reactions, identified genes associated to the ETC as ROS-inducing targets for improved antibiotic killing (111).

Discussion - Lines 1280-1284: Furthermore, an extended version of iML1515 has been developed that includes ROS specific reactions (111). Applying the approach developed here to this model would therefore be useful future work for exploring the most important genetic determinants for improving antibiotic efficacy via ROS associated cell death (111).

60. Pontes MH, Groisman EA. 2019. Slow growth determines nonheritable antibiotic resistance in *Salmonella enterica*. *Sci Signal* 12.
61. Brauner A, Fridman O, Gefen O, Balaban NQ. 2016. Distinguishing between resistance, tolerance and persistence to antibiotic treatment. *Nat Rev Microbiol* 14:320-30.
62. Greulich P, Scott M, Evans MR, Allen RJ. 2015. Growth-dependent bacterial susceptibility to ribosome-targeting antibiotics. *Mol Syst Biol* 11:796.
110. Taber HW, Mueller JP, Miller PF, Arrow AS. 1987. Bacterial uptake of aminoglycoside antibiotics. *Microbiol Rev* 51:439-57.
111. Brynildsen MP, Winkler JA, Spina CS, MacDonald IC, Collins JJ. 2013. Potentiating antibacterial activity by predictably enhancing endogenous microbial ROS production. *Nature biotechnology* 31:160-165.

Minor points:

Line 83: bacterias -> bacteria's
Corrected.

Line 145: Provide references for gradient boosting classifier

We have updated the results lines 180-184 to include references for the GBC, see also below.

Results – Lines 180 - 184: A gradient boosting classifier (GBC) (44, 45) was chosen as it is a powerful approach to quickly and efficiently scan entire genomes against selected phenotypes, allowing for the identification of arbitrary numbers of genomic features ranked on strength of correlation with the antimicrobial resistant and susceptible phenotype.

42. Friedman JH. 2001. Greedy function approximation: A gradient boosting machine. *The Annals of Statistics* 29:1189-1232, 44.
43. Friedman JH. 1999. Stochastic Gradient Boosting
doi:<https://statweb.stanford.edu/~jhf/ftp/stobst.pdf>.

Items included in this submission:

- i. Response to reviewers;
- ii. Revised manuscript marked-up copy (Revised Article with Changes Highlighted.docx);
- iii. Revised manuscript clean copy (Manuscript.docx);
- iv. Supplementary Tables 1, 2 and 3 – new supplementary files with new results using the updated list of strains;
- v. Figure 1 – updated results from re-run of pipeline with updated datasets, and according to Reviewer 1's suggestion.
- vi. Figure 2 - updated results from re-run of pipeline with updated datasets
- vii. Figure 3 – updated results from re-run of pipeline with updated datasets;
- viii. Figure 4 – updated results from re-run of pipeline with updated datasets, and different method used based on suggestion from Reviewer 2
- ix. Figure 5 – updated results from re-run of pipeline with updated datasets
- x. Figure 6 - updated results from re-run of pipeline with updated datasets
- xi. Supplementary Figure 1 – updated figure based on small changes to pipeline (AUC cutoff increased to 0.95 from 0.8 and select top 10% of strains reduced from 20%.
- xii. Supplementary Figure 2 – updated results from re-run of pipeline with updated datasets;
- xiii. Supplementary Figure 3 (previously Supplementary Figure 4) – updated results from re-run of pipeline with updated datasets;
- xiv. Supplementary Figure 4 (new based on suggestion by Reviewer 1) – new figure showing the genes linked to metabolic pathways they were found associated to via the metabolite reproducibility analysis.
- xv. Supplementary Figure 5 (new based on suggestion by Reviewer 1) – new figure showing the genes linked to metabolic pathways they were found associated to via the FVA analysis.
- xvi. Supplemental Figure 6 (previously Supplementary Figure 5) – updated results from re-run of pipeline with updated datasets;
- xvii. Supplemental Figure 7 (previously Supplementary Figure 6) – updated results from re-run of pipeline with updated datasets;
- xviii. Supplementary Table 1 – updated results from re-run of pipeline with updated datasets;
- xix. Supplementary Table 2 – updated results from re-run of pipeline with updated datasets;
- xx. Supplementary Table 3 – updated results from re-run of pipeline with updated datasets;
- xxi. PATRIC_metadata.xlsx and code provided on the dropbox have been updated to include the new list of updated strains and inclusion of new analysis, and additional information as requested by the reviewers;

May 30, 2021

Dr. Tania Dottorini
University of Nottingham
Loughborough
United Kingdom

Re: mSystems00913-20R1 (Genome-scale metabolic models and machine learning reveal genetic determinants of antibiotic resistance in *Escherichia coli* and unravel the underlying metabolic adaptation mechanisms.)

Dear Dr. Tania Dottorini:

Thank you for submitting your revised manuscript to mSystems. We have completed our review and I am pleased to inform you that, your revisions have adequately addressed comments raised by both reviewers previously. Therefore, in principle, we expect to accept it for publication in mSystems. However, Reviewer #1 has provided additional minor comments. Please address them to render your manuscript completely acceptable for publication.

Preparing Revision Guidelines

For complete guidelines on revision requirements, please see the Instructions to Authors at <https://msystems.asm.org/sites/default/files/additional-assets/mSys-ITA.pdf>. **Submissions of a paper that does not conform to mSystems guidelines will delay acceptance of your manuscript.**

Sincerely,

Xiaoxia "Nina" Lin

Editor, mSystems

Journals Department
Reviewer comments:

Reviewer #1 (Comments for the Author):

This work presents a series of novel methods based around genome scale metabolic modeling to better understand the metabolic effects of E. coli genes associated with AMR as determined by machine learning (or potentially any GWAS method). These analyses are technically sound, distinct from other approaches towards integrating ML, FBA, and network analysis to understand AMR, and backed with detailed biological interpretations.

The authors have painstakingly addressed my previous comments with both a refined methodology and numerous new clarifications in text. Aside from only a few minor clarifications on updated results and typo corrections, this manuscript is considerably stronger.

Minor edits

(Line 101) "approaches" -> "approaches"

(Line 181) Expand abbreviation, first use of "KO" for knockout

(Line 293) Not sure what "decrease in cyan bars to yellow bars" is referring to, cannot find them in Supplementary Figure 4.

(Line 298) "GSMM" -> "GSM", or expand first use of GSMM abbreviation

(Line 341) "environmentsl" -> "environmental"

(Line 344) "genes that were lethal" -> "gene KOs that were lethal"

(Line 372) Reference Table 4, currently unreferenced in results

(Line 412) "The lethality of each genetic determinant on all metabolites" - not clear what this refers to, is this whether a gene KO completely disables the ability of a strain to produce a metabolite?

(Line 417) "genres" -> "genes"

(Line 433) "affect" -> "effect"

(Line 578) "affect" -> "effect"

(Line 740) "GSMM" -> "GSM"?

(Line 836) "GenenomeTester4" -> "GenomeTester4"

(Line 1014) Fig 1 legend: Clarify values shown. Does mean = mean from 5-fold CV, or across the 50 iterations? Do the boxplots show those 50 iterations?

(Line 1020) Fig 2 legend: Same questions as for Fig 1 legend.

(Line 1074) Fig 5 legend: Are unlabeled nodes metabolites?

(Line 1087) Flg 6 legend: Are unlabeled nodes reactions?

(Figure 3b) Add color bars. May be necessary to make separate scales for Jaccard indices and raw gene overlap counts, or reporting only Jaccard indices here and moving overlap counts to a supplementary figure or table.

(Figure 3c) Use different color schemes for gene functions vs antibiotics.

(Figure 5b-c/6b-c) Add color bars. May be worth normalizing counts to the total number of genes within each cluster to better highlight enrichment.

(Table 1) Not 100% sure, but should gyrA be listed as a known AMR gene for levofloxacin (switch column for that entry)?

(Table 2) Similar to Table 1, should parC and gyrA be listed as known AMR genes for levofloxacin?

(Table 4/5) Mention what asterisks indicate as in previous tables

Reviewer #2 (Comments for the Author):

The authors have addressed my concerns thoroughly, and have increased significantly the dataset used for their analysis, strengthening the results.

Dear Prof. Xiaoxia (Nina) Lin,

Manuscript # mSystems00913-20

"Genome-scale metabolic models and machine learning reveal genetic determinants of antibiotic resistance in Escherichia coli and unravel the underlying metabolic adaptation mechanisms" by Nicole Percy, Yue Hu, Michelle Baker, Alexandre Maciel Guerra, Ning Xue, Wei Wang, Jasmeet Kaler, Zixin Peng, Fengqin Li and Tania Dottorini.

We wish to thank the reviewers for their kind comments regarding our revised manuscript and their identification of minor corrections. We feel the manuscript is considerably stronger after their input.

In the following, a point-by-point response to all the minor comments is provided. The original questions from the reviewers are in blue, whilst our responses are in black.

Reviewer #1

Minor edits

(Line 101) "approaches" -> "approaches"

Corrected.

(Line 181) Expand abbreviation, first use of "KO" for knockout

Thank you for identifying. We have changed all occurrences of 'KO' to 'knockout' since for the majority we used the expanded version.

(Line 293) Not sure what "decrease in cyan bars to yellow bars" is referring to, cannot find them in Supplementary Figure 4.

Apologies, we had incorrectly referred to Supplementary Figure 4 instead of Supplementary Figure 3. We have corrected this in the revised manuscript.

(Line 298) "GSMM" -> "GSM", or expand first use of GSMM abbreviation

Corrected.

(Line 341) "environments!" -> "environmental"

Corrected.

(Line 344) "genes that were lethal" -> "gene KOs that were lethal"

Corrected.

(Line 372) Reference Table 4, currently unreferenced in results

Added reference to Table 4 to line 380.

(Line 412) "The lethality of each genetic determinant on all metabolites" - not clear what this refers to, is this whether a gene KO completely disables the ability of a strain to produce a metabolite?

We agree that the lethality of a gene knockout on the metabolites was not clearly defined. We have added the following sentence to the results section, lines 427 – 430, see also below

Results - Lines 427-430: The lethality of each genetic determinant on all metabolites in the iML1515 model was determined using flux balance analysis (FBA). A gene knockout was considered lethal to the production of a specific metabolite if it results in blocking the biosynthesis of the metabolite (see also Materials and Methods).

(Line 417) "genres" -> "genes"
Corrected.

(Line 433) "affect" -> "effect"
Corrected.

(Line 578) "affect" -> "effect"
Corrected.

(Line 740) "GSMM" -> "GSM"?
Corrected.

(Line 836) "GenenomeTester4" -> "GenomeTester4"
Corrected.

(Line 1014) Fig 1 legend: Clarify values shown. Does mean = mean from 5-fold CV, or across the 50 iterations? Do the boxplots show those 50 iterations?

We agree with Reviewer 1 that the boxplot was poorly explained by the figure legend. The boxplot shows the performance metrics for the 50 iterations. We incorrectly included the word 'mean' and so have revised the legend to the following:

Figure 1. K-mer-based supervised machine learning prediction of antibiotic resistance signature profiles to 12 antibiotics in the *E. coli* cohort. Boxplots showing the prediction performance results of the gradient boosting classifier for the 50 iterations. The performance indicators (Y axis) are accuracy, precision, recall, and AUC. Predictive models were generated to classify the resistance vs. susceptibility profiles of twelve different antibiotics (X axis).

(Line 1020) Fig 2 legend: Same questions as for Fig 1 legend.

As before, with Figure 1, we incorrectly included the word 'mean' in the legend and have therefore slightly modified the legend to the following:

Figure 2. SNP-based supervised machine learning prediction of antibiotic resistance signature profiles to 12 antibiotics in the *E. coli* cohort. Boxplots showing the prediction performance results of the gradient boosting classifier of the 50 iterations. The performance indicators (Y axis) are accuracy, precision, recall and AUC. Predictive models were generated to classify the resistance vs. susceptibility profiles of twelve different antibiotics (X axis).

(Line 1074) Fig 5 legend: Are unlabeled nodes metabolites?

Yes, the labelled nodes are genes and unlabelled nodes are metabolites as the reviewer correctly suggests. We thank the reviewer for identifying that this information was excluded. We have added the following to Figure legend 6:

Lines 1120-1121: Labelled nodes represent the genes, whereas unlabelled nodes represent metabolites.

(Line 1087) Fig 6 legend: Are unlabeled nodes reactions?

Yes, the labelled nodes are genes and unlabelled nodes are reactions as the reviewer correctly suggests. We thank the reviewer for identifying that this information was excluded. We have added the following to Figure legend 6:

Line 1136-1137: Labelled nodes represent the genes, whereas unlabelled nodes represent reactions.

(Figure 3b) Add color bars. May be necessary to make separate scales for Jaccard indices and raw gene overlap counts, or reporting only Jaccard indices here and moving overlap counts to a supplementary figure or table.

We agree that having the Jaccard index and the overlapping counts was difficult to show due to the different scales. We have therefore replaced Figure 3b with a heatmap that shows just the Jaccard index scores, as suggested by the Reviewer. The legend for Figure 3 has been updated accordingly, as shown below. We have also added an extra sheet, named 'Antibiotics vs metabolic genes', in Supplementary Table 3, which includes two tables. The first table, titled 'Metabolic genes identified in each AMR classifier', presents a matrix of the total 289 genes (rows) and 11 antibiotic classes (columns), which were used in the GSM analysis. If the *i*th gene was identified in the AMR classifier for the *j*th antibiotic class, then the *i*th, *j*th position of the matrix is 1, and 0 otherwise. The second table, titled 'Number of overlapping genes between antibiotic classes', shows the total number of common genes that were identified in the AMR classifiers for pairs of antibiotic classes. Note that this table includes the information that was previously in Figure 3b, which was previously combined with the Jaccard Index. The diagonal entries show the total number of genes in the genome scale model that were identified in each AMR classifier.

Figure 3. Number of metabolic genes occurring in the 11 AMR classifiers. (a) Bar chart showing proportions of metabolic genes compared to the entire set of genes found in each AMR model. The blue lines represent gene proportions from the k-mer AMR models, whereas the red lines represent gene proportions from the SNP AMR models (AUC>95%). (b) Heatmap showing the Jaccard index comparing the gene sets between two antibiotic classes. (c) Pie chart showing the proportions of genes associated with 10 metabolic systems (outer ring presented using the 'tab10' color theme in Matplotlib). The inner ring shows the proportion of genes from each antibiotic class associated to each metabolic system and is presented using the 'Set3' color theme in Matplotlib. Note that genes contributing to multiple antibiotic classifications will contribute multiple times in the pie chart, and therefore the total area of the pie chart does not amount to 289. (d) Heatmap showing the normalised number of genes associated to each metabolic system. Note that the number of genes was normalised via column standardisation. Note that hierarchical clustering was applied to both rows (metabolic systems) and columns (antibiotic classes) using the single linkage method and Euclidean distance as the metric. Each subplot shows the results for the top 10% of genes identified in each AMR classifier. Subplot B, C, and D show the results for the 289 genes found by combining the genes that correspond to the features in the top 10% of the k-mer and SNPs classifications.

(Figure 3c) Use different color schemes for gene functions vs antibiotics.

Edited the figure to use the 'tab10' matplotlib color theme for the outer ring and the 'Set3' matplotlib color theme for inner ring of pie chart. This information has also been updated in the legend for Figure 3, please also see this update in the response to the previous comment.

(Figure 5b-c/6b-c) Add color bars. May be worth normalizing counts to the total number of genes within each cluster to better highlight enrichment.

We have added color bars to Figure 5b-c and Figure 6b-c. We have also normalised the gene counts by the total number of genes in each cluster, as suggested by the Reviewer. We have also updated the figure legends for Figure 5 and 6 to the following. We had also missed colorbars off of Supplementary Figure 4B-C and Supplementary Figure 6B-C so have also updated these figures.

Figure 5. Effect of genetic determinants on metabolite yields. (a) Bipartite network with genes and metabolites as nodes. Labelled nodes represent genes, whereas unlabelled nodes represent metabolites. A gene and metabolite are connected by an edge if the deletion of the gene blocks the metabolite production. Genes and metabolites are highlighted according to the cluster they were assigned to via the Networkx modularity algorithm. The number of clusters in the figure was reduced by considering only those of size greater than 10. (b) Heatmap showing the metabolic systems associated to each of the 6 clusters. A gene was associated with a metabolic system, if at least 1 metabolite correlated with the system could no longer be produced after the gene was deleted. (c) Heatmap showing the antibiotics associated with each cluster. Note that genes occurring in multiple antibiotics were accounted for twice. Hierarchical clustering was applied to the rows of each heatmap (metabolic systems or antibiotic class) using the single linkage method and Euclidean distance as the metric. The gene counts have been normalised by the total number of genes in each cluster in each heatmap.

Figure 6. Effect of genetic determinants on reaction fluxes. (a) Bipartite network with genes and reactions as nodes. Labelled nodes represent the genes, whereas unlabelled nodes represent reactions. A gene and reaction are connected by an edge if the deletion of the gene reduces the reaction flux by at least 10%. Genes and reactions are highlighted according to the cluster they were assigned to via the Networkx modularity algorithm. Note that to reduce the initial size of the network, we only included clusters of size greater than 10. (b) Heatmap showing the metabolic systems associated to each of the 9 clusters. A gene was associated with a metabolic system, if the flux span of at least 1 reaction correlated with the system was reduced after the gene was deleted. (c) Heatmap showing the antibiotics associated with each cluster. Genes occurring in multiple antibiotics were accounted for twice. Hierarchical clustering was applied to the rows of each heatmap (metabolic systems or antibiotic class) using the single linkage method and Euclidean distance as the metric. The gene counts have also been normalised by the total number of genes in each cluster in each heatmap.

(Table 1) Not 100% sure, but should *gyrA* be listed as a known AMR gene for levofloxacin (switch column for that entry)?

Reviewer 1 is correct in that *gyrA* is a known AMR for fluoroquinolone antibiotics, which includes levofloxacin. We have corrected Table 1 to take this information into account.

(Table 2) Similar to Table 1, should *parC* and *gyrA* be listed as known AMR genes for levofloxacin?

Reviewer 1 is correct in that *gyrA* and *parC* are known AMR for fluoroquinolone antibiotics, which includes levofloxacin. We have corrected Table 2 to take this information into account.

(Table 4/5) Mention what asterisks indicate as in previous tables

Corrected.

Please note that we have also updated Figure 4 due to finding a metabolite incorrectly labelled. The reaction encoded by *hisI* was incorrectly presented as 'PRFAR' being produced from 'PRFAR', so we have updated this to now show 'PRFAR' being produced from 'PRBATP' (which is the abbreviation used for phosphoribosyl-ATP). We have updated the abbreviation list in the legend for Figure 3 to include this additional metabolite.

June 24, 2021

Dr. Tania Dottorini
University of Nottingham
Loughborough
United Kingdom

Re: mSystems00913-20R2 (Genome-scale metabolic models and machine learning reveal genetic determinants of antibiotic resistance in *Escherichia coli* and unravel the underlying metabolic adaptation mechanisms.)

Dear Dr. Tania Dottorini:

Your manuscript has been accepted, and I am forwarding it to the ASM Journals Department for publication. For your reference, ASM Journals' address is given below. Before it can be scheduled for publication, your manuscript will be checked by the mSystems senior production editor, Ellie Ghatineh, to make sure that all elements meet the technical requirements for publication. She will contact you if anything needs to be revised before copyediting and production can begin. Otherwise, you will be notified when your proofs are ready to be viewed.

We recognize that the video files can become quite large, and so to avoid quality loss ASM